# NOODL: Provable Online Dictionary Learning and Sparse Coding

**Sirisha Rambhatla[†], Xingguo Li[‡], and Jarvis Haupt[†]**

[†]Dept. of Electrical and Computer Engineering, University of Minnesota – Twin Cities, USA
[‡]Dept. of Computer Science, Princeton University, Princeton, NJ, USA
 Email: `rambh002@umn.edu`, `xingguol@cs.princeton.edu`, and `jdhaupt@umn.edu`

## Abstract

We consider the dictionary learning problem, where the aim is to model the given data as a linear combination of a few columns of a matrix known as a *dictionary*, where the sparse weights forming the linear combination are known as *coefficients*. Since the dictionary and coefficients, parameterizing the linear model are unknown, the corresponding optimization is inherently non-convex. This was a major challenge until recently, when provable algorithms for dictionary learning were proposed. Yet, these provide guarantees only on the recovery of the dictionary, without explicit recovery guarantees on the coefficients. Moreover, any estimation error in the dictionary adversely impacts the ability to successfully localize and estimate the coefficients. This potentially limits the utility of existing provable dictionary learning methods in applications where coefficient recovery is of interest. To this end, we develop NOODL: a simple Neurally plausible alternating Optimization-based Online Dictionary Learning algorithm, which recovers *both* the dictionary and coefficients *exactly* at a geometric rate, when initialized appropriately. Our algorithm, NOODL, is also scalable and amenable for large scale distributed implementations in neural architectures, by which we mean that it only involves simple linear and non-linear operations. Finally, we corroborate these theoretical results via experimental evaluation of the proposed algorithm with the current state-of-the-art techniques.

## 1 Introduction

Sparse models avoid overfitting by favoring simple yet highly expressive representations. Since signals of interest may not be inherently sparse, expressing them as a sparse linear combination of a few columns of a dictionary is used to exploit the sparsity properties. Of specific interest are overcomplete dictionaries, since they provide a flexible way of capturing the richness of a dataset, while yielding sparse representations that are robust to noise; see Mallat and Zhang (1993); Chen et al. (1998); Donoho et al. (2006). In practice however, these dictionaries may not be known, warranting a need to learn such representations – known as *dictionary learning* (DL) or *sparse coding* (Olshausen and Field, 1997). Formally, this entails learning an *a priori* unknown dictionary $\mathbf{A} \in \mathbb{R}^{n \times m}$ and sparse coefficients $\mathbf{x}^*_{(j)} \in \mathbb{R}^m$ from data samples $\mathbf{y}_{(j)} \in \mathbb{R}^n$ generated as

$$\mathbf{y}_{(j)} = \mathbf{A}^* \mathbf{x}^*_{(j)}, \ \|\mathbf{x}^*_{(j)}\|_0 \leq k \ \text{ for all } \ j = 1, 2, \ldots \tag{1}$$

This particular model can also be viewed as an extension of the low-rank model (Pearson, 1901). Here, instead of sharing a low-dimensional structure, each data vector can now reside in a separate low-dimensional subspace. Therefore, together the data matrix admits a *union-of-subspace* model. As a result of this additional flexibility, DL finds applications in a wide range of signal processing and machine learning tasks, such as denoising (Elad and Aharon, 2006), image inpainting (Mairal et al., 2009), clustering and classification (Ramirez et al., 2010; Rambhatla and Haupt, 2013; Rambhatla et al., 2016; 2017; 2019b;a), and analysis of deep learning primitives (Ranzato et al., 2008; Gregor and LeCun, 2010); see also Elad (2010), and references therein.

Notwithstanding the non-convexity of the associated optimization problems (since both factors are unknown), alternating minimization-based dictionary learning techniques have enjoyed significant success in practice. Popular heuristics include regularized least squares-based (Olshausen and Field, 1997; Lee et al., 2007; Mairal et al., 2009; Lewicki and Sejnowski, 2000; Kreutz-Delgado et al., 2003), and greedy approaches such as the method of optimal directions (MOD) (Engan et al., 1999) and k-SVD (Aharon et al., 2006). However, dictionary learning, and matrix factorization models in general, are difficult to analyze in theory; see also Li et al. (2016a).

To this end, motivated from a string of recent theoretical works (Gribonval and Schnass, 2010; Jenatton et al., 2012; Geng and Wright, 2014), provable algorithms for DL have been proposed recently to explain the success of aforementioned alternating minimization-based algorithms (Agarwal et al., 2014; Arora et al., 2014; 2015). However, these works exclusively focus on guarantees for dictionary recovery. On the other hand, for applications of DL in tasks such as classification and clustering – which rely on coefficient recovery – it is crucial to have guarantees on coefficients recovery as well.

Contrary to conventional prescription, a sparse approximation step after recovery of the dictionary does not help; since any error in the dictionary – which leads to an error-in-variables (EIV) (Fuller, 2009) model for the dictionary – degrades our ability to even recover the support of the coefficients (Wainwright, 2009). Further, when this error is non-negligible, the existing results guarantee recovery of the sparse coefficients only in $\ell_2$-norm sense (Donoho et al., 2006). As a result, there is a need for scalable dictionary learning techniques with guaranteed recovery of both factors.

## 1.1 SUMMARY OF OUR CONTRIBUTIONS

In this work, we present a simple online DL algorithm motivated from the following regularized least squares-based problem, where $S(\cdot)$ is a nonlinear function that promotes sparsity.

$$\min_{\mathbf{A}, \left\{\mathbf{x}_{(j)}\right\}_{j=1}^p} \sum_{j=1}^p \|\mathbf{y}_{(j)} - \mathbf{A}\mathbf{x}_{(j)}\|_2^2 + \sum_{j=1}^p S(\mathbf{x}_{(j)}). \tag{P1}$$

Although our algorithm does not optimize this objective, it leverages the fact that the problem (P1) is convex w.r.t $\mathbf{A}$, given the sparse coefficients $\{\mathbf{x}_{(j)}\}$. Following this, we recover the dictionary by choosing an appropriate gradient descent-based strategy (Arora et al., 2015; Engan et al., 1999). To recover the coefficients, we develop an iterative hard thresholding (IHT)-based update step (Haupt and Nowak, 2006; Blumensath and Davies, 2009), and show that – given an appropriate initial estimate of the dictionary and a mini-batch of $p$ data samples at each iteration $t$ of the online algorithm – alternating between this IHT-based update for coefficients, and a gradient descent-based step for the dictionary leads to geometric convergence to the true factors, i.e., $\mathbf{x}_{(j)} \rightarrow \mathbf{x}_{(j)}^*$ and $\mathbf{A}_i^{(t)} \rightarrow \mathbf{A}_i^*$ as $t \rightarrow \infty$.

In addition to achieving exact recovery of both factors, our algorithm – Neurally plausible alternating Optimization-based Online Dictionary Learning (NOODL) – has linear convergence properties. Furthermore, it is scalable, and involves simple operations, making it an attractive choice for practical DL applications. Our major contributions are summarized as follows:

- **Provable coefficient recovery:** To the best of our knowledge, this is the first result on *exact* recovery of the sparse coefficients $\{\mathbf{x}_{(j)}^*\}$, including their support recovery, for the DL problem. The proposed IHT-based strategy to update coefficient under the EIV model, is of independent interest for recovery of the sparse coefficients via IHT, which is challenging even when the dictionary is known; see also Yuan et al. (2016) and Li et al. (2016b).
- **Unbiased estimation of factors and linear convergence:** The recovery guarantees on the coefficients also helps us to get rid of the bias incurred by the prior-art in dictionary estimation. Furthermore, our technique geometrically converges to the true factors.
- **Online nature and neural implementation:** The online nature of algorithm, makes it suitable for machine learning applications with streaming data. In addition, the separability of the coefficient update allows for distributed implementations in neural architectures (only involves simple linear and non-linear operations) to solve large-scale problems. To showcase this, we also present a prototype neural implementation of NOODL.

In addition, we also verify these theoretical properties of NOODL through experimental evaluations on synthetic data, and compare its performance with state-of-the-art provable DL techniques.

## 1.2 RELATED WORKS

With the success of the alternating minimization-based techniques in practice, a push to study the DL problem began when Gribonval and Schnass (2010) showed that for $m = n$, the solution pair $(\mathbf{A}^*, \mathbf{X}^*)$ lies at a local minima of the following non-convex optimization program, where $\mathbf{X} = [\mathbf{x}_{(1)}, \mathbf{x}_{(2)}, \ldots, \mathbf{x}_{(p)}]$ and $\mathbf{Y} = [\mathbf{y}_{(1)}, \mathbf{y}_{(2)}, \ldots, \mathbf{y}_{(p)}]$, with high probability over the randomness of the coefficients,

$$\min_{\mathbf{A}, \mathbf{X}} \|\mathbf{X}\|_1 \quad \text{s.t.} \ \mathbf{Y} = \mathbf{A}\mathbf{X}, \quad \|\mathbf{A}_i\| = 1, \forall \, i \in [m]. \tag{2}$$

**Table 1:** Comparison of provable algorithms for dictionary learning.

| Method | Conditions | | | Recovery Guarantees | |
|---|---|---|---|---|---|
| | Initial Gap of Dictionary | Maximum Sparsity | Sample Complexity | Dictionary | Coefficients |
| NOODL (this work) | $\mathcal{O}^*\left(\frac{1}{\log(n)}\right)$ | $\mathcal{O}^*\left(\frac{\sqrt{n}}{\mu\log(n)}\right)$ | $\widetilde{\Omega}\left(mk^2\right)$ | No bias | No bias |
| `Arora15("biased")`[†] | | | $\widetilde{\Omega}\left(mk\right)$ | $\mathcal{O}(\sqrt{k/n})$ | N/A |
| `Arora15("unbiased")`[†] | | | $\text{poly}(m)$ | *Negligible* bias [§] | N/A |
| Barak et al. (2015)[¶] | N/A | $\mathcal{O}(m^{(1-\delta)})$ for $\delta > 0$ | $n^{O(d)}/\text{poly}(k/m)$ | $\epsilon$ | N/A |
| Agarwal et al. (2014)[‡] | $\mathcal{O}^*(1/\text{poly(m)})$ | $\mathcal{O}\left(\sqrt[6]{n}/\mu\right)$ | $\Omega(m^2)$ | No bias | N/A |
| Spielman et al. (2012) (for $n \leq m$) | N/A | $\mathcal{O}(\sqrt{n})$ | $\widetilde{\Omega}(n^2)$ | No bias | N/A |

Dictionary recovery reported in terms of column-wise error. † See Section 5 for description. ‡ This procedure is not *online*. § The bias is not explicitly quantified. The authors claim it will be *negligible*. ¶ Here, $d = \Omega(\frac{1}{\epsilon}\log(m/n))$ for column-wise error of $\epsilon$.

Following this, Geng and Wright (2014) and Jenatton et al. (2012) extended these results to the overcomplete case ($n < m$), and the noisy case, respectively. Concurrently, Jung et al. (2014; 2016) studied the nature of the DL problem for $S(\cdot) = \|\cdot\|_1$ (in (P1)), and derived a lower-bound on the minimax risk of the DL problem. However, these works do not provide any algorithms for DL.

Motivated from these theoretical advances, Spielman et al. (2012) proposed an algorithm for the under-complete case $n \geq m$ that works up-to a sparsity of $k = O(\sqrt{n})$. Later, Agarwal et al. (2014) and Arora et al. (2014) proposed clustering-based provable algorithms for the overcomplete setting, motivated from MOD (Engan et al., 1999) and k-SVD (Aharon et al., 2006), respectively. Here, in addition to requiring stringent conditions on dictionary initialization, Agarwal et al. (2014) alternates between solving a quadratic program for coefficients and an MOD-like (Engan et al., 1999) update for the dictionary, which is too expensive in practice. Recently, a DL algorithm that works for almost linear sparsity was proposed by Barak et al. (2015); however, as shown in Table 1, this algorithm may result in exponential running time. Finally, Arora et al. (2015) proposed a provable online DL algorithm, which provided improvements on initialization, sparsity, and sample complexity, and is closely related to our work. A follow-up work by Chatterji and Bartlett (2017) extends this to random initializations while recovering the dictionary exactly, however the effect described therein kicks-in only in very high dimensions. We summarize the relevant provable DL techniques in Table 1.

The algorithms discussed above implicitly assume that the coefficients can be recovered, after dictionary recovery, via some sparse approximation technique. However, as alluded to earlier, the guarantees for coefficient recovery – when the dictionary is known approximately – may be limited to some $\ell_2$ norm bounds (Donoho et al., 2006). This means that, the resulting coefficient estimates may not even be sparse. Therefore, for practical applications, there is a need for efficient online algorithms with guarantees, which serves as the primary motivation for our work.

## 2 ALGORITHM

We now detail the specifics of our algorithm – NOODL, outlined in Algorithm 1. NOODL recovers both the dictionary and the coefficients exactly given an appropriate initial estimate $\mathbf{A}^{(0)}$ of the dictionary. Specifically, it requires $\mathbf{A}^{(0)}$ to be $(\epsilon_0, 2)$-close to $\mathbf{A}^*$ for $\epsilon_0 = \mathcal{O}^*(1/\log(n))$, where $(\epsilon, \kappa)$-closeness is defined as follows. This implies that, the initial dictionary estimate needs to be column-wise, and in spectral norm sense, close to $\mathbf{A}^*$, which can be achieved via certain initialization algorithms, such as those presented in Arora et al. (2015).

**Definition 1** (($\epsilon, \kappa$)-closeness). *A dictionary $\mathbf{A}$ is $(\epsilon, \kappa)$-close to $\mathbf{A}^*$ if $\|\mathbf{A} - \mathbf{A}^*\| \leq \kappa\|\mathbf{A}^*\|$, and if there is a permutation $\pi : [m] \to [m]$ and a collection of signs $\sigma : [m] \to \{\pm 1\}$ such that $\|\sigma(i)\mathbf{A}_{\pi(i)} - \mathbf{A}_i^*\| \leq \epsilon, \ \forall \ i \in [m]$.*

---

**Notation.** Given an integer $n$, we denote $[n] = \{1, 2, \ldots, n\}$. The bold upper-case and lower-case letters are used to denote matrices $\mathbf{M}$ and vectors $\mathbf{v}$, respectively. $\mathbf{M}_i$, $\mathbf{M}_{(i,:)}$, $\mathbf{M}_{ij}$, and $\mathbf{v}_i$ (and $\mathbf{v}(i)$) denote the $i$-th column, $i$-th row, $(i, j)$ element of a matrix, and $i$-th element of a vector, respectively. The superscript $(\cdot)^{(n)}$ denotes the $n$-th iterate, while the subscript $(\cdot)_{(n)}$ is reserved for the $n$-th data sample. Given a matrix $\mathbf{M}$, we use $\|\mathbf{M}\|$ and $\|\mathbf{M}\|_F$ as the spectral norm and Frobenius norm. Given a vector $\mathbf{v}$, we use $\|\mathbf{v}\|$, $\|\mathbf{v}\|_0$, and $\|\mathbf{v}\|_1$ to denote the $\ell_2$ norm, $\ell_0$ (number of non-zero entries), and $\ell_1$ norm, respectively. We also use standard notations $\mathcal{O}(\cdot), \Omega(\cdot)$ ($\widetilde{\mathcal{O}}(\cdot), \widetilde{\Omega}(\cdot)$) to indicate the asymptotic behavior (ignoring logarithmic factors). Further, we use $g(n) = \mathcal{O}^*(f(n))$ to indicate that $g(n) \leq Lf(n)$ for a small enough constant $L$, which is independent of $n$. We use $c(\cdot)$ for constants parameterized by the quantities in $(\cdot)$. $\mathcal{T}_\tau(z) := z \cdot \mathbb{1}_{|z| \geq \tau}$ denotes the hard-thresholding operator, where "$\mathbb{1}$" is the indicator function. We use $\text{supp}(\cdot)$ for the support (the set of non-zero elements) and $\text{sign}(\cdot)$ for the element-wise sign.

---

**Algorithm 1:** NOODL: Neurally plausible alternating Optimization-based Online Dictionary Learning.

---

**Input**: Fresh data samples $\mathbf{y}_{(j)} \in \mathbb{R}^n$ for $j \in [p]$ at each iteration $t$ generated as per (1), where
$|\mathbf{x}_i^*| \geq C$ for $i \in \text{supp}(\mathbf{x}^*)$. Parameters $\eta_A$, $\eta_x^{(r)}$ and $\tau^{(r)}$ chosen as per **A.5** and **A.6**. No. of iterations $T = \Omega(\log(1/\epsilon_T))$ and $R = \Omega(\log(1/\delta_R))$, for target tolerances $\epsilon_T$ and $\delta_R$.

**Output**: The dictionary $\mathbf{A}^{(t)}$ and coefficient estimates $\widehat{\mathbf{x}}_{(j)}^{(t)}$ for $j \in [p]$ at each iterate $t$.

**Initialize**: Estimate $\mathbf{A}^{(0)}$, which is $(\epsilon_0, 2)$-near to $\mathbf{A}^*$ for $\epsilon_0 = \mathcal{O}^*(1/\log(n))$

**for** $t = 0$ **to** $T - 1$ **do**

> **Predict: (Estimate Coefficients)**
>
> **for** $j = 1$ **to** $p$ **do**
>
>> **Initialize:** $\mathbf{x}_{(j)}^{(0)} = \mathcal{T}_{C/2}(\mathbf{A}^{(t)^\top} \mathbf{y}_{(j)})$    (3)
>>
>> **for** $r = 0$ **to** $R - 1$ **do**
>>
>>> **Update:** $\mathbf{x}_{(j)}^{(r+1)} = \mathcal{T}_{\tau^{(r)}}(\mathbf{x}_{(j)}^{(r)} - \eta_x^{(r)} \mathbf{A}^{(t)^\top}(\mathbf{A}^{(t)}\mathbf{x}_{(j)}^{(r)} - \mathbf{y}_{(j)}))$    (4)
>>
>> **end**
>
> **end**
>
> $\widehat{\mathbf{x}}_{(j)}^{(t)} := \mathbf{x}_{(j)}^{(R)}$ for $j \in [p]$
>
> **Learn: (Update Dictionary)**
>
> Form empirical gradient estimate: $\widehat{\mathbf{g}}^{(t)} = \frac{1}{p} \sum_{j=1}^p (\mathbf{A}^{(t)} \widehat{\mathbf{x}}_{(j)}^{(t)} - \mathbf{y}_{(j)}) \text{sign}(\widehat{\mathbf{x}}_{(j)}^{(t)})^\top$    (5)
>
> Take a gradient descent step: $\mathbf{A}^{(t+1)} = \mathbf{A}^{(t)} - \eta_A \, \widehat{\mathbf{g}}^{(t)}$    (6)
>
> Normalize: $\mathbf{A}_i^{(t+1)} = \mathbf{A}_i^{(t+1)}/\|\mathbf{A}_i^{(t+1)}\| \ \forall \ i \in [m]$

**end**

---

Due to the streaming nature of the incoming data, NOODL takes a mini-batch of $p$ data samples at the $t$-th iteration of the algorithm, as shown in Algorithm 1. It then proceeds by alternating between two update stages: coefficient estimation ("Predict") and dictionary update ("Learn") as follows.

**Predict Stage**: For a general data sample $\mathbf{y} = \mathbf{A}^*\mathbf{x}^*$, the algorithm begins by forming an initial coefficient estimate $\mathbf{x}^{(0)}$ based on a hard thresholding (HT) step as shown in (3), where $\mathcal{T}_\tau(z) := z \cdot \mathbb{1}_{|z| \geq \tau}$ for a vector $\mathbf{z}$. Given this initial estimate $\mathbf{x}^{(0)}$, the algorithm iterates over $R = \Omega(\log(1/\delta_R))$ IHT-based steps (4) to achieve a target tolerance of $\delta_R$, such that $(1 - \eta_x)^R \leq \delta_R$. Here, $\eta_x^{(r)}$ is the learning rate, and $\tau^{(r)}$ is the threshold at the $r$-th iterate of the IHT. In practice, these can be fixed to some constants for all iterations; see **A.6** for details. Finally at the end of this stage, we have estimate $\widehat{\mathbf{x}}^{(t)} := \mathbf{x}^{(R)}$ of $\mathbf{x}^*$.

**Learn Stage:** Using this estimate of the coefficients, we update the dictionary at $t$-th iteration $\mathbf{A}^{(t)}$ by an approximate gradient descent step (6), using the empirical gradient estimate (5) and the learning rate $\eta_A = \Theta(m/k)$; see also **A.5**. Finally, we normalize the columns of the dictionary and continue to the next batch. The running time of each step $t$ of NOODL is therefore $\mathcal{O}(mnp \log(1/\delta_R))$. For a target tolerance of $\epsilon_T$ and $\delta_T$, such that $\|\mathbf{A}_i^{(T)} - \mathbf{A}_i^*\| \leq \epsilon_T, \forall i \in [m]$ and $|\widehat{\mathbf{x}}_i^{(T)} - \mathbf{x}_i^*| \leq \delta_T$ we choose $T = \max(\Omega(\log(1/\epsilon_T)), \Omega(\log(\sqrt{k}/\delta_T)))$.

NOODL uses an initial HT step and an approximate gradient descent-based strategy as in Arora et al. (2015). Following which, our IHT-based coefficient update step yields an estimate of the coefficients at each iteration of the online algorithm. Coupled with the guaranteed progress made on the dictionary, this also removes the bias in dictionary estimation. Further, the simultaneous recovery of both factors also avoids an often expensive post-processing step for recovery of the coefficients.

## 3 MAIN RESULT

We start by introducing a few important definitions. First, as discussed in the previous section we require that the initial estimate $\mathbf{A}^{(0)}$ of the dictionary is $(\epsilon_0, 2)$-close to $\mathbf{A}^*$. In fact, we require this closeness property to hold at each subsequent iteration $t$, which is a key ingredient in our analysis. This initialization achieves two goals. First, the $\|\sigma(i)\mathbf{A}_{\pi(i)} - \mathbf{A}_i^*\| \leq \epsilon_0$ condition ensures that the signed-support of the coefficients are recovered correctly (with high probability) by the hard thresholding-based coefficient initialization step, where signed-support is defined as follows.

**Definition 2.** *The signed-support of a vector* $\mathbf{x}$ *is defined as* $\text{sign}(\mathbf{x}) \cdot \text{supp}(\mathbf{x})$.

Next, the $\|\mathbf{A} - \mathbf{A}^*\| \leq 2\|\mathbf{A}^*\|$ condition keeps the dictionary estimates close to $\mathbf{A}^*$ and is used in our analysis to ensure that the gradient direction (5) makes progress. Further, in our analysis, we ensure $\epsilon_t$ (defined as $\|\mathbf{A}_i^{(t)} - \mathbf{A}_i^*\| \leq \epsilon_t$) contracts at every iteration, and assume $\epsilon_0, \epsilon_t = \mathcal{O}^*(1/\log(n))$. Also, we assume that the dictionary $\mathbf{A}$ is fixed (deterministic) and $\mu$-incoherent, defined as follows.

**Definition 3.** *A matrix* $\mathbf{A} \in \mathbb{R}^{n \times m}$ *with unit-norm columns is* $\mu$*-incoherent if for all* $i \neq j$ *the inner-product between the columns of the matrix follow* $|\langle \mathbf{A}_i, \mathbf{A}_j \rangle| \leq \mu/\sqrt{n}$.

The incoherence parameter measures the degree of closeness of the dictionary elements. Smaller values (i.e., close to 0) of $\mu$ are preferred, since they indicate that the dictionary elements do not resemble each other. This helps us to effectively tell dictionary elements apart (Donoho and Huo, 2001; Candès and Romberg, 2007). We assume that $\mu = \mathcal{O}(\log(n))$ (Donoho and Huo, 2001). Next, we assume that the coefficients are drawn from a distribution class $\mathcal{D}$ defined as follows.

**Definition 4** (Distribution class $\mathcal{D}$)**.** *The coefficient vector* $\mathbf{x}^*$ *belongs to an unknown distribution* $\mathcal{D}$*, where the support* $S = \text{supp}(\mathbf{x}^*)$ *is at most of size* $k$*,* $\mathbf{Pr}[i \in S] = \Theta(k/m)$ *and* $\mathbf{Pr}[i, j \in S] = \Theta(k^2/m^2)$*. Moreover, the distribution is normalized such that* $\mathbf{E}[\mathbf{x}_i^* | i \in S] = 0$ *and* $\mathbf{E}[\mathbf{x}_i^{*^2} | i \in S] = 1$*, and when* $i \in S$*,* $|\mathbf{x}_i^*| \geq C$ *for some constant* $C \leq 1$*. In addition, the non-zero entries are sub-Gaussian and pairwise independent conditioned on the support.*

The randomness of the coefficient is necessary for our finite sample analysis of the convergence. Here, there are two sources of randomness. The first is the randomness of the support, where the non-zero elements are assumed to pair-wise independent. The second is the value an element in the support takes, which is assumed to be zero mean with variance one, and bounded in magnitude. Similar conditions are also required for support recovery of sparse coefficients, even when the dictionary is known (Wainwright, 2009; Yuan et al., 2016). Note that, although we only consider the case $|\mathbf{x}_i^*| \geq C$ for ease of discussion, analogous results may hold more generally for $\mathbf{x}_i^*$s drawn from a distribution with sufficiently (exponentially) small probability of taking values in $[-C, C]$.

Recall that, given the coefficients, we recover the dictionary by making progress on the least squares objective (P1) (ignoring the term penalizing $S(\cdot)$). Note that, our algorithm is based on finding an appropriate direction to ensure descent based on the geometry of the objective. To this end, we adopt a gradient descent-based strategy for dictionary update. However, since the coefficients are not exactly known, this results in an approximate gradient descent-based approach, where the empirical gradient estimate is formed as (5). In our analysis, we establish the conditions under which both the empirical gradient vector (corresponding to each dictionary element) and the gradient matrix concentrate around their means. To ensure progress at each iterate $t$, we show that the expected gradient vector is $(\Omega(k/m), \Omega(m/k), 0)$-correlated with the descent direction, defined as follows.

**Definition 5.** *A vector* $\mathbf{g}^{(t)}$ *is* $(\rho_-, \rho_+, \zeta_t)$*-correlated with a vector* $\mathbf{z}^*$ *if*

$$\langle \mathbf{g}^{(t)}, \mathbf{z}^{(t)} - \mathbf{z}^* \rangle \geq \rho_- \|\mathbf{z}^{(t)} - \mathbf{z}^*\|^2 + \rho_+ \|\mathbf{g}^{(t)}\|^2 - \zeta_t.$$

This can be viewed as a local descent condition which leads to the true dictionary columns; see also Candès et al. (2015), Chen and Wainwright (2015) and Arora et al. (2015). In convex optimization literature, this condition is implied by the $2\rho_-$-strong convexity, and $1/2\rho_+$-smoothness of the objective. We show that for NOODL, $\zeta_t = 0$, which facilitates linear convergence to $\mathbf{A}^*$ without incurring any bias. Overall our specific model assumptions for the analysis can be formalized as:

**A.1** $\mathbf{A}^*$ is $\mu$-incoherent (Def. 3), where $\mu = \mathcal{O}(\log(n))$, $\|\mathbf{A}^*\| = \mathcal{O}(\sqrt{m/n})$ and $m = \mathcal{O}(n)$;

**A.2** The coefficients are drawn from the distribution class $\mathcal{D}$, as per Def. 4;

**A.3** The sparsity $k$ satisfies $k = \mathcal{O}^*(\sqrt{n}/\mu \, \log(n))$;

**A.4** $\mathbf{A}^{(0)}$ is $(\epsilon_0, 2)$-close to $\mathbf{A}^*$ as per Def. 1, and $\epsilon_0 = \mathcal{O}^*(1/\log(n))$;

**A.5** The step-size for dictionary update satisfies $\eta_A = \Theta(m/k)$;

**A.6** The step-size and threshold for coefficient estimation satisfies $\eta_x^{(r)} < c_1(\epsilon_t, \mu, n, k) = \widetilde{\Omega}(k/\sqrt{n}) < 1$ and $\tau^{(r)} = c_2(\epsilon_t, \mu, k, n) = \widetilde{\Omega}(k^2/n)$ for small constants $c_1$ and $c_2$.

We are now ready to state our main result. A summary of the notation followed by a details of the analysis is provided in Appendix A and Appendix B, respectively.

**Theorem 1** (Main Result). *Suppose that assumptions **A.1**-**A.6** hold, and Algorithm 1 is provided with $p = \widetilde{\Omega}(mk^2)$ new samples generated according to model (1) at each iteration $t$. Then, with probability at least $(1 - \delta_{alg}^{(t)})$ for some small constant $\delta_{alg}^{(t)}$, given $R = \Omega(\log(n))$, the coefficient estimate $\widehat{\mathbf{x}}_i^{(t)}$ at $t$-th iteration has the correct signed-support and satisfies*

$$(\widehat{\mathbf{x}}_i^{(t)} - \mathbf{x}_i^*)^2 = \mathcal{O}(k(1-\omega)^{t/2}\|\mathbf{A}_i^{(0)} - \mathbf{A}_i^*\|), \text{ for all } i \in \text{supp}(\mathbf{x}^*).$$

*Furthermore, for some $0 < \omega < 1/2$, the estimate $\mathbf{A}^{(t)}$ at $(t)$-th iteration satisfies*

$$\|\mathbf{A}_i^{(t)} - \mathbf{A}_i^*\|^2 \leq (1-\omega)^t\|\mathbf{A}_i^{(0)} - \mathbf{A}_i^*\|^2, \text{ for all } t = 1, 2, \ldots..$$

Our main result establishes that when the model satisfies **A.1**$\sim$**A.3**, the errors corresponding to the dictionary and coefficients geometrically decrease to the true model parameters, given appropriate dictionary initialization and learning parameters (step sizes and threshold); see **A.4**$\sim$**A.6**. In other words, to attain a target tolerance of $\epsilon_T$ and $\delta_T$, where $\|\mathbf{A}_i^{(T)} - \mathbf{A}_i^*\| \leq \epsilon_T$, $|\widehat{\mathbf{x}}_i^{(T)} - \mathbf{x}_i^*| \leq \delta_T$, we require $T = \max(\Omega(\log(1/\epsilon_T)), \Omega(\log(\sqrt{k}/\delta_T)))$ outer iterations and $R = \Omega(\log(1/\delta_R))$ IHT steps per outer iteration. Here, $\delta_R \geq (1 - \eta_x)^R$ is the target decay tolerance for the IHT steps. An appropriate number of IHT steps, $R$, remove the dependence of final coefficient error (per outer iteration) on the initial $\mathbf{x}^{(0)}$. In Arora et al. (2015), this dependence in fact results in an irreducible error, which is the source of bias in dictionary estimation. As a result, since (for NOODL) the error in the coefficients only depends on the error in the dictionary, it can be made arbitrarily small, at a geometric rate, by the choice of $\epsilon_T$, $\delta_T$, and $\delta_R$. Also, note that, NOODL can tolerate i.i.d. noise, as long as the noise variance is controlled to enable the concentration results to hold; we consider the noiseless case here for ease of discussion, which is already highly involved.

Intuitively, Theorem 1 highlights the symbiotic relationship between the two factors. It shows that, to make progress on one, it is imperative to make progress on the other. The primary condition that allows us to make progress on both factors is the signed-support recovery (Def. 2). However, the introduction of IHT step adds complexity in the analysis of both the dictionary and coefficients. To analyze the coefficients, in addition to deriving conditions on the parameters to preserve the correct signed-support, we analyze the recursive IHT update step, and decompose the noise term into a component that depends on the error in the dictionary, and the other that depends on the initial coefficient estimate. For the dictionary update, we analyze the interactions between elements of the coefficient vector (introduces by the IHT-based update step) and show that the gradient vector for the dictionary update is $(\Omega(k/m), \Omega(m/k), 0)$-correlated with the descent direction. In the end, this leads to exact recovery of the coefficients and removal of bias in the dictionary estimation. Note that our analysis pipeline is standard for the convergence analysis for iterative algorithms. However, the introduction of the IHT-based strategy for coefficient update makes the analysis highly involved as compared to existing results, e.g., the simple HT-based coefficient estimate in Arora et al. (2015).

NOODL has an overall running time of $\mathcal{O}(mnp\log(1/\delta_R)\max(\log(1/\epsilon_T), \log(\sqrt{k}/\delta_T))$ to achieve target tolerances $\epsilon_T$ and $\delta_T$, with a total sample complexity of $p \cdot T = \widetilde{\Omega}(mk^2)$. Thus to remove bias, the IHT-based coefficient update introduces a factor of $\log(1/\delta_R)$ in the computational complexity as compared to Arora et al. (2015) (has a total sample complexity of $p \cdot T = \widetilde{\Omega}(mk)$), and also does not have the exponential running time and sample complexity as Barak et al. (2015); see Table 1.

## 4   NEURAL IMPLEMENTATION OF NOODL

The neural plausibility of our algorithm implies that it can be implemented as a neural network. This is because, NOODL employs simple linear and non-linear operations (such as inner-product and hard-thresholding) and the coefficient updates are separable across data samples, as shown in (4) of Algorithm 1. To this end, we present a neural implementation of our algorithm in Fig. 1, which showcases the applicability of NOODL in large-scale distributed learning tasks, motivated from the implementations described in (Olshausen and Field, 1997) and (Arora et al., 2015).

The neural architecture shown in Fig. 1(a) has three layers – input layer, weighted residual evaluation layer, and the output layer. The input to the network is a data and step-size pair $(\mathbf{y}_{(j)}, \eta_x)$ to each input node. Given an input, the second layer evaluates the weighted residuals as shown in Fig. 1. Finally, the output layer neurons evaluate the IHT iterates $\mathbf{x}_{(j)}^{(r+1)}$ (4). We illustrate the operation of this architecture using the timing diagram in Fig. 1(b). The main stages of operation are as follows.

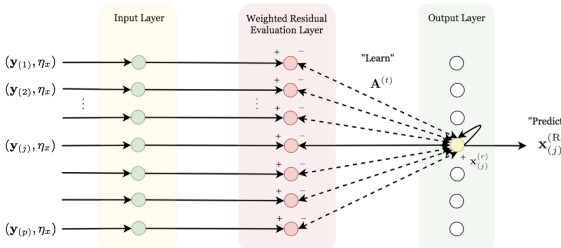

**Figure 1:** A neural implementation of NOODL. Panel (a) shows the neural architecture, which consists of three layers: an input layer, a weighted residual evaluation layer (evaluates $\eta_x\big(\mathbf{y}_{(j)} - \mathbf{A}^{(t)}\mathbf{x}^{(r)}_{(j)}\big)$), and an output layer. Panel (b) shows the operation of the neural architecture in panel (a). The update of $\mathbf{x}^{(r+1)}_{(j)}$ is given by (4).

(a) Neural implementation of NOODL

| $\ell = 0$ | $\ell = 1$ | $\ell = 2$ | $\ell = 3$ | $\ell = 4$ | $\ell = 5$ | $\dots$ | $\ell = 2R+1$ | **Hebbian Learning**: |
|---|---|---|---|---|---|---|---|---|
| Output: $\mathbf{x} \leftarrow \mathbf{0}$ | $\mathbf{0}$ | $\mathbf{x}^{(0)}_{(j)} = \mathcal{T}_\tau(\mathbf{A}^{(t)\top}\mathbf{y}_{(j)})$ | $\mathbf{x}^{(0)}_{(j)}$ | $\mathbf{x}^{(1)}_{(j)}$ | $\mathbf{x}^{(1)}_{(j)}$ | $\dots$ | $\mathbf{x}^{(R)}_{(j)}$ | Residual sharing and dictionary update. |
| Residual: $\mathbf{0}$ | $\mathbf{y}_{(j)}$ | $\mathbf{y}_{(j)}$ | $\eta_x(\mathbf{y}_{(j)} - \mathbf{A}^{(t)}\mathbf{x}^{(0)}_{(j)})$ | $\eta_x(\mathbf{y}_{(j)} - \mathbf{A}^{(t)}\mathbf{x}^{(0)}_{(j)})$ | $\eta_x(\mathbf{y}_{(j)} - \mathbf{A}^{(t)}\mathbf{x}^{(1)}_{(j)})$ | $\dots$ | $\eta_x(\mathbf{y}_{(j)} - \mathbf{A}^{(t)}\mathbf{x}^{(R-1)}_{(j)})$ | |
| Input: $(\mathbf{y}_{(j)}, 1)$ | . | $(\mathbf{y}_{(j)}, \eta_x)$ | . | . | . | $\dots$ | $(\mathbf{y}_{(j)}, 1)$ | |

(b) The timing sequence of the neural implementation.

**Initial Hard Thresholding Phase**: The coefficients initialized to zero, and an input $(\mathbf{y}_{(j)}, 1)$ is provided to the input layer at a time instant $\ell = 0$, which communicates these to the second layer. Therefore, the residual at the output of the weighted residual evaluation layer evaluates to $\mathbf{y}_{(j)}$ at $\ell = 1$. Next, at $\ell = 2$, this residual is communicated to the output layer, which results in evaluation of the initialization $\mathbf{x}^{(0)}_{(j)}$ as per (3). This iterate is communicated to the second layer for the next residual evaluation. Also, at this time, the input layer is injected with $(\mathbf{y}_{(j)}, \eta_x)$ to set the step size parameter $\eta_x$ for the IHT phase, as shown in Fig. 1(b).

**Iterative Hard Thresholding (IHT) Phase**: Beginning $\ell = 3$, the timing sequence enters the IHT phase. Here, the output layer neurons communicate the iterates $\mathbf{x}^{(r+1)}_{(j)}$ to the second layer for evaluation of subsequent iterates as shown in Fig. 1(b). The process then continues till the time instance $\ell = 2R + 1$, for $R = \Omega(\log(1/\delta_R))$ to generate the final coefficient estimate $\widehat{\mathbf{x}}^{(t)}_{(j)} := \mathbf{x}^{(R)}_{(j)}$ for the current batch of data. At this time, the input layer is again injected with $(\mathbf{y}_{(j)}, 1)$ to prepare the network for residual sharing and gradient evaluation for dictionary update.

**Dictionary Update Phase:** The procedure now enters the dictionary update phase, denoted as "Hebbian Learning" in the timing sequence. In this phase, each output layer neuron communicates the final coefficient estimate $\widehat{\mathbf{x}}^{(t)}_{(j)} = \mathbf{x}^{(R)}_{(j)}$ to the second layer, which evaluates the residual for one last time (with $\eta_x = 1$), and shares it across all second layer neurons ("Hebbian learning"). This allows each second layer neuron to evaluate the empirical gradient estimate (5), which is used to update the current dictionary estimate (stored as weights) via an approximate gradient descent step. This completes one outer iteration of Algorithm 1, and the process continues for $T$ iterations to achieve target tolerances $\epsilon_T$ and $\delta_T$, with each step receiving a new mini-batch of data.

## 5 EXPERIMENTS

We now analyze the convergence properties and sample complexity of NOODL via experimental evaluations [2]. The experimental data generation set-up, additional results, including analysis of computational time, are shown in Appendix E.

### 5.1 CONVERGENCE ANALYSIS

We compare the performance of our algorithm NOODL with the current state-of-the-art alternating optimization-based online algorithms presented in Arora et al. (2015), and the popular algorithm presented in Mairal et al. (2009) (denoted as `Mairal '09`). First of these, `Arora15('`biased'')`, is a simple neurally plausible method which incurs a bias and has a sample complexity of $\Omega(mk)$. The other, referred to as `Arora15('`unbiased'')`, incurs no bias as per Arora et al. (2015), but the sample complexity results were not established.

**Discussion:** Fig. 2 panels (a-i), (b-i), (c-i), and (d-i) show the performance of the aforementioned methods for $k = 10$, 20, 50, and 100, respectively. Here, for all experiments we set $\eta_x = 0.2$ and $\tau = 0.1$. We terminate NOODL when the error in dictionary is less than $10^{-10}$. Also, for coefficient update, we terminate when change in the iterates is below $10^{-12}$. For $k = 10$, 20 and $k = 50$,

---

[2]The associated code is made available at `https://github.com/srambhatla/NOODL`.

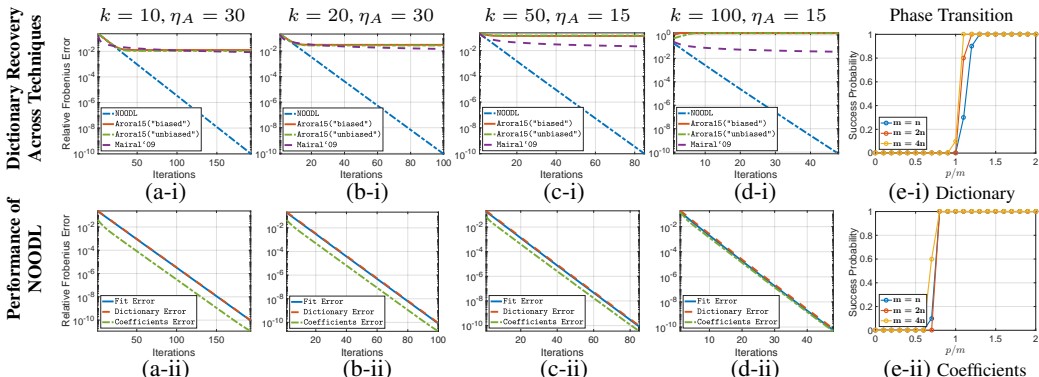

**Figure 2:** Comparative analysis of convergence properties. Panels (a-i), (b-i), (c-i), and (d-i) show the convergence of NOODL, `Arora15(''biased'')`, `Arora15(''unbiased'')` and `Mairal '09`, for different sparsity levels for $n = 1000$, $m = 1500$ and $p = 5000$. Since NOODL also recovers the coefficients, we show the corresponding recovery of the dictionary, coefficients, and overall fit in panels (a-ii), (b-ii), (c-ii), and (d-ii), respectively. Further, panels (e-i) and (e-ii) show the phase transition in samples $p$ (per iteration) with the size of the dictionary $m$ averaged across 10 Monte Carlo simulations for the two factors. Here, $n = 100$, $k = 3$, $\eta_x = 0.2$, $\tau = 0.1$, $\epsilon_0 = 2/\log(n)$, $\eta_A$ is chosen as per **A.5**. A trial is considered successful if the relative Frobenius error incurred by $\widehat{\mathbf{A}}$ and $\widehat{\mathbf{X}}$ is below $5 \times 10^{-7}$ after 50 iterations.

we note that `Arora15(''biased'')` and `Arora15(''unbiased'')` incur significant bias, while NOODL converges to $\mathbf{A}^*$ *linearly*. NOODL also converges for significantly higher choices of sparsity $k$, i.e., for $k = 100$ as shown in panel (d), beyond $k = \mathcal{O}(\sqrt{n})$, indicating a potential for improving this bound. Further, we observe that `Mairal '09` exhibits significantly slow convergence as compared to NOODL. Also, in panels (a-ii), (b-ii), (c-ii) and (d-ii) we show the corresponding performance of NOODL in terms of the error in the overall fit ($\|\mathbf{Y} - \mathbf{A}\mathbf{X}\|_F/\|\mathbf{Y}\|_F$), and the error in the coefficients and the dictionary, in terms of relative Frobenius error metric discussed above. We observe that the error in dictionary and coefficients drops linearly as indicated by our main result.

## 5.2 PHASE TRANSITIONS

Fig. 2 panels (e-i) and (e-ii), shows the phase transition in number of samples with respect to the size of the dictionary $m$. We observe a sharp phase transition at $\frac{p}{m} = 1$ for the dictionary, and at $\frac{p}{m} = 0.75$ for the coefficients. This phenomenon is similar to that observed by Agarwal et al. (2014) (however, theoretically they required $p = \mathcal{O}(m^2)$). Here, we confirm number of samples required by NOODL are linearly dependent on the dictionary elements $m$.

## 6 FUTURE WORK

We consider the online DL setting in this work. We note that, empirically NOODL works for the batch setting also. However, analysis for this case will require more sophisticated concentration results, which can address the resulting dependence between iterations of the algorithm. In addition, our experiments indicate that NOODL works beyond the sparsity ranges prescribed by our theoretical results. Arguably, the bounds on sparsity can potentially be improved by moving away from the incoherence-based analysis. We also note that in our experiments, NOODL converges even when initialized outside the prescribed initialization region, albeit it achieves the linear rate once it satisfies the closeness condition **A.4**. These potential directions may significantly impact the analysis and development of provable algorithms for other factorization problems as well. We leave these research directions, and a precise analysis under the noisy setting, for future explorations.

## 7 CONCLUSIONS

We present NOODL, to the best of our knowledge, the first neurally plausible provable online algorithm for exact recovery of both factors of the dictionary learning (DL) model. NOODL alternates between: (a) an iterative hard thresholding (IHT)-based step for coefficient recovery, and (b) a gradient descent-based update for the dictionary, resulting in a simple and scalable algorithm, suitable for large-scale distributed implementations. We show that once initialized appropriately, the sequence of estimates produced by NOODL converge *linearly* to the true dictionary and coefficients without incurring any bias in the estimation. Complementary to our theoretical and numerical results, we also design an implementation of NOODL in a neural architecture for use in practical applications. In essence, the analysis of this inherently non-convex problem impacts other matrix and tensor factorization tasks arising in signal processing, collaborative filtering, and machine learning.

ACKNOWLEDGMENT

The authors would like to graciously acknowledge support from DARPA Young Faculty Award, Grant No. N66001-14-1-4047.

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

## A  SUMMARY OF NOTATION

We summarizes the definitions of some frequently used symbols in our analysis in Table 2. In addition, we use $\mathbf{D_{(v)}}$ as a diagonal matrix with elements of a vector $\mathbf{v}$ on the diagonal. Given a matrix $\mathbf{M}$, we use $\mathbf{M}_{-i}$ to denote a resulting matrix without $i$-th column. Also note that, since we show that $\|\mathbf{A}_i^{(t)} - \mathbf{A}_i^*\| \leq \epsilon_t$ contracts in every step, therefore we fix $\epsilon_t, \epsilon_0 = \mathcal{O}^*(1/\log(n))$ in our analysis.

**Table 2:** Frequently used symbols

**Dictionary Related**

| Symbol | Definition | |
|---|---|---|
| $\mathbf{A}_i^{(t)}$ | $i$-th column of the dictionary estimate at the $t$-th iterate. | |
| $\epsilon_t$ | $\|\mathbf{A}_i^{(t)} - \mathbf{A}_i^*\| \leq \epsilon_t = \mathcal{O}^*(\frac{1}{\log(n)})$ | Upper-bound on column-wise error at the $t$-th iterate. |
| $\mu_t$ | $\frac{\mu_t}{\sqrt{n}} = \frac{\mu}{\sqrt{n}} + 2\epsilon_t$ | Incoherence between the columns of $\mathbf{A}^{(t)}$; See Claim 1. |
| $\lambda_j^{(t)}$ | $\lambda_j^{(t)} := |\langle \mathbf{A}_j^{(t)} - \mathbf{A}_j^*, \mathbf{A}_j^* \rangle| \leq \frac{\epsilon_t^2}{2}$ | Inner-product between the error and the dictionary element. |
| $\Lambda_S^{(t)}(i,j)$ | $\Lambda_S^{(t)}(i,j) = \begin{cases} \lambda_j^{(t)}, & \text{for } j = i, i \in S \\ 0, & \text{otherwise.} \end{cases}$ | A diagonal matrix of size $|S| \times |S|$ with $\lambda_j^{(t)}$ on the diagonal for $j \in S$. |

**Coefficient Related**

| Symbol | Definition | |
|---|---|---|
| $\mathbf{x}_i^{(r)}$ | $i$-th element the coefficient estimate at the $r$-th IHT iterate. | |
| $C$ | $|\mathbf{x}_i^*| \geq C$ for $i \in \text{supp}(\mathbf{x}^*)$ and $C \leq 1$ | Lower-bound on $\mathbf{x}_i^*$s. |
| $S$ | $S := \text{supp}(\mathbf{x}^*)$ where $|S| \leq k$ | Support of $\mathbf{x}^*$ |
| $\delta_R$ | $\delta_R := (1 - \eta_x + \eta_x \frac{\mu_t}{\sqrt{n}})^R \geq (1 - \eta_x)^R$ | Decay parameter for coefficients. |
| $\delta_T$ | $|\widehat{\mathbf{x}}_i^{(T)} - \mathbf{x}_i^*| \leq \delta_T \forall i \in \text{supp}(\mathbf{x}^*)$ | Target coefficient element error tolerance. |
| $C_i^{(\ell)}$ | $C_i^{(\ell)} := |\mathbf{x}_i^* - \mathbf{x}_i^{(\ell)}|$ for $i \in \text{supp}(\mathbf{x}^*)$ | Error in non-zero elements of the coefficient vector. |

**Probabilities**

| Symbol | Definition | Symbol | Definition |
|---|---|---|---|
| $q_i$ | $q_i = \mathbf{Pr}[i \in S] = \Theta(\frac{k}{m})$ | $q_{i,j}$ | $q_{i,j} = \mathbf{Pr}[i,j \in S] = \Theta(\frac{k^2}{m^2})$ |
| $p_i$ | $p_i = \mathbf{E}[\mathbf{x}_i^* \text{sign}(\mathbf{x}_i^*) | \mathbf{x}_i^* \neq 0]$ | $\delta_T^{(t)}$ | $\delta_T^{(t)} = 2m \exp(-C^2/\mathcal{O}^*(\epsilon_t^2))$ |
| $\delta_\beta^{(t)}$ | $\delta_\beta^{(t)} = 2k \exp(-1/\mathcal{O}(\epsilon_t))$ | $\delta_{\text{HW}}^{(t)}$ | $\delta_{\text{HW}}^{(t)} = \exp(-1/\mathcal{O}(\epsilon_t))$ |
| $\delta_{\mathbf{g}_i}^{(t)}$ | $\delta_{\mathbf{g}_i}^{(t)} = \exp(-\Omega(k))$ | $\delta_{\mathbf{g}}^{(t)}$ | $\delta_{\mathbf{g}}^{(t)} = (n+m)\exp(-\Omega(m\sqrt{\log(n)}))$ |

**Other terms**

| Symbol | Definition |
|---|---|
| $\xi_j^{(r+1)}$ | $\xi_j^{(r+1)} := \sum_{i \neq j}(\langle \mathbf{A}_j^{(t)} - \mathbf{A}_j^*, \mathbf{A}_i^* \rangle + \langle \mathbf{A}_j^*, \mathbf{A}_i^* \rangle)\mathbf{x}_i^* - \sum_{i \neq j} \langle \mathbf{A}_j^{(t)}, \mathbf{A}_i^{(t)} \rangle \mathbf{x}_i^{(r)}$ |
| $\beta_j^{(t)}$ | $\beta_j^{(t)} := \sum_{i \neq j}(\langle \mathbf{A}_j^*, \mathbf{A}_i^* - \mathbf{A}_i^{(t)} \rangle + \langle \mathbf{A}_j^* - \mathbf{A}_j^{(t)}, \mathbf{A}_i^{(t)} \rangle + \langle \mathbf{A}_j^{(t)} - \mathbf{A}_j^*, \mathbf{A}_i^* \rangle)\mathbf{x}_i^*$ |
| $t_\beta$ | $t_\beta = \mathcal{O}(\sqrt{k}\epsilon_t)$ is an upper-bound on $\beta_j^{(t)}$ with probability at least $(1 - \delta_\beta^{(t)})$ |
| $\widetilde{\xi}_j^{(r+1)}$ | $\widetilde{\xi}_j^{(r+1)} := \beta_j^{(t)} + \sum_{i \neq j} |\langle \mathbf{A}_j^{(t)}, \mathbf{A}_i^{(t)} \rangle| \, |\mathbf{x}_i^* - \mathbf{x}_i^{(r)}|$ |
| $\Delta_j^{(t)}$ | $\Delta_j^{(t)} := \mathbf{E}[\mathbf{A}_S^{(t)} \vartheta_S^{(R)} \text{sign}(\mathbf{x}_j^*)]$ |
| $\vartheta_i^{(R)}$ | $\vartheta_i^{(R)} := \sum_{r=1}^R \eta_x \xi_i^{(r)} (1 - \eta_x)^{R-r} + \gamma_i^{(R)}$ |
| $\gamma_i^{(R)}$ | $\gamma_i^{(R)} := (1 - \eta_x)^R (\mathbf{x}_i^{(0)} - \mathbf{x}_i^*(1 - \lambda_i^{(t)}))$ |
| $\gamma$ | $\gamma := \mathbf{E}[(\mathbf{A}^{(t)}\widehat{\mathbf{x}} - \mathbf{y})\text{sign}(\mathbf{x}_j^*)\mathbb{1}_{\overline{\mathcal{F}}_{\mathbf{x}^*}}]$; See † below. |
| $\widehat{\mathbf{x}}_i$ | $\widehat{\mathbf{x}}_i := \mathbf{x}_i^{(R)} = \mathbf{x}_i^*(1 - \lambda_i^{(t)}) + \vartheta_i^{(R)}$ |

† $\mathbb{1}_{\mathcal{F}_{\mathbf{x}^*}}$ is the indicator function corresponding to the event that $\text{sign}(\mathbf{x}^*) = \text{sign}(\widehat{\mathbf{x}})$, denoted by $\mathcal{F}_{\mathbf{x}^*}$, and similarly for the complement $\overline{\mathcal{F}}_{\mathbf{x}^*}$

## B    PROOF OF THEOREM 1

We now prove our main result. The detailed proofs of intermediate lemmas and claims are organized in Appendix C and Appendix D, respectively. Furthermore, the standard concentration results are stated in Appendix F for completeness. Also, see Table 3 for a map of dependence between the results.

### OVERVIEW

Given an $(\epsilon_0, 2)$-close estimate of the dictionary, the main property that allows us to make progress on the dictionary is the recovery of the correct sign and support of the coefficients. Therefore, we first show that the initial coefficient estimate (3) recovers the correct signed-support in Step I.A. Now, the IHT-based coefficient update step also needs to preserve the correct signed-support. This is to ensure that the approximate gradient descent-based update for the dictionary makes progress. Therefore, in Step I.B, we derive the conditions under which the signed-support recovery condition is preserved by the IHT update.

To get a handle on the coefficients, in Step II.A, we derive an upper-bound on the error incurred by each non-zero element of the estimated coefficient vector, i.e., $|\widehat{\mathbf{x}}_i - \mathbf{x}_i^*|$ for $i \in S$ for a general coefficient vector $\mathbf{x}^*$, and show that this error only depends on $\epsilon_t$ (the column-wise error in the dictionary) given enough IHT iterations $R$ as per the chosen decay parameter $\delta_R$. In addition, for analysis of the dictionary update, we develop an expression for the estimated coefficient vector in Step II.B.

We then use the coefficient estimate to show that the gradient vector satisfies the local descent condition (Def. 5). This ensures that the gradient makes progress after taking the gradient descent-based step (6). To begin, we first develop an expression for the expected gradient vector (corresponding to each dictionary element) in Step III.A. Here, we use the closeness property Def 1 of the dictionary estimate. Further, since we use an empirical estimate, we show that the empirical gradient vector concentrates around its mean in Step III.B. Now using Lemma 15, we have that descent along this direction makes progress.

Next in Step IV.A and Step IV.B, we show that the updated dictionary estimate maintains the closeness property Def 1. This sets the stage for the next dictionary update iteration. As a result, our main result establishes the conditions under which any $t$-th iteration succeeds.

Our main result is as follows.

**Theorem 1** (Main Result) *Suppose that assumptions **A.1**-**A.6** hold, and Algorithm 1 is provided with $p = \widetilde{\Omega}(mk^2)$ new samples generated according to model (1) at each iteration $t$. Then, with probability at least $(1 - \delta_{alg}^{(t)})$, given $R = \Omega(\log(n))$, the coefficient estimate $\widehat{\mathbf{x}}_i^{(t)}$ at $t$-th iteration has the correct signed-support and satisfies*

$$(\widehat{\mathbf{x}}_i^{(t)} - \mathbf{x}_i^*)^2 = \mathcal{O}(k(1 - \omega)^{t/2} \|\mathbf{A}_i^{(0)} - \mathbf{A}_i^*\|), \text{ for all } i \in \mathrm{supp}(\mathbf{x}^*).$$

*Furthermore, for some $0 < \omega < 1/2$, the estimate $\mathbf{A}^{(t)}$ at $(t)$-th iteration satisfies*

$$\|\mathbf{A}_i^{(t)} - \mathbf{A}_i^*\|^2 \leq (1 - \omega)^t \|\mathbf{A}_i^{(0)} - \mathbf{A}_i^*\|^2, \text{ for all } t = 1, 2, \ldots..$$

*Here, $\delta_{alg}^{(t)}$ is some small constant, where $\delta_{alg}^{(t)} = \delta_{\mathcal{T}}^{(t)} + \delta_{\beta}^{(t)} + \delta_{\mathrm{HW}} + \delta_{\mathbf{g}_i}^{(t)} + \delta_{\mathbf{g}}^{(t)}$, $\delta_{\mathcal{T}}^{(t)} = 2m \exp(-C^2/\mathcal{O}^*(\epsilon_t^2))$, $\delta_{\beta}^{(t)} = 2k \exp(-1/\mathcal{O}(\epsilon_t))$, $\delta_{\mathrm{HW}}^{(t)} = \exp(-1/\mathcal{O}(\epsilon_t))$, $\delta_{\mathbf{g}_i}^{(t)} = \exp(-\Omega(k))$, $\delta_{\mathbf{g}}^{(t)} = (n + m) \exp(-\Omega(m\sqrt{\log(n)}))$, and $\|\mathbf{A}_i^{(t)} - \mathbf{A}_i^*\| \leq \epsilon_t$.*

### STEP I: COEFFICIENTS HAVE THE CORRECT SIGNED-SUPPORT

As a first step, we ensure that our coefficient estimate has the correct signed-support (Def. 2). To this end, we first show that the initialization has the correct signed-support, and then show that the iterative hard-thresholding (IHT)-based update step preserves the correct signed-support for a suitable choice of parameters.

- **Step I.A: Showing that the initial coefficient estimate has the correct signed-support–** Given an $(\epsilon_0, 2)$-close estimate $\mathbf{A}^{(0)}$ of $\mathbf{A}^*$, we first show that for a general sample $\mathbf{y}$ the initialization step (3) recovers the correct signed-support with probability at least $(1 - \delta_{\mathcal{T}}^{(t)})$, where $\delta_{\mathcal{T}}^{(t)} = 2m \exp(-\frac{C^2}{\mathcal{O}^*(\epsilon_t^2)})$. This is encapsulated by the following lemma.

  **Lemma 1** (**Signed-support recovery by coefficient initialization step**)**.** Suppose $\mathbf{A}^{(t)}$ is $\epsilon_t$-close to $\mathbf{A}^*$. Then, if $\mu = \mathcal{O}(\log(n))$, $k = \mathcal{O}^*(\sqrt{n}/\mu \log(n))$, and $\epsilon_t = \mathcal{O}^*(1/\sqrt{\log(m)})$, with probability at least $(1 - \delta_{\mathcal{T}}^{(t)})$ for each random sample $\mathbf{y} = \mathbf{A}^* \mathbf{x}^*$:

  $$\text{sign}(\mathcal{T}_{C/2}((\mathbf{A}^{(t)})^\top \mathbf{y}) = \text{sign}(\mathbf{x}^*),$$

  where $\delta_{\mathcal{T}}^{(t)} = 2m \exp(-\frac{C^2}{\mathcal{O}^*(\epsilon_t^2)})$.

  Note that this result only requires the dictionary to be column-wise close to the true dictionary, and works for less stringent conditions on the initial dictionary estimate, i.e., requires $\epsilon_t = \mathcal{O}^*(1/\sqrt{\log(m)})$ instead of $\epsilon_t = \mathcal{O}^*(1/\log(m))$; see also (Arora et al., 2015).

- **Step I.B: The iterative IHT-type updates preserve the correct signed support–** Next, we show that the IHT-type coefficient update step (4) preserves the correct signed-support for an appropriate choice of step-size parameter $\eta_x^{(r)}$ and threshold $\tau^{(r)}$. The choice of these parameters arises from the analysis of the IHT-based update step. Specifically, we show that at each iterate $r$, the step-size $\eta_x^{(r)}$ should be chosen to ensure that the component corresponding to the true coefficient value is greater than the "interference" introduced by other non-zero coefficient elements. Then, if the threshold is chosen to reject this "noise", each iteration of the IHT-based update step preserves the correct signed-support.

  **Lemma 2** (**IHT update step preserves the correct signed-support**)**.** Suppose $\mathbf{A}^{(t)}$ is $\epsilon_t$-close to $\mathbf{A}^*$, $\mu = \mathcal{O}(\log(n))$, $k = \mathcal{O}^*(\sqrt{n}/\mu \log(n))$, and $\epsilon_t = \mathcal{O}^*(1/\log(m))$ Then, with probability at least $(1 - \delta_{\beta}^{(t)} - \delta_{\mathcal{T}}^{(t)})$, each iterate of the IHT-based coefficient update step shown in (4) has the correct signed-support, if for a constant $c_1^{(r)}(\epsilon_t, \mu, k, n) = \widetilde{\Omega}(k^2/n)$, the step size is chosen as $\eta_x^{(r)} \leq c_1^{(r)}$ , and the threshold $\tau^{(r)}$ is chosen as

  $$\tau^{(r)} = \eta_x^{(r)}(t_\beta + \tfrac{\mu_t}{\sqrt{n}} \|\mathbf{x}^{(r-1)} - \mathbf{x}^*\|_1) := c_2^{(r)}(\epsilon_t, \mu, k, n) = \widetilde{\Omega}(k^2/n),$$

  for some constants $c_1$ and $c_2$. Here, $t_\beta = \mathcal{O}(\sqrt{k}\epsilon_t)$, $\delta_{\mathcal{T}}^{(t)} = 2m \exp(-\frac{C^2}{\mathcal{O}^*(\epsilon_t^2)})$ ,and $\delta_{\beta}^{(t)} = 2k \exp(-\frac{1}{\mathcal{O}(\epsilon_t)})$.

  Note that, although we have a dependence on the iterate $r$ in choice of $\eta_x^{(r)}$ and $\tau^{(r)}$, these can be set to some constants independent of $r$. In practice, this dependence allows for greater flexibility in the choice of these parameters.

STEP II: ANALYZING THE COEFFICIENT ESTIMATE

We now derive an upper-bound on the error incurred by each non-zero coefficient element. Further, we derive an expression for the coefficient estimate at the $t$-th round of the online algorithm $\widehat{\mathbf{x}}^{(t)} := \mathbf{x}^{(R)}$; we use $\widehat{\mathbf{x}}$ instead of $\widehat{\mathbf{x}}^{(t)}$ for simplicity.

- **Step II.A: Derive a bound on the error incurred by the coefficient estimate–** Since Lemma 2 ensures that $\widehat{\mathbf{x}}$ has the correct signed-support, we now focus on the error incurred by each coefficient element on the support by analyzing $\widehat{\mathbf{x}}$. To this end, we carefully analyze the effect of the recursive update (4), to decompose the error incurred by each element on the support into two components – one that depends on the initial coefficient estimate $\mathbf{x}^{(0)}$ and other that depends on the error in the dictionary.

  We show that the effect of the component that depends on the initial coefficient estimate diminishes by a factor of $(1 - \eta_x + \eta_x \frac{\mu_t}{\sqrt{n}})$ at each iteration $r$. Therefore, for a decay parameter $\delta_R$, we can choose the number of IHT iterations $R$, to make this component arbitrarily small. Therefore, the error in the coefficients only depends on the per column error in the dictionary, formalized by the following result.

**Lemma 3** (**Upper-bound on the error in coefficient estimation**). With probability at least $(1 - \delta_\beta^{(t)} - \delta_\mathcal{T}^{(t)})$ the error incurred by each element $i_1 \in \text{supp}(\mathbf{x}^*)$ of the coefficient estimate is upper-bounded as

$$|\widehat{\mathbf{x}}_{i_1} - \mathbf{x}_{i_1}^*| \leq \mathcal{O}(t_\beta) + \left((R+1)k\eta_x \tfrac{\mu_t}{\sqrt{n}} \max_i |\mathbf{x}_i^{(0)} - \mathbf{x}_i^*| + |\mathbf{x}_{i_1}^{(0)} - \mathbf{x}_{i_1}^*|\right)\delta_R, = \mathcal{O}(t_\beta)$$

where $t_\beta = \mathcal{O}(\sqrt{k}\epsilon_t)$, $\delta_R := (1 - \eta_x + \eta_x \tfrac{\mu_t}{\sqrt{n}})^R$, $\delta_\mathcal{T}^{(t)} = 2m \exp(-\tfrac{C^2}{\mathcal{O}^*(\epsilon_t^2)})$, $\delta_\beta^{(t)} = 2k \exp(-\tfrac{1}{\mathcal{O}(\epsilon_t)})$, and $\mu_t$ is the incoherence between the columns of $\mathbf{A}^{(t)}$; see Claim 1.

This result allows us to show that if the column-wise error in the dictionary decreases at each iteration $t$, then the corresponding estimates of the coefficients also improve.

- *Step II.B: Developing an expression for the coefficient estimate*– Next, we derive the expression for the coefficient estimate in the following lemma. This expression is used to analyze the dictionary update.

  **Lemma 4** (**Expression for the coefficient estimate at the end of $R$-th IHT iteration**). With probability at least $(1 - \delta_\mathcal{T}^{(t)} - \delta_\beta^{(t)})$ the $i_1$-th element of the coefficient estimate, for each $i_1 \in \text{supp}(\mathbf{x}^*)$, is given by

  $$\widehat{\mathbf{x}}_{i_1} := \mathbf{x}_{i_1}^{(R)} = \mathbf{x}_{i_1}^*(1 - \lambda_{i_1}^{(t)}) + \vartheta_{i_1}^{(R)}.$$

  Here, $\vartheta_{i_1}^{(R)}$ is $|\vartheta_{i_1}^{(R)}| = \mathcal{O}(t_\beta)$, where $t_\beta = \mathcal{O}(\sqrt{k}\epsilon_t)$. Further, $\lambda_{i_1}^{(t)} = |\langle \mathbf{A}_{i_1}^{(t)} - \mathbf{A}_{i_1}^*, \mathbf{A}_{i_1}^* \rangle| \leq \tfrac{\epsilon_t^2}{2}$, $\delta_\mathcal{T}^{(t)} = 2m \exp(-\tfrac{C^2}{\mathcal{O}^*(\epsilon_t^2)})$ and $\delta_\beta^{(t)} = 2k \exp(-\tfrac{1}{\mathcal{O}(\epsilon_t)})$.

  We again observe that the error in the coefficient estimate depends on the error in the dictionary via $\lambda_{i_1}^{(t)}$ and $\vartheta_{i_1}^{(R)}$.

STEP III: ANALYZING THE GRADIENT FOR DICTIONARY UPDATE

Given the coefficient estimate we now show that the choice of the gradient as shown in (5) makes progress at each step. To this end, we analyze the gradient vector corresponding to each dictionary element to see if it satisfies the local descent condition of Def. 5. Our analysis of the gradient is motivated from Arora et al. (2015). However, as opposed to the simple HT-based coefficient update step used by Arora et al. (2015), our IHT-based coefficient estimate adds to significant overhead in terms of analysis. Notwithstanding the complexity of the analysis, we show that this allows us to remove the bias in the gradient estimate.

To this end, we first develop an expression for each expected gradient vector, show that the empirical gradient estimate concentrates around its mean, and finally show that the empirical gradient vector is $(\Omega(k/m), \Omega(m/k), 0)$-correlated with the descent direction, i.e. has no bias.

- **Step III.A: Develop an expression for the expected gradient vector corresponding to each dictionary element**– The expression for the expected gradient vector $\mathbf{g}_j^{(t)}$ corresponding to $j$-th dictionary element is given by the following lemma.

  **Lemma 5** (**Expression for the expected gradient vector**). Suppose that $\mathbf{A}^{(t)}$ is $(\epsilon_t, 2)$-near to $\mathbf{A}^*$. Then, the dictionary update step in Algorithm 1 amounts to the following for the $j$-th dictionary element

  $$\mathbf{E}[\mathbf{A}_j^{(t+1)}] = \mathbf{A}_j^{(t)} + \eta_A \mathbf{g}_j^{(t)},$$

  where $\mathbf{g}_j^{(t)}$ is given by

  $$\mathbf{g}_j^{(t)} = q_j p_j \left((1 - \lambda_j^{(t)})\mathbf{A}_j^{(t)} - \mathbf{A}_j^* + \tfrac{1}{q_j p_j}\Delta_j^{(t)} \pm \gamma\right),$$

  $\lambda_j^{(t)} = |\langle \mathbf{A}_j^{(t)} - \mathbf{A}_j^*, \mathbf{A}_j^* \rangle|$, and $\Delta_j^{(t)} := \mathbf{E}[\mathbf{A}_S^{(t)} \vartheta_S^{(R)} \text{sign}(\mathbf{x}_S^*)]$, where $\|\Delta_j^{(t)}\| = \mathcal{O}(\sqrt{m}q_{i,j} p_j \epsilon_t \|\mathbf{A}^{(t)}\|)$.

- **Step III.B: Show that the empirical gradient vector concentrates around its expectation**– Since we only have access to the empirical gradient vectors, we show that these concentrate around their expected value via the following lemma.

**Lemma 6** (**Concentration of the empirical gradient vector**). *Given $p = \widetilde{\Omega}(mk^2)$ samples, the empirical gradient vector estimate corresponding to the $i$-th dictionary element, $\widehat{\mathbf{g}}_i^{(t)}$ concentrates around its expectation, i.e.,*

$$\|\widehat{\mathbf{g}}_i^{(t)} - \mathbf{g}_i^{(t)}\| \leq o(\tfrac{k}{m}\epsilon_t).$$

*with probability at least $(1 - \delta_{\mathbf{g}_i}^{(t)} - \delta_\beta^{(t)} - \delta_\mathcal{T}^{(t)} - \delta_{\text{HW}}^{(t)})$, where $\delta_{\mathbf{g}_i}^{(t)} = \exp(-\Omega(k))$.*

- *Step III.C: Show that the empirical gradient vector is correlated with the descent direction–* Next, in the following lemma we show that the empirical gradient vector $\widehat{\mathbf{g}}_j^{(t)}$ is correlated with the descent direction. This is the main result which enables the progress in the dictionary (and coefficients) at each iteration $t$.

  **Lemma 7** (**Empirical gradient vector is correlated with the descent direction**). *Suppose $\mathbf{A}^{(t)}$ is $(\epsilon_t, 2)$-near to $\mathbf{A}^*$, $k = \mathcal{O}(\sqrt{n})$ and $\eta_A = \mathcal{O}(m/k)$. Then, with probability at least $(1 - \delta_\mathcal{T}^{(t)} - \delta_\beta^{(t)} - \delta_{\text{HW}}^{(t)} - \delta_{\mathbf{g}_i}^{(t)})$ the empirical gradient vector $\widehat{\mathbf{g}}_j^{(t)}$ is $(\Omega(k/m), \Omega(m/k), 0)$-correlated with $(\mathbf{A}_j^{(t)} - \mathbf{A}_j^*)$, and for any $t \in [T]$,*

  $$\|\mathbf{A}_j^{(t+1)} - \mathbf{A}_j^*\|^2 \leq (1 - \rho_-\eta_A)\|\mathbf{A}_j^{(t)} - \mathbf{A}_j^*\|^2.$$

  This result ensures for at any $t \in [T]$, the gradient descent-based updates made via (5) gets the columns of the dictionary estimate closer to the true dictionary, i.e., $\epsilon_{t+1} \leq \epsilon_t$. Moreover, this step requires closeness between the dictionary estimate $\mathbf{A}^{(t)}$ and $\mathbf{A}^*$, in the spectral norm-sense, as per Def 1.

STEP IV: SHOW THAT THE DICTIONARY MAINTAINS THE CLOSENESS PROPERTY

As discussed above, the closeness property (Def 1) is crucial to show that the gradient vector is correlated with the descent direction. Therefore, we now ensure that the updated dictionary $\mathbf{A}^{(t+1)}$ maintains this closeness property. Lemma 7 already ensures that $\epsilon_{t+1} \leq \epsilon_t$. As a result, we show that $\mathbf{A}^{(t+1)}$ maintains closeness in the spectral norm-sense as required by our algorithm, i.e., that it is still $(\epsilon_{t+1}, 2)$-close to the true dictionary. Also, since we use the gradient matrix in this analysis, we show that the empirical gradient matrix concentrates around its mean.

- *Step IV.A: The empirical gradient matrix concentrates around its expectation*: We first show that the empirical gradient matrix concentrates as formalized by the following lemma.

  **Lemma 8** (**Concentration of the empirical gradient matrix**). *With probability at least $(1 - \delta_\beta^{(t)} - \delta_\mathcal{T}^{(t)} - \delta_{\text{HW}}^{(t)} - \delta_{\mathbf{g}}^{(t)})$, $\|\widehat{\mathbf{g}}^{(t)} - \mathbf{g}^{(t)}\|$ is upper-bounded by $\mathcal{O}^*(\tfrac{k}{m}\|\mathbf{A}^*\|)$, where $\delta_{\mathbf{g}}^{(t)} = (n + m)\exp(-\Omega(m\sqrt{\log(n)}))$.*

- *Step IV.B: The "closeness" property is maintained after the updates made using the empirical gradient estimate*: Next, the following lemma shows that the updated dictionary $\mathbf{A}^{(t+1)}$ maintains the closeness property.

  **Lemma 9** ($\mathbf{A}^{(t+1)}$ **maintains closeness**). *Suppose $\mathbf{A}^{(t)}$ is $(\epsilon_t, 2)$ near to $\mathbf{A}^*$ with $\epsilon_t = \mathcal{O}^*(1/\log(n))$, and number of samples used in step $t$ is $p = \widetilde{\Omega}(mk^2)$, then with probability at least $(1 - \delta_\mathcal{T}^{(t)} - \delta_\beta^{(t)} - \delta_{\text{HW}}^{(t)} - \delta_{\mathbf{g}}^{(t)})$, $\mathbf{A}^{(t+1)}$ satisfies $\|\mathbf{A}^{(t+1)} - \mathbf{A}^*\| \leq 2\|\mathbf{A}^*\|$.*

STEP V: COMBINE RESULTS TO SHOW THE MAIN RESULT

*Proof of Theorem 1.* From Lemma 7 we have that with probability at least $(1 - \delta_\mathcal{T}^{(t)} - \delta_\beta^{(t)} - \delta_{\text{HW}}^{(t)} - \delta_{\mathbf{g}_i}^{(t)})$, $\mathbf{g}_j^{(t)}$ is $(\Omega(k/m), \Omega(m/k), 0)$-correlated with $\mathbf{A}_j^*$. Further, Lemma 9 ensures that each iterate maintains the closeness property. Now, applying Lemma 15 we have that, for $\eta_A \leq \Theta(m/k)$, with probability at least $(1 - \delta_{\text{alg}}^{(t)})$ any $t \in [T]$ satisfies

$$\|\mathbf{A}_j^{(t)} - \mathbf{A}_j^*\|^2 \leq (1 - \omega)^t\|\mathbf{A}_j^{(0)} - \mathbf{A}_j^*\|^2 \leq (1 - \omega)^t\epsilon_0^2.$$

where for $0 < \omega < 1/2$ with $\omega = \Omega(k/m)\eta_A$. That is, the updates converge geometrically to $\mathbf{A}^*$. Further, from Lemma 3, we have that the result on the error incurred by the coefficients. Here,

$\delta_{\mathrm{alg}}^{(t)} = \delta_{\mathcal{T}}^{(t)} + \delta_{\beta}^{(t)} + \delta_{\mathrm{HW}}^{(t)} + \delta_{\mathbf{g}_i}^{(t)} + \delta_{\mathbf{g}}^{(t)})$. That is, the updates converge geometrically to $\mathbf{A}^*$. Further, from Lemma 3, we have that the error in the coefficients only depends on the error in the dictionary, which leads us to our result on the error incurred by the coefficients. This completes the proof of our main result. □

## C APPENDIX: PROOF OF LEMMAS

We present the proofs of the Lemmas used to establish our main result. Also, see Table 3 for a map of dependence between the results, and Appendix D for proofs of intermediate results.

**Table 3:** Proof map: dependence of results.

| Lemmas | Result | Dependence |
|---|---|---|
| Lemma 1 | Signed-support recovery by coefficient initialization step | – |
| Lemma 2 | IHT update step preserves the correct signed-support | Claim 1, Lemma 1, and Claim 2 |
| Lemma 3 | Upper-bound on the error in coefficient estimation | Claim 1, Claim 2, Claim 3, and Claim 4 |
| Lemma 4 | Expression for the coefficient estimate at the end of $R$-th IHT iteration | Claim 5 |
| Lemma 5 | Expression for the expected gradient vector | Lemma 4 and Claim 7 |
| Lemma 6 | Concentration of the empirical gradient vector | Claim 8 and Claim 9 |
| Lemma 7 | Empirical gradient vector is correlated with the descent direction | Lemma 5, Claim 7 and Lemma 6 |
| Lemma 8 | Concentration of the empirical gradient matrix | Claim 8 and Claim 10 |
| Lemma 9 | $\mathbf{A}^{(t+1)}$ maintains closeness | Lemma 5, Claim 7 and Lemma 8 |

| Claims | Result | Dependence |
|---|---|---|
| Claim 1 | Incoherence of $\mathbf{A}^{(t)}$ | – |
| Claim 2 | Bound on $\beta_j^{(t)}$: the noise component in coefficient estimate that depends on $\epsilon_t$ | – |
| Claim 3 | Error in coefficient estimation for a general iterate $(r+1)$ | – |
| Claim 4 | An intermediate result for bounding the error in coefficient calculations | Claim 2 |
| Claim 5 | Bound on the noise term in the estimation of a coefficient element in the support | Claim 6 |
| Claim 6 | An intermediate result for $\vartheta_{i_1}^{(R)}$ calculations | Claim 3 |
| Claim 7 | Bound on the noise term in expected gradient vector estimate | Claim 6 and Claim 2 |
| Claim 8 | An intermediate result for concentration results | Lemma 2 ,Lemma 4 and Claim 5 |
| Claim 9 | Bound on variance parameter for concentration of gradient vector | Claim 5 |
| Claim 10 | Bound on variance parameter for concentration of gradient matrix | Lemma 2 , Lemma 4 and Claim 5 |

*Proof of Lemma 1.* Let $\mathbf{y} \in \mathbb{R}^n$ be general sample generated as $\mathbf{y} = \mathbf{A}^*\mathbf{x}^*$, where $\mathbf{x}^* \in \mathbb{R}^m$ is a sparse random vector with support $S = \mathrm{supp}(\mathbf{x}^*)$ distributed according to **D.4**.

The initial decoding step at the $t$-th iteration (shown in Algorithm 1) involves evaluating the inner-product between the estimate of the dictionary $\mathbf{A}^{(t)}$, and $\mathbf{y}$. The $i$-th element of the resulting vector

can be written as

$$\langle \mathbf{A}_i^{(t)}, \mathbf{y} \rangle = \langle \mathbf{A}_i^{(t)}, \mathbf{A}_i^* \rangle \mathbf{x}_i^* + \mathbf{w}_i,$$

where $\mathbf{w}_i = \langle \mathbf{A}_i^{(t)}, \mathbf{A}_{-i}^* \mathbf{x}_{-i}^* \rangle$. Now, since $\|\mathbf{A}_i^* - \mathbf{A}_i^{(t)}\|_2 \leq \epsilon_t$ and

$$\|\mathbf{A}_i^* - \mathbf{A}_i^{(t)}\|_2^2 = \|\mathbf{A}_i^*\|^2 + \|\mathbf{A}_i^{(t)}\|^2 - 2\langle \mathbf{A}_i^{(t)}, \mathbf{A}_i^* \rangle = 2 - 2\langle \mathbf{A}_i^{(t)}, \mathbf{A}_i^* \rangle,$$

we have

$$|\langle \mathbf{A}_i^{(t)}, \mathbf{A}_i^* \rangle| \geq 1 - \epsilon_t^2/2.$$

Therefore, the term

$$|\langle \mathbf{A}_i^{(t)}, \mathbf{A}_i^* \rangle \mathbf{x}_i^*| \begin{cases} \geq (1 - \frac{\epsilon_t^2}{2})C & \text{, if } i \in S, \\ = 0 & \text{, otherwise.} \end{cases}$$

Now, we focus on the $\mathbf{w}_i$ and show that it is small. By the definition of $\mathbf{w}_i$ we have

$$\mathbf{w}_i = \langle \mathbf{A}_i^{(t)}, \mathbf{A}_{-i}^* \mathbf{x}_{-i}^* \rangle = \sum_{\ell \neq i} \langle \mathbf{A}_i^{(t)}, \mathbf{A}_\ell^* \rangle \mathbf{x}_\ell^* = \sum_{\ell \in S \setminus \{i\}} \langle \mathbf{A}_i^{(t)}, \mathbf{A}_\ell^* \rangle \mathbf{x}_\ell^*.$$

Here, since $var(\mathbf{x}_\ell^*) = 1$, $\mathbf{w}_i$ is a zero-mean random variable with variance

$$var(\mathbf{w}_i) = \sum_{\ell \in S \setminus \{i\}} \langle \mathbf{A}_i^{(t)}, \mathbf{A}_\ell^* \rangle^2.$$

Now, each term in this sum can be bounded as,

$$\begin{aligned} \langle \mathbf{A}_i^{(t)}, \mathbf{A}_\ell^* \rangle^2 &= (\langle \mathbf{A}_i^{(t)} - \mathbf{A}_i^*, \mathbf{A}_\ell^* \rangle + \langle \mathbf{A}_i^*, \mathbf{A}_\ell^* \rangle)^2 \\ &\leq 2(\langle \mathbf{A}_i^{(t)} - \mathbf{A}_i^*, \mathbf{A}_\ell^* \rangle^2 + \langle \mathbf{A}_i^*, \mathbf{A}_\ell^* \rangle^2) \\ &\leq 2(\langle \mathbf{A}_i^{(t)} - \mathbf{A}_i^*, \mathbf{A}_\ell^* \rangle^2 + \frac{\mu^2}{n}). \end{aligned}$$

Next, $\sum_{\ell \neq i} \langle \mathbf{A}_i^{(t)} - \mathbf{A}_i^*, \mathbf{A}_\ell^* \rangle^2$ can be upper-bounded as

$$\sum_{\ell \in S \setminus \{i\}} \langle \mathbf{A}_i^{(t)} - \mathbf{A}_i^*, \mathbf{A}_\ell^* \rangle^2 \leq \|\mathbf{A}_{S \setminus \{i\}}^*\|^2 \epsilon_t^2.$$

Therefore, we have the following as per our assumptions on $\mu$ and $k$,

$$\|\mathbf{A}_{S \setminus \{i\}}^*\|^2 \leq (1 + k\frac{\mu}{\sqrt{n}}) \leq 2,$$

using Gershgorin Circle Theorem (Gershgorin, 1931). Therefore, we have

$$\sum_{\ell \in S \setminus \{i\}} \langle \mathbf{A}_i^{(t)} - \mathbf{A}_i^*, \mathbf{A}_\ell^* \rangle^2 \leq 2\epsilon_t^2.$$

Finally, we have that

$$\sum_{\ell \in S \setminus \{i\}} \langle \mathbf{A}_i^{(t)}, \mathbf{A}_\ell^* \rangle^2 \leq 2(2\epsilon_t^2 + k\frac{\mu^2}{n}) = \mathcal{O}^*(\epsilon_t^2).$$

Now, we apply the Chernoff bound for sub-Gaussian random variables $\mathbf{w}_i$ (shown in Lemma 12) to conclude that

$$\mathbf{Pr}[|\mathbf{w}_i| \geq C/4] \leq 2\exp(-\frac{C^2}{\mathcal{O}^*(\epsilon_t^2)}).$$

Further, $\mathbf{w}_i$ corresponding to each $m$ should follow this bound, applying union bound we conclude that

$$\mathbf{Pr}[\max_i |\mathbf{w}_i| \geq C/4] \leq 2m\exp(-\frac{C^2}{\mathcal{O}^*(\epsilon_t^2)}) := \delta_\mathcal{T}^{(t)}.$$

$\square$

*Proof of Lemma 2.* Consider the $(r+1)$-th iterate $\mathbf{x}^{(r+1)}$ for the $t$-th dictionary iterate, where $\|\mathbf{A}_i^{(t)} - \mathbf{A}_i^*\| \leq \epsilon_t$ for all $i \in [1, m]$ evaluated as the following by the update step described in Algorithm 1,

$$
\begin{aligned}
\mathbf{x}^{(r+1)} &= \mathbf{x}^{(r)} - \eta_x^{(r+1)} \mathbf{A}^{(t)^\top}(\mathbf{A}^{(t)}\mathbf{x}^{(r)} - \mathbf{y}) \\
&= (\mathbf{I} - \eta_x^{(r+1)} \mathbf{A}^{(t)^\top} \mathbf{A}^{(t)})\mathbf{x}^{(r)} - \eta_x^{(r+1)} \mathbf{A}^{(t)^\top} \mathbf{A}^* \mathbf{x}^*,
\end{aligned}
\tag{7}
$$

where $\eta_x^{(1)} < 1$ is the learning rate or the step-size parameter. Now, using Lemma 1 we know that $\mathbf{x}^{(0)}$ (3) has the correct signed-support with probability at least $(1 - \delta_{\mathcal{T}}^{(t)})$. Further, since $\mathbf{A}^{(t)^\top} \mathbf{A}^*$ can be written as

$$
\mathbf{A}^{(t)^\top} \mathbf{A}^* = (\mathbf{A}^{(t)} - \mathbf{A}^*)^\top \mathbf{A}^* + \mathbf{A}^{*\top} \mathbf{A}^*,
$$

we can write the $(r+1)$-th iterate of the coefficient update step using (7) as

$$
\mathbf{x}^{(r+1)} = (\mathbf{I} - \eta_x^{(r+1)} \mathbf{A}^{(t)^\top} \mathbf{A}^{(t)})\mathbf{x}^{(r)} - \eta_x^{(r+1)}(\mathbf{A}^{(t)} - \mathbf{A}^*)^\top \mathbf{A}^* \mathbf{x}^* + \eta_x^{(r+1)} \mathbf{A}^{*\top} \mathbf{A}^* \mathbf{x}^*.
$$

Further, the $j$-th entry of this vector is given by

$$
\mathbf{x}_j^{(r+1)} = (\mathbf{I} - \eta_x^{(r+1)} \mathbf{A}^{(t)^\top} \mathbf{A}^{(t)})_{(j,:)}\mathbf{x}^{(r)} - \eta_x^{(r+1)}((\mathbf{A}^{(t)} - \mathbf{A}^*)^\top \mathbf{A}^*)_{(j,:)}\mathbf{x}^* + \eta_x^{(r+1)}(\mathbf{A}^{*\top} \mathbf{A}^*)_{(j,:)}\mathbf{x}^*.
\tag{8}
$$

We now develop an expression for the $j$-th element of each of the term in (8) as follows. First, we can write the first term as

$$
(\mathbf{I} - \eta_x^{(r+1)} \mathbf{A}^{(t)^\top} \mathbf{A}^{(t)})_{(j,:)}\mathbf{x}^{(r)} = (1 - \eta_x^{(r+1)})\mathbf{x}_j^{(r)} - \eta_x^{(r+1)} \sum_{i \neq j} \langle \mathbf{A}_j^{(t)}, \mathbf{A}_i^{(t)} \rangle \mathbf{x}_i^{(r)}.
$$

Next, the second term in (8) can be expressed as

$$
\begin{aligned}
\eta_x^{(r+1)}((\mathbf{A}^{(t)} - \mathbf{A}^*)^\top \mathbf{A}^*)_{(j,:)}\mathbf{x}^* &= \eta_x^{(r+1)} \sum_i \langle \mathbf{A}_j^{(t)} - \mathbf{A}_j^*, \mathbf{A}_i^* \rangle \mathbf{x}_i^* \\
&= \eta_x^{(r+1)} \langle \mathbf{A}_j^{(t)} - \mathbf{A}_j^*, \mathbf{A}_j^* \rangle \mathbf{x}_j^* + \eta_x^{(r+1)} \sum_{i \neq j} \langle \mathbf{A}_j^{(t)} - \mathbf{A}_j^*, \mathbf{A}_i^* \rangle \mathbf{x}_i^*.
\end{aligned}
$$

Finally, we have the following expression for the third term,

$$
\eta_x^{(r+1)}(\mathbf{A}^{*\top} \mathbf{A}^*)_{(j,:)}\mathbf{x}^* = \eta_x^{(r+1)} \mathbf{x}_j^* + \eta_x^{(r+1)} \sum_{i \neq j} \langle \mathbf{A}_j^*, \mathbf{A}_i^* \rangle \mathbf{x}_i^*.
$$

Now using our definition of $\lambda_j^{(t)} = |\langle \mathbf{A}_j^{(t)} - \mathbf{A}_j^*, \mathbf{A}_j^* \rangle| \leq \frac{\epsilon_t^2}{2}$, combining all the results for (8), and using the fact that since $\mathbf{A}^{(t)}$ is close to $\mathbf{A}^*$, vectors $\mathbf{A}_j^{(t)} - \mathbf{A}_j^*$ and $\mathbf{A}_j^*$ enclose an obtuse angle, we have the following for the $j$-th entry of the $(r+1)$-th iterate, $\mathbf{x}^{(r+1)}$ is given by

$$
\mathbf{x}_j^{(r+1)} = (1 - \eta_x^{(r+1)})\mathbf{x}_j^{(r)} + \eta_x^{(r+1)}(1 - \lambda_j^{(t)})\mathbf{x}_j^* + \eta_x^{(r+1)}\xi_j^{(r+1)}.
\tag{9}
$$

Here $\xi_j^{(r+1)}$ is defined as

$$
\xi_j^{(r+1)} := \sum_{i \neq j}(\langle \mathbf{A}_j^{(t)} - \mathbf{A}_j^*, \mathbf{A}_i^* \rangle + \langle \mathbf{A}_j^*, \mathbf{A}_i^* \rangle)\mathbf{x}_i^* - \sum_{i \neq j} \langle \mathbf{A}_j^{(t)}, \mathbf{A}_i^{(t)} \rangle \mathbf{x}_i^{(r)}.
$$

Since, $\langle \mathbf{A}_j^*, \mathbf{A}_i^* \rangle - \langle \mathbf{A}_j^{(t)}, \mathbf{A}_i^{(t)} \rangle = \langle \mathbf{A}_j^*, \mathbf{A}_i^* - \mathbf{A}_i^{(t)} \rangle + \langle \mathbf{A}_j^* - \mathbf{A}_j^{(t)}, \mathbf{A}_i^{(t)} \rangle$, we can write $\xi_j^{(r+1)}$ as

$$
\xi_j^{(r+1)} = \beta_j^{(t)} + \sum_{i \neq j} \langle \mathbf{A}_j^{(t)}, \mathbf{A}_i^{(t)} \rangle(\mathbf{x}_i^* - \mathbf{x}_i^{(r)}),
\tag{10}
$$

where $\beta_j^{(t)}$ is defined as

$$
\beta_j^{(t)} := \sum_{i \neq j}(\langle \mathbf{A}_j^*, \mathbf{A}_i^* - \mathbf{A}_i^{(t)} \rangle + \langle \mathbf{A}_j^* - \mathbf{A}_j^{(t)}, \mathbf{A}_i^{(t)} \rangle + \langle \mathbf{A}_j^{(t)} - \mathbf{A}_j^*, \mathbf{A}_i^* \rangle)\mathbf{x}_i^*.
\tag{11}
$$

Note that $\beta_j^{(t)}$ does not change for each iteration $r$ of the coefficient update step. Further, by Claim 2 we show that $|\beta_j^{(t)}| \leq t_\beta = \mathcal{O}(\sqrt{k}\epsilon_t)$ with probability at least $(1 - \delta_\beta^{(t)})$. Next, we define $\widetilde{\xi}_j^{(r+1)}$ as

$$\widetilde{\xi}_j^{(r+1)} := \beta_j^{(t)} + \sum_{i \neq j} |\langle \mathbf{A}_j^{(t)}, \mathbf{A}_i^{(t)} \rangle| |\mathbf{x}_i^* - \mathbf{x}_i^{(r)}|. \tag{12}$$

where $\xi_j^{(r+1)} \leq \widetilde{\xi}_j^{(r+1)}$. Further, using Claim 1,

$$\widetilde{\xi}_j^{(r+1)} \leq t_\beta + \tfrac{\mu_t}{\sqrt{n}} \|\mathbf{x}_j^* - \mathbf{x}_j^{(r)}\|_1 := \widetilde{\xi}_{\max}^{(r+1)} = \widetilde{\mathcal{O}}(\tfrac{k}{\sqrt{n}}), \tag{13}$$

since $\|\mathbf{x}^{(r-1)} - \mathbf{x}^*\|_1 = \mathcal{O}(k)$. Therefore, for the $(r+1)$-th iteration, we choose the threshold to be

$$\tau^{(r+1)} := \eta_x^{(r+1)} \widetilde{\xi}_{\max}^{(r+1)}, \tag{14}$$

and the step-size by setting the "noise" component of (9) to be smaller than the "signal" part, specifically, half the signal component, i.e.,

$$\eta_x^{(r+1)} \widetilde{\xi}_{\max}^{(r+1)} \leq \tfrac{(1-\eta_x^{(r+1)})}{2} \mathbf{x}_{\min}^{(r)} + \tfrac{\eta_x^{(r+1)}}{2}(1 - \tfrac{\epsilon_t^2}{2})C,$$

Also, since we choose the threshold as $\tau^{(r)} := \eta_x^{(r)} \widetilde{\xi}_{\max}^{(r)}$, $\mathbf{x}_{\min}^{(r)} = \eta_x^{(r)} \widetilde{\xi}_{\max}^{(r)}$, where $\mathbf{x}_{\min}^{(0)} = C/2$, we have the following for the $(r+1)$-th iteration,

$$\eta_x^{(r+1)} \widetilde{\xi}_{\max}^{(r+1)} \leq \tfrac{(1-\eta_x^{(r+1)})}{2} \eta_x^{(r)} \widetilde{\xi}_{\max}^{(r)} + \tfrac{\eta_x^{(r+1)}}{2}(1 - \tfrac{\epsilon_t^2}{2})C.$$

Therefore, for this step we choose $\eta_x^{(r+1)}$ as

$$\eta_x^{(r+1)} \leq \frac{\tfrac{\eta_x^{(r)}}{2} \widetilde{\xi}_{\max}^{(r)}}{\widetilde{\xi}_{\max}^{(r+1)} + \tfrac{\eta_x^{(r)}}{2} \widetilde{\xi}_{\max}^{(r)} - \tfrac{1}{2}(1 - \tfrac{\epsilon_t^2}{2})C}, \tag{15}$$

Therefore, $\eta_x^{(r+1)}$ can be chosen as

$$\eta_x^{(r+1)} \leq c^{(r+1)}(\epsilon_t, \mu, k, n),$$

for a small constant $c^{(r+1)}(\epsilon_t, \mu, k, n)$, $\eta_x^{(r+1)}$. In addition, if we set all $\eta_x^{(r)} = \eta_x$, we have that $\eta_x = \widetilde{\Omega}(\tfrac{k}{\sqrt{n}})$ and therefore $\tau^{(r)} = \tau = \widetilde{\Omega}(\tfrac{k^2}{n})$. Further, since we initialize with the hard-thresholding step, the entries in $|\mathbf{x}^{(0)}| \geq C/2$. Here, we define $\widetilde{\xi}_{\max}^{(0)} = C$ and $\eta_x^{(0)} = 1/2$, and set the threshold for initial step as $\eta_x^{(0)} \widetilde{\xi}_{\max}^{(0)}$.

$\square$

*Proof of Lemma 3.* Using the definition of $\widetilde{\xi}_{i_1}^{(\ell)}$ as in (12), we have

$$\widetilde{\xi}_{i_1}^{(\ell)} = \beta_{i_1}^{(t)} + \sum_{i_2 \neq i_1} |\langle \mathbf{A}_{i_1}^{(t)}, \mathbf{A}_{i_2}^{(t)} \rangle| |\mathbf{x}_{i_2}^* - \mathbf{x}_{i_2}^{(\ell-1)}|.$$

From Claim 2 we have that $|\beta_{i_1}^{(t)}| \leq t_\beta$ with probability at least $(1 - \delta_\beta^{(t)})$. Further, using Claim 1 , and letting $C_i^{(\ell)} := |\mathbf{x}_i^* - \mathbf{x}_i^{(\ell)}| = |\mathbf{x}_i^{(\ell)} - \mathbf{x}_i^*|$, $\widetilde{\xi}_{i_1}^{(\ell)}$ can be upper-bounded as

$$\widetilde{\xi}_{i_1}^{(\ell)} \leq \beta_{i_1}^{(t)} + \tfrac{\mu_t}{\sqrt{n}} \sum_{i_2 \neq i_1} C_{i_2}^{(\ell-1)}. \tag{16}$$

Rearranging the expression for $(r+1)$-th update (9), and using (16) we have the following upper-bound

$$C_{i_1}^{(r+1)} \leq (1 - \eta_x^{(r+1)})C_{i_1}^{(r)} + \eta_x^{(r+1)} \lambda_{i_1}^{(t)} |\mathbf{x}_{i_1}^*| + \eta_x^{(r+1)} \widetilde{\xi}_{i_1}^{(r+1)}.$$

Next, recursively substituting in for $C_{i_1}^{(r)}$, where we define $\prod_{q=\ell}^{\ell}(1 - \eta_x^{(q+1)}) = 1$,

$$C_{i_1}^{(r+1)} \leq C_{i_1}^{(0)} \prod_{q=0}^{r}(1 - \eta_x^{(q+1)}) + \lambda_{i_1}^{(t)} |\mathbf{x}_{i_1}^*| \sum_{\ell=1}^{r+1} \eta_x^{(\ell)} \prod_{q=\ell}^{r+1}(1 - \eta_x^{(q+1)}) + \sum_{\ell=1}^{r+1} \eta_x^{(\ell)} \widetilde{\xi}_{i_1}^{(\ell)} \prod_{q=\ell}^{r+1}(1 - \eta_x^{(q+1)}).$$

Substituting for the upper-bound of $\widetilde{\xi}_{i_1}^{(\ell)}$ from (16),

$$C_{i_1}^{(r+1)} \leq \alpha_{i_1}^{(r+1)} + \frac{\mu_t}{\sqrt{n}} \sum_{\ell=1}^{r+1} \eta_x^{(\ell)} \sum_{i_2 \neq i_1} C_{i_2}^{(\ell-1)} \prod_{q=\ell}^{r+1} (1 - \eta_x^{(q+1)}). \tag{17}$$

Here, $\alpha_{i_1}^{(r+1)}$ is defined as

$$\alpha_{i_1}^{(r+1)} = C_{i_1}^{(0)} \prod_{q=0}^{r} (1 - \eta_x^{(q+1)}) + (\lambda_{i_1}^{(t)} |\mathbf{x}_{i_1}^*| + \beta_{i_1}^{(t)}) \sum_{\ell=1}^{r+1} \eta_x^{(\ell)} \prod_{q=\ell}^{r+1} (1 - \eta_x^{(q+1)}). \tag{18}$$

Our aim now will be to express $C_{i_1}^{(\ell)}$ for $\ell > 0$ in terms of $C_{i_2}^{(0)}$. Let each $\alpha_j^{(\ell)} \leq \alpha_i^{(\ell)}$ where $j = i_1, i_2, \ldots, i_k$. Similarly, let $C_j^{(0)} \leq C_i^{(0)}$ for $j = i_1, i_2, \ldots, i_k$, and all $\eta_x^{(\ell)} = \eta_x$. Then, using Claim 3 we have the following expression for $C_{i_1}^{(R+1)}$,

$$C_{i_1}^{(R+1)} \leq \alpha_{i_1}^{(R+1)} + (k-1)\eta_x \frac{\mu_t}{\sqrt{n}} \sum_{\ell=1}^{R} \alpha_{\max}^{(\ell)} \left(1 - \eta_x + \eta_x \frac{\mu_t}{\sqrt{n}}\right)^{R-\ell}$$
$$+ (k-1)\eta_x \frac{\mu_t}{\sqrt{n}} C_{\max}^{(0)} \left(1 - \eta_x + \eta_x \frac{\mu_t}{\sqrt{n}}\right)^{R}.$$

Here, $(1 - \eta_x)^R \leq (1 - \eta_x + \eta_x \frac{\mu_t}{\sqrt{n}})^R \leq \delta_R$. Next from Claim 4 we have that with probability at least $(1 - \delta_\beta^{(t)})$,

$$\sum_{\ell=1}^{R} \alpha_{\max}^{(\ell)} \left(1 - \eta_x + \eta_x \frac{\mu_t}{\sqrt{n}}\right)^{R-\ell} \leq C_{\max}^{(0)} R \delta_R + \frac{1}{\eta_x (1 - \frac{\mu_t}{\sqrt{n}})} \left(\frac{\epsilon_t^2}{2} |\mathbf{x}_{\max}^*| + t_\beta\right).$$

Therefore, for $c_x = \frac{\mu_t}{\sqrt{n}} / (1 - \frac{\mu_t}{\sqrt{n}})$

$$C_{i_1}^{(R+1)} \leq \alpha_{i_1}^{(R+1)} + (k-1)c_x \left(\frac{\epsilon_t^2}{2} |\mathbf{x}_{\max}^*| + t_\beta\right) + (R+1)(k-1)\eta_x \frac{\mu_t}{\sqrt{n}} C_{\max}^{(0)} \delta_R.$$

Now, using the definition of $\alpha_{i_1}^{(R+1)}$, and using the result on sum of geometric series, we have

$$\alpha_{i_1}^{(R+1)} = C_{i_1}^{(0)} (1 - \eta_x)^{R+1} + (\lambda_{i_1}^{(t)} |\mathbf{x}_{i_1}^*| + \beta_{i_1}^{(t)}) \sum_{s=1}^{R+1} \eta_x (1 - \eta_x)^{R-s+1},$$
$$= C_{i_1}^{(0)} \delta_R + \lambda_{i_1}^{(t)} |\mathbf{x}_{i_1}^*| + \beta_{i_1}^{(t)} \leq C_{i_1}^{(0)} \delta_{R+1} + \frac{\epsilon_t^2}{2} |\mathbf{x}_{\max}^*| + t_\beta.$$

Therefore, $C_{i_1}^{(R)}$ is upper-bounded as

$$C_{i_1}^{(R)} \leq (c_x k + 1)\left(\frac{\epsilon_t^2}{2} |\mathbf{x}_{\max}^*| + t_\beta\right) + (R+1)k\eta_x \frac{\mu_t}{\sqrt{n}} C_{\max}^{(0)} \delta_R + C_{i_1}^{(0)} \delta_R.$$

Further, since $k = \mathcal{O}(\sqrt{n}/\mu \log(n))$, $kc_x < 1$, therefore, we have

$$C_{i_1}^{(R)} \leq \mathcal{O}(t_\beta) + (R+1)k\eta_x \frac{\mu_t}{\sqrt{n}} C_{\max}^{(0)} \delta_R + C_{i_1}^{(0)} \delta_R,$$

with probability at least $(1 - \delta_\beta^{(t)})$. Here, $(R+1)k\eta_x \frac{\mu_t}{\sqrt{n}} C_{\max}^{(0)} \delta_R + C_{i_1}^{(0)} \delta_R \cong 0$ for an appropriately large $R$. Therefore, the error in each non-zero coefficient is

$$C_{i_1}^{(R)} = \mathcal{O}(t_\beta).$$

with probability at least $(1 - \delta_\beta^{(t)})$. □

*Proof of Lemma 4.* Using the expression for $\mathbf{x}_{i_1}^{(R)}$ as defined in (9), and recursively substituting for $\mathbf{x}_{i_1}^{(r)}$ we have

$$\mathbf{x}_{i_1}^{(R)} = (1 - \eta_x)^R \mathbf{x}_j^{(0)} + \mathbf{x}_{i_1}^* \sum_{r=1}^{R} \eta_x (1 - \lambda_{i_1}^{(t)})(1 - \eta_x)^{R-r} + \sum_{r=1}^{R} \eta_x \xi_{i_1}^{(r)} (1 - \eta_x)^{R-r},$$

where we set all $\eta_x^r$ to be $\eta_x$. Further, on defining

$$\vartheta_{i_1}^{(R)} := \sum_{r=1}^{R} \eta_x \xi_{i_1}^{(r)} (1 - \eta_x)^{R-r} + \gamma_{i_1}^{(R)}, \tag{19}$$

where $\gamma_{i_1}^{(R)} := (1 - \eta_x)^R (\mathbf{x}_{i_1}^{(0)} - \mathbf{x}_{i_1}^* (1 - \lambda_{i_1}^{(t)}))$, we have

$$\mathbf{x}_{i_1}^{(R)} = (1 - \eta_x)^R \mathbf{x}_{i_1}^{(0)} + \mathbf{x}_{i_1}^* (1 - \lambda_{i_1}^{(t)})(1 - (1 - \eta_x)^R) + \sum_{r=1}^{R} \eta_x \xi_{i_1}^{(r)} (1 - \eta_x)^{R-r},$$

$$= \mathbf{x}_{i_1}^* (1 - \lambda_{i_1}^{(t)}) + \vartheta_{i_1}^{(R)}. \tag{20}$$

Note that $\gamma_{i_1}^{(R)}$ can be made appropriately small by choice of $R$. Further, by Claim 5 we have

$$|\vartheta_{i_1}^{(R)}| \le \mathcal{O}(t_\beta).$$

with probability at least $(1 - \delta_\beta^{(t)})$, where $t_\beta = \mathcal{O}(\sqrt{k\epsilon_t})$. $\qquad\square$

*Proof of Lemma 5.* From Lemma 4 we have that for each $j \in S$,

$$\widehat{\mathbf{x}}_S := \mathbf{x}_S^{(R)} = (\mathbf{I} - \Lambda_S^{(t)})\mathbf{x}_S^* + \vartheta_S^{(R)},$$

with probability at least $(1 - \delta_\mathcal{T}^{(t)} - \delta_\beta^{(t)})$. Further, let $\mathcal{F}_{\mathbf{x}^*}$ be the event that $\mathrm{sign}(\mathbf{x}^*) = \mathrm{sign}(\widehat{\mathbf{x}})$, and let $\mathbb{1}_{\mathcal{F}_{\mathbf{x}^*}}$ denote the indicator function corresponding to this event. As we show in Lemma 2, this event occurs with probability at least $(1 - \delta_\beta^{(t)} - \delta_\mathcal{T}^{(t)})$. Using this, we can write the expected gradient vector corresponding to the $j$-th sample as $\mathbb{1}_{\mathcal{F}_{\mathbf{x}^*}}$

$$\mathbf{g}_j^{(t)} = \mathbf{E}[(\mathbf{A}^{(t)}\widehat{\mathbf{x}} - \mathbf{y})\mathrm{sign}(\mathbf{x}_j^*)\mathbb{1}_{\mathcal{F}_{\mathbf{x}^*}}] + \mathbf{E}[(\mathbf{A}^{(t)}\widehat{\mathbf{x}} - \mathbf{y})\mathrm{sign}(\mathbf{x}_j^*)\mathbb{1}_{\overline{\mathcal{F}}_{\mathbf{x}^*}}],$$

$$= \mathbf{E}[(\mathbf{A}^{(t)}\widehat{\mathbf{x}} - \mathbf{y})\mathrm{sign}(\mathbf{x}_j^*)\mathbb{1}_{\mathcal{F}_{\mathbf{x}^*}}] \pm \gamma.$$

Here, $\gamma := \mathbf{E}[(\mathbf{A}^{(t)}\widehat{\mathbf{x}} - \mathbf{y})\mathrm{sign}(\mathbf{x}_j^*)\mathbb{1}_{\overline{\mathcal{F}}_{\mathbf{x}^*}}]$ is small and depends on $\delta_\mathcal{T}^{(t)}$ and $\delta_\beta^{(t)}$, which in turn drops with $\epsilon_t$. Therefore, $\gamma$ diminishes with $\epsilon_t$. Further, since $\mathbb{1}_{\mathcal{F}_{\mathbf{x}^*}} + \mathbb{1}_{\overline{\mathcal{F}}_{\mathbf{x}^*}} = 1$, and $\mathbf{Pr}[\mathcal{F}_{\mathbf{x}^*}] = (1 - \delta_\beta^{(t)} - \delta_\mathcal{T}^{(t)})$, is very large,

$$\mathbf{g}_j^{(t)} = \mathbf{E}[(\mathbf{A}^{(t)}\widehat{\mathbf{x}} - \mathbf{y})\mathrm{sign}(\mathbf{x}_j^*)(1 - \mathbb{1}_{\overline{\mathcal{F}}_{\mathbf{x}^*}})] \pm \gamma,$$

$$= \mathbf{E}[(\mathbf{A}^{(t)}\widehat{\mathbf{x}} - \mathbf{y})\mathrm{sign}(\mathbf{x}_j^*)] \pm \gamma.$$

Therefore, we can write $\mathbf{g}_j^{(t)}$ as

$$\mathbf{g}_j^{(t)} = \mathbf{E}[(\mathbf{A}^{(t)}\widehat{\mathbf{x}} - \mathbf{y})\mathrm{sign}(\mathbf{x}_j^*)] \pm \gamma,$$

$$= \mathbf{E}[(1 - \eta_x)^R \mathbf{A}_S^{(t)} \mathbf{x}_S^{(0)} + \mathbf{A}_S^{(t)}(\mathbf{I} - \Lambda_S^{(t)})\mathbf{x}_S^* + \mathbf{A}_S^{(t)}\vartheta_S^{(R)} - \mathbf{A}_S^* x_S^*)\mathrm{sign}(\mathbf{x}_j^*)] \pm \gamma.$$

Since $\mathbf{E}[(1 - \eta_x)^R \mathbf{A}_S^{(t)} \mathbf{x}_S^{(0)}]$ can be made very small by choice of $R$, we absorb this term in $\gamma$. Therefore,

$$\mathbf{g}_j^{(t)} = \mathbf{E}[\mathbf{A}_S^{(t)}(\mathbf{I} - \Lambda_S^{(t)})\mathbf{x}_S^* + \mathbf{A}_S^{(t)}\vartheta_S^{(R)} - \mathbf{A}_S^* x_S^*)\mathrm{sign}(\mathbf{x}_j^*)] \pm \gamma.$$

Writing the expectation by sub-conditioning on the support,

$$\mathbf{g}_j^{(t)} = \mathbf{E}_S[\mathbf{E}_{x_S^*}[\mathbf{A}_S^{(t)}(\mathbf{I} - \Lambda_S^{(t)})\mathbf{x}_S^* \mathrm{sign}(\mathbf{x}_j^*) - \mathbf{A}_S^* x_S^* \mathrm{sign}(\mathbf{x}_j^*) + \mathbf{A}_S^{(t)}\vartheta_S^{(R)}\mathrm{sign}(\mathbf{x}_j^*)|S]] \pm \gamma,$$

$$= \mathbf{E}_S[\mathbf{A}_S^{(t)}(\mathbf{I} - \Lambda_S^{(t)})\mathbf{E}_{x_S^*}[\mathbf{x}_S^* \mathrm{sign}(\mathbf{x}_j^*)|S] - \mathbf{A}_S^* \mathbf{E}_{x_S^*}[\mathbf{x}_S^* \mathrm{sign}(\mathbf{x}_j^*)|S]] + \mathbf{E}[\mathbf{A}_S^{(t)}\vartheta_S^{(R)}\mathrm{sign}(\mathbf{x}_j^*)] \pm \gamma,$$

$$= \mathbf{E}_S[p_j(1 - \lambda_j^{(t)})\mathbf{A}_j^{(t)} - p_j \mathbf{A}_j^*] + \Delta_j^{(t)} \pm \gamma,$$

where we have used the fact that $\mathbf{E}_{x_S^*}[\mathrm{sign}(\mathbf{x}_j^*)] = 0$ and introduced

$$\Delta_j^{(t)} = \mathbf{E}[\mathbf{A}_S^{(t)}\vartheta_S^{(R)}\mathrm{sign}(\mathbf{x}_j^*)].$$

Next, since $p_j = \mathbf{E}_{x_S^*}[\mathbf{x}_j^* \text{sign}(\mathbf{x}_j^*)|j \in S]$, therefore,

$$\mathbf{g}_j^{(t)} = \mathbf{E}_S[p_j(1 - \lambda_j^{(t)})\mathbf{A}_j^{(t)} - p_j\mathbf{A}_j^*] + \Delta_j^{(t)} \pm \gamma.$$

Further, since $q_j = \mathbf{Pr}[j \in S] = \mathcal{O}(k/m)$,

$$\mathbf{g}_j^{(t)} = q_j p_j\big((1 - \lambda_j^{(t)})\mathbf{A}_j^{(t)} - \mathbf{A}_j^* + \tfrac{1}{q_j p_j}\Delta_j^{(t)} \pm \gamma\big).$$

Further, by Claim 7 we have that

$$\|\Delta_j^{(t)}\| = \mathcal{O}(\sqrt{m}q_{i,j}p_j\epsilon_t\|\mathbf{A}^{(t)}\|)].$$

This completes the proof. $\qquad\square$

*Proof of Lemma 6.* Let $W = \{j : i \in \text{supp}(\mathbf{x}_{(j)}^*)\}$ and then we have that

$$\widehat{\mathbf{g}}_i^{(t)} = \tfrac{|W|}{p}\tfrac{1}{|W|}\sum_j(\mathbf{y}_{(j)} - \mathbf{A}^{(t)}\widehat{\mathbf{x}}_{(j)})\text{sign}(\widehat{\mathbf{x}}_{(j)}(i)),$$

where $\widehat{\mathbf{x}}_{(j)}(i)$ denotes the $i$-th element of the coefficient estimate corresponding to the $(j)$-th sample. Here, for $\ell = |W|$ the summation

$$\sum_j \tfrac{1}{\ell}(\mathbf{y}_{(j)} - \mathbf{A}^{(t)}\widehat{\mathbf{x}}_{(j)})\text{sign}(\widehat{\mathbf{x}}_{(j)}(i)),$$

has the same distribution as $\Sigma_{j=1}^\ell \mathbf{z}_j$, where each $\mathbf{z}_j$ belongs to a distribution as

$$\mathbf{z} := \tfrac{1}{\ell}(\mathbf{y} - \mathbf{A}^{(t)}\widehat{\mathbf{x}})\text{sign}(\widehat{\mathbf{x}}_i)|i \in S.$$

Also, $\mathbf{E}[(\mathbf{y} - \mathbf{A}^{(t)}\widehat{\mathbf{x}})\text{sign}(\widehat{\mathbf{x}}_i)] = q_i\mathbf{E}[\mathbf{z}]$, where $q_i = \mathbf{Pr}[\mathbf{x}_i^* \neq 0] = \Theta(\tfrac{k}{m})$. Therefore, since $p = \widetilde{\Omega}(mk^2)$, we have $\ell = pq_i = \widetilde{\Omega}(k^3)$ non-zero vectors,

$$\|\widehat{\mathbf{g}}_i^{(t)} - \mathbf{g}_i^{(t)}\| = \mathcal{O}(\tfrac{k}{m})\|\Sigma_{j=1}^\ell(\mathbf{z}_j - \mathbf{E}[\mathbf{z}])\|. \tag{21}$$

Let $\mathbf{w}_j = \mathbf{z}_j - \mathbf{E}[\mathbf{z}]$, we will now apply the vector Bernstein result shown in Lemma 11. For this, we require bounds on two parameters for these – $L := \|\mathbf{w}_j\|$ and $\sigma^2 := \|\Sigma_j\mathbf{E}[\|\mathbf{w}_j\|^2]\|$. Note that, since the quantity of interest is a function of $\mathbf{x}_i^*$, which are sub-Gaussian, they are only bounded *almost surely*. To this end, we will employ Lemma 14 (Lemma 45 in (Arora et al., 2015)) to get a handle on the concentration.

**Bound on the norm $\|\mathbf{w}\|$:** This bound is evaluated in Claim 8, which states that with probability at least $(1 - \delta_\beta^{(t)} - \delta_{\mathcal{T}}^{(t)} - \delta_{\text{HW}}^{(t)})$,

$$L := \|\mathbf{w}\| = \|\mathbf{z} - \mathbf{E}[\mathbf{z}]\| = \tfrac{2}{\ell}\|(\mathbf{y} - \mathbf{A}^{(t)}\widehat{\mathbf{x}})\text{sign}(\widehat{\mathbf{x}}_i)|i \in S\| \leq \tfrac{2}{\ell}\|(\mathbf{y} - \mathbf{A}^{(t)}\widehat{\mathbf{x}})\| = \widetilde{\mathcal{O}}(\tfrac{kt_\beta}{\ell}).$$

**Bound on variance parameter $\mathbf{E}[\|\mathbf{w}\|^2]$:** Using Claim 9, we have $\mathbf{E}[\|\mathbf{z}\|^2] = \mathcal{O}(k\epsilon_t^2) + \mathcal{O}(kt_\beta^2)$. Therefore, the bound on the variance parameter $\sigma^2$ is given by

$$\sigma^2 := \|\Sigma_j\mathbf{E}[\|\mathbf{w}_j\|^2]\| \leq \|\Sigma_j\mathbf{E}[\|\mathbf{z}_j\|^2]\| \leq \mathcal{O}(\tfrac{k}{\ell}\epsilon_t^2) + \mathcal{O}(\tfrac{kt_\beta^2}{\ell}).$$

From Claim 2 we have that with probability at least $(1 - \delta_\beta^{(t)})$, $t_\beta = \mathcal{O}(\sqrt{k}\epsilon_t)$. Applying vector Bernstein inequality shown in Lemma 11 and using Lemma 14 (Lemma 45 in (Arora et al., 2015)), choosing $\ell = \widetilde{\Omega}(k^3)$, we conclude

$$\|\Sigma_{j=1}^\ell \mathbf{z}_j - \mathbf{E}[\mathbf{z}]\| = \mathcal{O}(L) + \mathcal{O}(\sigma) = o(\epsilon_t),$$

with probability at least $(1 - \delta_{\mathbf{g}_i}^{(t)})$, where $\delta_{\mathbf{g}_i}^{(t)} = \exp(-\Omega(k))$. Finally, substituting in (21) we have

$$\|\widehat{\mathbf{g}}_i^{(t)} - \mathbf{g}_i^{(t)}\| = \mathcal{O}(\tfrac{k}{m})o(\epsilon_t).$$

with probability at least $(1 - \delta_{\mathbf{g}_i}^{(t)} - \delta_\beta^{(t)} - \delta_{\mathcal{T}}^{(t)} - \delta_{\text{HW}}^{(t)})$. $\qquad\square$

*Proof of Lemma 7.* Since we only have access to the empirical estimate of the gradient $\widehat{\mathbf{g}}_i^{(t)}$, we will show that this estimate is correlated with $(\mathbf{A}_j^{(t)} - \mathbf{A}_j^*)$. To this end, first from Lemma 6 we have that the empirical gradient vector concentrates around its mean, specifically,

$$\|\widehat{\mathbf{g}}_i^{(t)} - \mathbf{g}_i^{(t)}\| \leq o(\tfrac{k}{m}\epsilon_t),$$

with probability at least $(1 - \delta_{\mathbf{g}_i}^{(t)} - \delta_\beta^{(t)} - \delta_{\mathcal{T}}^{(t)} - \delta_{\mathrm{HW}}^{(t)})$. From Lemma 5, we have the following expression for the expected gradient vector

$$\mathbf{g}_j^{(t)} = p_j q_j (\mathbf{A}_j^{(t)} - \mathbf{A}_j^*) + p_j q_j(-\lambda_j^{(t)}\mathbf{A}_j^{(t)} + \tfrac{1}{p_j q_j}\Delta_j^{(t)} \pm \gamma).$$

Let $\mathbf{g}_j^{(t)} = 4\rho_-(\mathbf{A}_j^{(t)} - \mathbf{A}_j^*) + v$, where $4\rho_- = p_j q_j$ and $v$ is defined as

$$v = p_j q_j(-\lambda_j^{(t)}\mathbf{A}_j^{(t)} + \tfrac{1}{p_j q_j}\Delta_j^{(t)} \pm \gamma). \tag{22}$$

Then, $\widehat{\mathbf{g}}_i^{(t)}$ can be written as

$$\begin{aligned}
\widehat{\mathbf{g}}_i^{(t)} &= \widehat{\mathbf{g}}_i^{(t)} - \mathbf{g}_i^{(t)} + \mathbf{g}_i^{(t)}, \\
&= (\widehat{\mathbf{g}}_i^{(t)} - \mathbf{g}_i^{(t)}) + 4\rho_-(\mathbf{A}_j^{(t)} - \mathbf{A}_j^*) + v, \\
&= 4\rho_-(\mathbf{A}_j^{(t)} - \mathbf{A}_j^*) + \widetilde{v}, \tag{23}
\end{aligned}$$

where $\widetilde{v} = v + (\widehat{\mathbf{g}}_i^{(t)} - \mathbf{g}_i^{(t)})$. Let $\|\widetilde{v}\| \leq \rho_-\|\mathbf{A}_i^{(t)} - \mathbf{A}_i^*\|$. Using the definition of $v$ as shown in (22) we have

$$\|\widetilde{v}\| \leq q_j p_j \lambda_j^{(t)}\|\mathbf{A}_j^{(t)}\| + \|\Delta_j^{(t)}\| + o(\tfrac{k}{m}\epsilon_t) \pm \gamma.$$

Now for the first term, since $\|\mathbf{A}_j^{(t)}\| = 1$, we have $\lambda_j^{(t)} = |\langle \mathbf{A}_j^{(t)} - \mathbf{A}_j^*, \mathbf{A}_j^*\rangle| = \tfrac{1}{2}\|\mathbf{A}_j^{(t)} - \mathbf{A}_j^*\|^2$, therefore

$$q_j p_j \lambda_j^{(t)}\|\mathbf{A}_j^{(t)}\| = q_j p_j \tfrac{1}{2}\|\mathbf{A}_j^{(t)} - \mathbf{A}_j^*\|^2,$$

Further, using Claim 7

$$\|\Delta_j^{(t)}\| = \mathcal{O}(\sqrt{m} q_{i,j} p_{i_1} \epsilon_t \|\mathbf{A}^{(t)}\|).$$

Now, since $\|\mathbf{A}^{(t)} - \mathbf{A}^*\| \leq 2\|\mathbf{A}^*\|$ (the closeness property (Def.1) is maintained at every step using Lemma 9), and further since $\|\mathbf{A}^*\| = \mathcal{O}(\sqrt{m/n})$, we have that

$$\|\mathbf{A}^{(t)}\| \leq \|\mathbf{A}^{(t)} - \mathbf{A}^*\| + \|\mathbf{A}^*\| = \mathcal{O}(\sqrt{\tfrac{m}{n}}).$$

Therefore, we have

$$\|\Delta_j^{(t)}\| + o(\tfrac{k}{m}\epsilon_t) \pm \gamma = \mathcal{O}(\sqrt{m} q_{i,j} p_{i_1} \epsilon_t \|\mathbf{A}^{(t)}\|).$$

Here, we use the fact that $\gamma$ drops with decreasing $\epsilon_t$ as argued in Lemma 5. Next, using (23), we have

$$\|\widehat{\mathbf{g}}_j^{(t)}\| \leq 4\rho_-\|\mathbf{A}_j^{(t)} - \mathbf{A}_j^*\| + \|\widetilde{v}\|.$$

Now, letting

$$\|\Delta_j^{(t)}\| + o(\tfrac{k}{m}\epsilon_t) \pm \gamma = \mathcal{O}(\sqrt{m} q_{i,j} p_{i_1} \epsilon_t \|\mathbf{A}^{(t)}\|) \leq \tfrac{q_i p_i}{2}\|\mathbf{A}_j^{(t)} - \mathbf{A}_j^*\|, \tag{24}$$

we have that, for $k = \mathcal{O}(\sqrt{n})$

$$\|\widetilde{v}\| \leq q_i p_i \|\mathbf{A}_j^{(t)} - \mathbf{A}_j^*\|.$$

Substituting for $\|\widetilde{v}\|$, this implies that $\|\widehat{\mathbf{g}^{(t)}}_j\|^2 \leq 25\rho_-^2\|\mathbf{A}_j^{(t)} - \mathbf{A}_j^*\|^2$. Further, we also have the following lower-bound

$$\langle \widehat{\mathbf{g}}_j^{(t)}, \mathbf{A}_j^{(t)} - \mathbf{A}_j^*\rangle \geq 4\rho_-\|\mathbf{A}_j^{(t)} - \mathbf{A}_j^*\|^2 - \|\widetilde{v}\|\|\mathbf{A}_j^{(t)} - \mathbf{A}_j^*\|.$$

Here, we use the fact that R.H.S. can be minimized only if $\widetilde{v}$ is directed opposite to the direction of $\mathbf{A}_j^{(t)} - \mathbf{A}_j^*$. Now, we show that this gradient is $(\rho_-, 1/100\rho_-, 0)$ correlated,

$$\langle \widehat{\mathbf{g}}_i^{(t)}, \mathbf{A}_i^{(t)} - \mathbf{A}_i^* \rangle - \rho_- \|\mathbf{A}_i^{(t)} - \mathbf{A}_i^*\|^2 - \frac{1}{100\rho_-} \|\widehat{\mathbf{g}}_i^{(t)}\|^2,$$

$$\geq 4\rho_- \|\mathbf{A}_i^{(t)} - \mathbf{A}_i^*\|^2 - \|\widetilde{v}\| \|\mathbf{A}_i^{(t)} - \mathbf{A}_i^*\| - \rho_- \|\mathbf{A}_i^{(t)} - \mathbf{A}_i^*\|^2 - \frac{1}{100\rho_-} \|\widehat{\mathbf{g}}_i^{(t)}\|^2,$$

$$\geq 4\rho_- \|\mathbf{A}_i^{(t)} - \mathbf{A}_i^*\|^2 - 2\rho_- \|\mathbf{A}_i^{(t)} - \mathbf{A}_i^*\|^2 - \frac{25\rho_-^2 \|\mathbf{A}_i^{(t)} - \mathbf{A}_i^*\|^2}{100\rho_-},$$

$$\geq \rho_- \|\mathbf{A}_i^{(t)} - \mathbf{A}_i^*\|^2 \geq 0.$$

Therefore, for this choice of $k$, i.e. $k = \mathcal{O}(\sqrt{n})$, there is no bias in dictionary estimation in comparison to Arora et al. (2015). This gain can be attributed to estimating the coefficients simultaneously with the dictionary. Further, since we choose $4\rho_- = p_j q_j$, we have that $\rho_- = \Theta(k/m)$, as a result $\rho_+ = 1/100\rho_- = \Omega(m/k)$. Applying Lemma 15 we have

$$\|\mathbf{A}_j^{(t+1)} - \mathbf{A}_j^*\|^2 \leq (1 - \rho_- \eta_A) \|\mathbf{A}_j^{(t)} - \mathbf{A}_j^*\|^2,$$

for $\eta_A = \mathcal{O}(m/k)$ with probability at least $(1 - \delta_{\mathcal{T}}^{(t)} - \delta_\beta^{(t)} - \delta_{\mathbf{g}_i}^{(t)})$. $\qquad \square$

*Proof of Lemma 8.* Here, we will prove that $\widehat{\mathbf{g}}^{(t)}$ defined as

$$\widehat{\mathbf{g}}^{(t)} = \sum_j (\mathbf{y}_{(j)} - \mathbf{A}^{(t)} \widehat{\mathbf{x}}_{(j)}) \text{sign}(\widehat{\mathbf{x}}_{(j)})^\top,$$

concentrates around its mean. Notice that each summand $(\mathbf{y}_{(j)} - \mathbf{A}^{(t)} \widehat{\mathbf{x}}_{(j)}) \text{sign}(\widehat{\mathbf{x}}_{(j)})^\top$ is a random matrix of the form $(\mathbf{y} - \mathbf{A}^{(t)} \widehat{\mathbf{x}}) \text{sign}(\widehat{\mathbf{x}})^\top$. Also, we have $\mathbf{g}^{(t)}$ defined as

$$\mathbf{g}^{(t)} = \mathbf{E}[(\mathbf{y} - \mathbf{A}^{(t)} \widehat{\mathbf{x}}) \text{sign}(\widehat{\mathbf{x}})^\top].$$

To bound $\|\widehat{\mathbf{g}}^{(t)} - \mathbf{g}^{(t)}\|$, we are interested in $\|\sum_{j=1}^p \mathbf{W}_j\|$, where each matrix $\mathbf{W}_j$ is given by

$$\mathbf{W}_j = \frac{1}{p} (\mathbf{y}_{(j)} - \mathbf{A}^{(t)} \widehat{\mathbf{x}}_{(j)}) \text{sign}(\widehat{\mathbf{x}}_{(j)})^\top - \frac{1}{p} \mathbf{E}[(\mathbf{y} - \mathbf{A}^{(t)} \widehat{\mathbf{x}}) \text{sign}(\widehat{\mathbf{x}})^\top].$$

Noting that $\mathbf{E}[\mathbf{W}_j] = 0$, we will employ the matrix Bernstein result (Lemma 10) to bound $\|\widehat{\mathbf{g}}^{(t)} - \mathbf{g}^{(t)}\|$. To this end, we will bound $\|\mathbf{W}_j\|$ and the variance proxy

$$v(\mathbf{W}_j) = \max\{\|\sum_{j=1}^p \mathbf{E}[\mathbf{W}_j \mathbf{W}_j^\top]\|, \|\sum_{j=1}^p \mathbf{E}[\mathbf{W}_j^\top \mathbf{W}_j]\|\}.$$

**Bound on $\|\mathbf{W}_j\|$**– First, we can bound both terms in the expression for $\mathbf{W}_j$ by triangle inequality as

$$\|\mathbf{W}_j\| \leq \frac{1}{p} \|(\mathbf{y}_{(j)} - \mathbf{A}^{(t)} \widehat{\mathbf{x}}_{(j)}) \text{sign}(\widehat{\mathbf{x}}_{(j)})^\top\| + \frac{1}{p} \|\mathbf{E}[(\mathbf{y} - \mathbf{A}^{(t)} \widehat{\mathbf{x}}) \text{sign}(\widehat{\mathbf{x}})^\top\|,$$

$$\leq \frac{2}{p} \|(\mathbf{y} - \mathbf{A}^{(t)} \widehat{\mathbf{x}}) \text{sign}(\widehat{\mathbf{x}})^\top\|.$$

Here, we use Jensen's inequality for the second term, followed by upper-bounding the expected value of the argument by $\|(\mathbf{y} - \mathbf{A}^{(t)} \widehat{\mathbf{x}}) \text{sign}(\widehat{\mathbf{x}})^\top\|$.

Next, using Claim 8 we have that with probability at least $(1 - \delta_\beta^{(t)} - \delta_{\mathcal{T}}^{(t)} - \delta_{\text{HW}}^{(t)})$, $\|\mathbf{y} - \mathbf{A}^{(t)} \widehat{\mathbf{x}}\|$ is $\widetilde{\mathcal{O}}(k t_\beta)$, and the fact that $\|\text{sign}(x)^T\| = \sqrt{k}$,

$$\|\mathbf{W}_j\| \leq \frac{2}{p} \sqrt{k} \|(\mathbf{y} - \mathbf{A}^{(t)} \widehat{\mathbf{x}})\| = \mathcal{O}(\frac{k\sqrt{k}}{p} t_\beta).$$

**Bound on the variance statistic $v(\mathbf{W}_j)$**– For the variance statistic, we first look at $\|\sum \mathbf{E}[\mathbf{W}_j \mathbf{W}_j^\top]\|$,

$$\mathbf{E}[\mathbf{W}_j \mathbf{W}_j^\top] = \frac{1}{p^2} \mathbf{E}[(\mathbf{y}_{(j)} - \mathbf{A}^{(t)} \widehat{\mathbf{x}}_{(j)}) \text{sign}(\widehat{\mathbf{x}}_{(j)})^\top - \mathbf{E}[(\mathbf{y} - \mathbf{A}^{(t)} \widehat{\mathbf{x}}) \text{sign}(\widehat{\mathbf{x}})^\top]$$
$$\times [\text{sign}(\widehat{\mathbf{x}}_{(j)}) (\mathbf{y}_{(j)} - \mathbf{A}^{(t)} \widehat{\mathbf{x}}_{(j)})^\top - (\mathbf{E}[(\mathbf{y} - \mathbf{A}^{(t)} \widehat{\mathbf{x}}) \text{sign}(\widehat{\mathbf{x}})^\top)^\top].$$

Since $\mathbf{E}[(\mathbf{y} - \mathbf{A}^{(t)} \widehat{\mathbf{x}}) \text{sign}(\widehat{\mathbf{x}})^\top] \mathbf{E}[(\mathbf{y} - \mathbf{A}^{(t)} \widehat{\mathbf{x}}) \text{sign}(\widehat{\mathbf{x}})^\top]^\top$ is positive semidefinite,

$$\mathbf{E}[\mathbf{W}_j \mathbf{W}_j^\top] \preceq \frac{1}{p^2} \mathbf{E}[(\mathbf{y}_{(j)} - \mathbf{A}^{(t)} \widehat{\mathbf{x}}_{(j)}) \text{sign}(\widehat{\mathbf{x}}_{(j)})^\top \text{sign}(\widehat{\mathbf{x}}_{(j)}) (\mathbf{y}_{(j)} - \mathbf{A}^{(t)} \widehat{\mathbf{x}}_{(j)})^\top].$$

Now, since each $\widehat{\mathbf{x}}_{(j)}$ has $k$ non-zeros, $\text{sign}(\widehat{\mathbf{x}}_{(j)})^\top \text{sign}(\widehat{\mathbf{x}}_{(j)}) = k$, and using Claim 10, with probability at least $(1 - \delta_{\mathcal{T}}^{(t)} - \delta_{\beta}^{(t)})$

$$\| \sum \mathbf{E}[\mathbf{W}_j \mathbf{W}_j^\top] \| \leq \frac{k}{p} \| \mathbf{E}[(\mathbf{y}_{(j)} - \mathbf{A}^{(t)}\widehat{\mathbf{x}}_{(j)})(\mathbf{y}_{(j)} - \mathbf{A}^{(t)}\widehat{\mathbf{x}}_{(j)})^\top] \|,$$
$$= \mathcal{O}(\frac{k^3 t_\beta^2}{pm}) \| \mathbf{A}^* \|^2.$$

Similarly, expanding $\mathbf{E}[\mathbf{W}_j^\top \mathbf{W}_j]$, and using the fact that $\mathbf{E}[(\mathbf{y} - \mathbf{A}^{(t)}\widehat{\mathbf{x}})\text{sign}(\widehat{\mathbf{x}})^\top]^\top \mathbf{E}[(\mathbf{y} - \mathbf{A}^{(t)}\widehat{\mathbf{x}})\text{sign}(\widehat{\mathbf{x}})^\top]$ is positive semi-definite. Now, using Claim 8 and the fact that entries of $\mathbf{E}[(\text{sign}(\widehat{\mathbf{x}}_{(j)})\text{sign}(\widehat{\mathbf{x}}_{(j)})^\top]$ are $q_i$ on the diagonal and zero elsewhere, where $q_i = \mathcal{O}(k/m)$,

$$\| \sum \mathbf{E}[\mathbf{W}_j^\top \mathbf{W}_j] \| \preceq \frac{1}{p} \| \mathbf{E}[(\text{sign}(\widehat{\mathbf{x}}_{(j)})(\mathbf{y}_{(j)} - \mathbf{A}^{(t)}\widehat{\mathbf{x}}_{(j)})^\top(\mathbf{y}_{(j)} - \mathbf{A}^{(t)}\widehat{\mathbf{x}}_{(j)})\text{sign}(x_{(j)}^{(R)})^\top] \|,$$
$$\leq \frac{1}{p} \| \mathbf{E}[(\text{sign}(\widehat{\mathbf{x}}_{(j)})\text{sign}(\widehat{\mathbf{x}}_{(j)})^\top] \| \| \mathbf{y}_{(j)} - \mathbf{A}^{(t)}\widehat{\mathbf{x}}_{(j)} \|^2,$$
$$\leq \mathcal{O}(\frac{k}{mp})\widetilde{\mathcal{O}}(k^2 t_\beta^2) = \widetilde{\mathcal{O}}(\frac{k^3 t_\beta^2}{mp}).$$

Now, we are ready to apply the matrix Bernstein result. Since, $m = O(n)$ the variance statistic comes out to be $\mathcal{O}(\frac{k^3 t_\beta^2}{pm}) \| \mathbf{A}^* \|^2$, then as long as we choose $p = \widetilde{\Omega}(mk^2)$ (using the bound on $t_\beta$), with probability at least $(1 - \delta_\beta^{(t)} - \delta_{\mathcal{T}}^{(t)} - \delta_{\text{HW}}^{(t)} - \delta_{\mathbf{g}}^{(t)})$

$$\| \widehat{\mathbf{g}}^{(t)} - \mathbf{g}^{(t)} \| \leq \mathcal{O}(\frac{k\sqrt{k}}{p} t_\beta) + \| \mathbf{A}^* \| \sqrt{\mathcal{O}(\frac{k^3 t_\beta^2}{pm})},$$
$$= \mathcal{O}^*(\frac{k}{m} \| \mathbf{A}^* \|).$$

where $\delta_{\mathbf{g}}^{(t)} = (n + m)\exp(-\Omega(m\sqrt{\log(n)}))$. □

*Proof of Lemma 9.* This lemma ensures that the dictionary iterates maintain the closeness property (Def.1) and satisfies the prerequisites for Lemma 7.

The update step for the $i$-th dictionary element at the $s + 1$ iteration can be written as

$$\mathbf{A}_i^{(t+1)} - \mathbf{A}_i^* = \mathbf{A}_i^{(t)} - \mathbf{A}_i^* - \eta_A \widehat{\mathbf{g}}_i^{(t)},$$
$$= \mathbf{A}_i^{(t)} - \mathbf{A}_i^* - \eta_A \mathbf{g}_i^{(t)} - \eta_A(\widehat{\mathbf{g}}_i^{(t)} - \mathbf{g}_i^{(t)}).$$

Here, $\mathbf{g}_i^{(t)}$ is given by the following as per Lemma 5 with probability at least $(1 - \delta_{\mathcal{T}}^{(t)} - \delta_\beta^{(t)})$

$$\mathbf{g}_i^{(t)} = q_i p_i(\mathbf{A}_i^{(t)} - \mathbf{A}_i^*) + q_i p_i(-\lambda_i^{(t)}\mathbf{A}_i^{(t)} + \frac{1}{q_i p_i}\Delta_i^{(t)} \pm \gamma).$$

Substituting the expression for $\mathbf{g}_i^{(t)}$ in the dictionary update step,

$$\mathbf{A}_i^{(t+1)} - \mathbf{A}_i^* = (1 - \eta_A p_i q_i)(\mathbf{A}_i^{(t)} - \mathbf{A}_i^*) - \eta_A p_i q_i \lambda_i^{(t)}\mathbf{A}_i^{(t)} - \eta_A \Delta_i^{(t)} - \eta_A(\widehat{\mathbf{g}}_i^{(t)} - \mathbf{g}_i^{(t)}) \pm \gamma,$$

where $\Delta_j^{(t)} = \mathbf{E}[\mathbf{A}^{(t)}\vartheta^{(R)}\text{sign}(\mathbf{x}_j^*)]_j$. Therefore, the update step for the dictionary (matrix) can be written as

$$\mathbf{A}^{(t+1)} - \mathbf{A}^* = (\mathbf{A}^{(t)} - \mathbf{A}^*)\text{diag}((1 - \eta_A p_i q_i)) + \eta_A \mathbf{U} - \eta_A \mathbf{V} \pm \gamma - \eta_A(\widehat{\mathbf{g}}^{(t)} - \mathbf{g}^{(t)}), \quad (25)$$

where, $\mathbf{U} = \mathbf{A}^{(t)}\text{diag}(p_i q_i \lambda_i^{(t)})$ and $\mathbf{V} = \mathbf{A}^{(t)}\mathbf{Q}$, with the matrix $\mathbf{Q}$ given by,

$$\mathbf{Q}_{i,j} = q_{i,j}\mathbf{E}_{\mathbf{x}_S^*}[\vartheta_i^{(R)}\text{sign}(\mathbf{x}_j^*)|S],$$

and using the following intermediate result shown in Claim 7,

$$\mathbf{E}_{\mathbf{x}_S^*}[\vartheta_i^{(R)}\text{sign}(\mathbf{x}_j^*)|S] \begin{cases} \leq \gamma_i^{(R)}, & \text{for } i = j, \\ = \mathcal{O}(p_j \epsilon_t), & \text{for } i \neq j, \end{cases}$$

we have $\| \mathbf{Q}_i \| = \mathcal{O}(\sqrt{m}q_{i,j}p_i \epsilon_t)$. Hence, we have

$$\| \mathbf{Q} \|_F \leq \mathcal{O}(mq_{i,j}p_i \epsilon_t).$$

Therefore,

$$\|\mathbf{V}\| \le \|\mathbf{A}^{(t)}\mathbf{Q}\| \le \|\mathbf{A}^{(t)}\|\|\mathbf{Q}\|_F = \mathcal{O}(mq_{i,j}p_i\epsilon_t\|\mathbf{A}^*\|) = \mathcal{O}(\tfrac{k^2}{m\log(n)})\|\mathbf{A}^*\|.$$

We will now proceed to bound each term in (25). Starting with $(\mathbf{A}^{(t)} - \mathbf{A}^*)\mathrm{diag}(1 - \eta_A p_i q_i)$, and using the fact that $p_i = O(1)$, $q_i = O(k/m)$, and $\|\mathbf{A}^{(t)} - \mathbf{A}^*\| \le 2\|\mathbf{A}^*\|$, we have

$$\|(\mathbf{A}^{(t)} - \mathbf{A}^*)\mathrm{diag}(1 - \eta_A p_i q_i)\| \le (1 - \min_i \eta_A p_i q_i)\|(\mathbf{A}^{(t)} - \mathbf{A}^*)\| \le 2(1 - \Omega(\eta_A k/m))\|\mathbf{A}^*\|.$$

Next, since $\|\mathbf{A}_j^{(t)}\| = 1$, we have $\lambda_j^{(t)} = |\langle \mathbf{A}_j^{(t)} - \mathbf{A}_j^*, \mathbf{A}_j^*\rangle| = \tfrac{1}{2}\|\mathbf{A}_j^{(t)} - \mathbf{A}_j^*\|^2$, and $\lambda_i^{(t)} \le \epsilon_t^2/2$, therefore

$$\|\mathbf{U}\| = \|\mathbf{A}^{(t)}\mathrm{diag}(p_i q_i \lambda_i^{(t)})\| \le \max_i p_i q_i \tfrac{\epsilon_t^2}{2}\|\mathbf{A}^{(t)} - \mathbf{A}^* + \mathbf{A}^*\| \le o(k/m)\|\mathbf{A}^*\|.$$

Using the results derived above, and the the result derived in Lemma 8 which states that with probability at least $(1 - \delta_\beta^{(t)} - \delta_{\mathcal{T}}^{(t)} - \delta_{\mathrm{HW}}^{(t)} - \delta_{\mathbf{g}}^{(t)})$, $\|\widehat{\mathbf{g}}^{(t)} - \mathbf{g}^{(t)}\| = \mathcal{O}^*(\tfrac{k}{m}\|\mathbf{A}^*\|))$ we have

$$\begin{aligned}
\|\mathbf{A}^{(t+1)} - \mathbf{A}^*\| &= \|(\mathbf{A}^{(t)} - \mathbf{A}^*)\mathbf{D}_{(1-\eta_A p_i q_i)}\| + \eta_A\|\mathbf{U}\| + \eta_A\|\mathbf{V}\| + \eta_A\|(\widehat{\mathbf{g}}^{(t)} - \mathbf{g}^{(t)})\| \pm \gamma, \\
&\le 2(1 - \Omega(\eta_A \tfrac{k}{m})\|\mathbf{A}^*\| + o(\eta_A \tfrac{k}{m})\|\mathbf{A}^*\| + \mathcal{O}(\eta_A \tfrac{k^2}{m\log(n)})\|\mathbf{A}^*\| + o(\eta_A \tfrac{k}{m}\|\mathbf{A}^*\|) \pm \gamma, \\
&\le 2\|\mathbf{A}^*\|.
\end{aligned}$$

$\square$

## D  APPENDIX: PROOFS OF INTERMEDIATE RESULTS

**Claim 1** (**Incoherence of $\mathbf{A}^{(t)}$**). *If $\mathbf{A}^* \in \mathbb{R}^{n \times m}$ is $\mu$-incoherent and $\|\mathbf{A}_i^* - \mathbf{A}_i^{(t)}\| \leq \epsilon_t$ holds for each $i \in [1 \ldots m]$, then $\mathbf{A}^{(t)} \in \mathbb{R}^{n \times m}$ is $\mu_t$-incoherent, where $\mu_t = \mu + 2\sqrt{n}\epsilon_t$.*

*Proof of Claim 1.* We start by looking at the incoherence between the columns of $\mathbf{A}^*$, for $j \neq i$,

$$\langle \mathbf{A}_i^*, \mathbf{A}_j^* \rangle = \langle \mathbf{A}_i^* - \mathbf{A}_i^{(t)}, \mathbf{A}_j^* \rangle + \langle \mathbf{A}_i^{(t)}, \mathbf{A}_j^* \rangle,$$
$$= \langle \mathbf{A}_i^* - \mathbf{A}_i^{(t)}, \mathbf{A}_j^* \rangle + \langle \mathbf{A}_i^{(t)}, \mathbf{A}_j^* - \mathbf{A}_j^{(t)} \rangle + \langle \mathbf{A}_i^{(t)}, \mathbf{A}_j^{(t)} \rangle.$$

Since $\langle \mathbf{A}_i^*, \mathbf{A}_j^* \rangle \leq \frac{\mu}{\sqrt{n}}$,

$$|\langle \mathbf{A}_i^{(t)}, \mathbf{A}_j^{(t)} \rangle| \leq \langle \mathbf{A}_i^*, \mathbf{A}_j^* \rangle - \langle \mathbf{A}_i^* - \mathbf{A}_i^{(t)}, \mathbf{A}_j^* \rangle - \langle \mathbf{A}_i^{(t)}, \mathbf{A}_j^* - \mathbf{A}_j^{(t)} \rangle,$$
$$\leq \frac{\mu}{\sqrt{n}} + 2\epsilon_t.$$

$\square$

**Claim 2** (**Bound on $\beta_j^{(t)}$: the noise component in coefficient estimate that depends on $\epsilon_t$**). *With probability $(1 - \delta_\beta^{(t)})$, $|\beta_j^{(t)}|$ is upper-bounded by $t_\beta = \mathcal{O}(\sqrt{k}\epsilon_t)$, where $\delta_\beta^{(t)} = 2k \exp(-\frac{1}{\mathcal{O}(\epsilon_t)})$.*

*Proof of Claim 2.* We have the following definition for $\beta_j^{(t)}$ from (11),

$$\beta_j^{(t)} = \sum_{i \neq j} (\langle \mathbf{A}_j^*, \mathbf{A}_i^* - \mathbf{A}_i^{(t)} \rangle + \langle \mathbf{A}_j^* - \mathbf{A}_j^{(t)}, \mathbf{A}_i^{(t)} \rangle + \langle \mathbf{A}_j^{(t)} - \mathbf{A}_j^*, \mathbf{A}_i^* \rangle) \mathbf{x}_i^*.$$

Here, since $\mathbf{x}_i^*$ are independent sub-Gaussian random variables, $\beta_j^{(t)}$ is a sub-Gaussian random variable with the variance parameter evaluated as shown below

$$var[\beta_j^{(t)}] = \sum_{i \neq j} (\langle \mathbf{A}_j^*, \mathbf{A}_i^{(t)} - \mathbf{A}_i^* \rangle + \langle \mathbf{A}_j^{(t)} - \mathbf{A}_j^*, \mathbf{A}_i^{(t)} \rangle + \langle \mathbf{A}_j^{(t)} - \mathbf{A}_j^*, \mathbf{A}_i^* \rangle)^2 \leq 9k\epsilon_t^2.$$

Therefore, by Lemma 12

$$\mathbf{Pr}[|\beta_j^{(t)}| > t_\beta] \leq 2\exp(-\frac{t_\beta^2}{18k\epsilon_t^2}).$$

Now, we need this for each $\beta_j^{(t)}$ for $j \in \mathrm{supp}(\mathbf{x}^*)$, union bounding over $k$ coefficients

$$\mathbf{Pr}[\max |\beta_j^{(t)}| > t_\beta] \leq \delta_\beta^{(t)},$$

where $\delta_\beta^{(t)} = 2k \exp(-\frac{t_\beta^2}{18k\epsilon_t^2})$. Choosing $t_\beta = \mathcal{O}(\sqrt{k}\epsilon_t)$, we have that $\delta_\beta^{(t)} = 2k \exp(-\frac{1}{\mathcal{O}(\epsilon_t)})$. $\square$

**Claim 3** (**Error in coefficient estimation for a general iterate $(r + 1)$**). *The error in a general iterate $r$ of the coefficient estimation is upper-bounded as*

$$C_{i_1}^{(r+1)} \leq \alpha_{i_1}^{(r+1)} + (k-1)\eta_x \frac{\mu_t}{\sqrt{n}} \sum_{\ell=1}^{r} \alpha_{\max}^{(\ell)} \big(1 - \eta_x + \eta_x \frac{\mu_t}{\sqrt{n}}\big)^{r-\ell}$$
$$+ (k-1)\eta_x \frac{\mu_t}{\sqrt{n}} C_{\max}^{(0)} \big(1 - \eta_x + \eta_x \frac{\mu_t}{\sqrt{n}}\big)^r.$$

*Proof of Claim 3.* From (17) we have the following expression for $C_{i_1}^{(r+1)}$

$$C_{i_1}^{(r+1)} \leq \alpha_{i_1}^{(r+1)} + \frac{\mu_t}{\sqrt{n}} \sum_{\ell=1}^{r+1} \eta_x^{(\ell)} \sum_{i_2 \neq i_1} C_{i_2}^{(\ell-1)} \prod_{q=\ell}^{r+1} (1 - \eta_x^{(q+1)}).$$

Our aim will be to recursively substitute for $C_{i_1}^{(\ell-1)}$ to develop an expression for $C_{i_1}^{(r+1)}$ as a function of $C_{\max}^0$. To this end, we start by analyzing the iterates $C_{i_1}^{(1)}$, $C_{i_1}^{(2)}$, and so on to develop an expression for $C_{i_1}^{(r+1)}$ as follows.

**Expression for $C_{i_1}^{(1)}$** – Consider $C_{i_1}^{(1)}$

$$C_{i_1}^{(1)} \leq \alpha_{i_1}^{(1)} + \tfrac{\mu_t}{\sqrt{n}} \sum_{\ell=1}^{1} \eta_x \sum_{i_2 \neq i_1} C_{i_2}^{(\ell-1)} \prod_{q=\ell}^{1} (1 - \eta_x),$$
$$= \alpha_{i_1}^{(1)} + \eta_x \big( \tfrac{\mu_t}{\sqrt{n}} \sum_{i_1 \neq i_2} C_{i_2}^{(0)} \big). \tag{26}$$

**Expression for $C_{i_1}^{(2)}$** – Next, $C_{i_1}^{(2)}$ is given by

$$C_{i_1}^{(2)} \leq \alpha_{i_1}^{(2)} + \eta_x \tfrac{\mu_t}{\sqrt{n}} \sum_{\ell=1}^{2} \sum_{i_2 \neq i_1} C_{i_2}^{(\ell-1)} \prod_{q=\ell}^{2} (1 - \eta_x),$$
$$\leq \alpha_{i_1}^{(2)} + \eta_x \tfrac{\mu_t}{\sqrt{n}} \big( \sum_{i_2 \neq i_1} C_{i_2}^{(1)} + \sum_{i_2 \neq i_1} C_{i_2}^{(0)} (1 - \eta_x) \big).$$

Further, we know from (26) we have

$$C_{i_2}^{(1)} = \alpha_{i_2}^{(1)} + \eta_x \tfrac{\mu_t}{\sqrt{n}} \sum_{i_3 \neq i_2} C_{i_3}^{(0)}.$$

Therefore, since $\sum_{i_2 \neq i_1} \sum_{i_3 \neq i_2} = \sum_{i_3 \neq i_2, i_1}$,

$$C_{i_1}^{(2)} \leq \alpha_{i_1}^{(2)} + \eta_x \tfrac{\mu_t}{\sqrt{n}} \big( \sum_{i_2 \neq i_1} \big( \alpha_{i_2}^{(1)} + \eta_x \tfrac{\mu_t}{\sqrt{n}} \sum_{i_3 \neq i_2} C_{i_3}^{(0)} \big) + \sum_{i_2 \neq i_1} C_{i_2}^{(0)} (1 - \eta_x) \big),$$
$$= \alpha_{i_1}^{(2)} + \eta_x \tfrac{\mu_t}{\sqrt{n}} \sum_{i_2 \neq i_1} \alpha_{i_2}^{(1)} + \eta_x \tfrac{\mu_t}{\sqrt{n}} \big( \eta_x \tfrac{\mu_t}{\sqrt{n}} \sum_{i_3 \neq i_2, i_1} C_{i_3}^{(0)} + \sum_{i_2 \neq i_1} C_{i_2}^{(0)} (1 - \eta_x) \big). \tag{27}$$

**Expression for $C_{i_1}^{(3)}$** – Next, we writing $C_{i_1}^{(3)}$,

$$C_{i_1}^{(3)} \leq \alpha_{i_1}^{(3)} + \eta_x \tfrac{\mu_t}{\sqrt{n}} \sum_{\ell=1}^{3} \sum_{i_2 \neq i_1} C_{i_2}^{(\ell-1)} (1 - \eta_x)^{3-\ell},$$
$$= \alpha_{i_1}^{(3)} + \eta_x \tfrac{\mu_t}{\sqrt{n}} \sum_{i_2 \neq i_1} \big( C_{i_2}^{(0)} (1 - \eta_x)^2 + C_{i_2}^{(1)} (1 - \eta_x) + C_{i_2}^{(2)} \big),$$
$$\leq \alpha_{i_1}^{(3)} + \eta_x \tfrac{\mu_t}{\sqrt{n}} \sum_{i_2 \neq i_1} \big( C_{i_2}^{(0)} (1 - \eta_x)^2 + \big( \alpha_{i_2}^{(1)} + \eta_x \tfrac{\mu_t}{\sqrt{n}} \sum_{i_3 \neq i_2} C_{i_3}^{(0)} \big) (1 - \eta_x) + C_{i_2}^{(2)} \big).$$

Here, using (27) we have the following expression for $C_{i_2}^{(2)}$

$$C_{i_2}^{(2)} \leq \alpha_{i_2}^{(2)} + \eta_x \tfrac{\mu_t}{\sqrt{n}} \sum_{i_3 \neq i_2} \alpha_{i_3}^{(1)} + \eta_x \tfrac{\mu_t}{\sqrt{n}} \big( \eta_x \tfrac{\mu_t}{\sqrt{n}} \sum_{i_4 \neq i_3, i_2} C_{i_4}^{(0)} + \sum_{i_3 \neq i_2} C_{i_3}^{(0)} (1 - \eta_x) \big).$$

Substituting for $C_{i_2}^{(2)}$ in the expression for $C_{i_1}^{(3)}$, and rearranging the terms in the expression for $C_{i_1}^{(3)}$, we have

$$C_{i_1}^{(3)} \leq \alpha_{i_1}^{(3)} + \eta_x \tfrac{\mu_t}{\sqrt{n}} \sum_{i_2 \neq i_1} \alpha_{i_2}^{(2)} + \eta_x \tfrac{\mu_t}{\sqrt{n}} \big( (1 - \eta_x) \sum_{i_2 \neq i_1} \alpha_{i_2}^{(1)} + \eta_x \tfrac{\mu_t}{\sqrt{n}} \sum_{i_3 \neq i_2, i_1} \alpha_{i_3}^{(1)} \big)$$
$$+ \eta_x \tfrac{\mu_t}{\sqrt{n}} \big( (1 - \eta_x)^2 \sum_{i_2 \neq i_1} C_{i_2}^{(0)} + 2(1 - \eta_x)(\eta_x \tfrac{\mu_t}{\sqrt{n}}) \sum_{i_3 \neq i_2, i_1} C_{i_3}^{(0)} + (\eta_x \tfrac{\mu_t}{\sqrt{n}})^2 \sum_{i_4 \neq i_3, i_2, i_1} C_{i_4}^{(0)} \big). \tag{28}$$

**Expression for $C_{i_1}^{(4)}$** – Now, consider $C_{i_1}^{(4)}$

$$C_{i_1}^{(4)} \leq \alpha_{i_1}^{(4)} + \eta_x \tfrac{\mu_t}{\sqrt{n}} \sum_{\ell=1}^{4} \sum_{i_2 \neq i_1} C_{i_2}^{(\ell-1)} (1 - \eta_x)^{4-\ell},$$
$$\leq \alpha_{i_1}^{(4)} + \eta_x \tfrac{\mu_t}{\sqrt{n}} \big( \sum_{i_2 \neq i_1} C_{i_2}^{(0)} (1 - \eta_x)^3 + \sum_{i_2 \neq i_1} C_{i_2}^{(1)} (1 - \eta_x)^2 + \sum_{i_2 \neq i_1} C_{i_2}^{(2)} (1 - \eta_x)^1$$
$$+ \sum_{i_2 \neq i_1} C_{i_2}^{(3)} (1 - \eta_x)^0 \big).$$

Substituting for $C_{i_2}^{(3)}$ from (28), $C_{i_2}^{(2)}$ from (27), $C_{i_2}^{(1)}$ using (26), and rearranging,

$$C_{i_1}^{(4)} \leq \alpha_{i_1}^{(4)} + \eta_x \tfrac{\mu_t}{\sqrt{n}} \bigg[ \sum_{i_2 \neq i_1} \alpha_{i_2}^{(3)} + \bigg( (1 - \eta_x)^1 \sum_{i_2 \neq i_1} \alpha_{i_2}^{(2)} + \eta_x \tfrac{\mu_t}{\sqrt{n}} \sum_{i_3 \neq i_2, i_1} \alpha_{i_3}^{(2)} \bigg)$$

$$+ \Big( \sum_{i_2 \neq i_1} \alpha_{i_2}^{(1)}(1-\eta_x)^2 + 2\eta_x \tfrac{\mu_t}{\sqrt{n}}(1-\eta_x) \sum_{i_3 \neq i_2, i_1} \alpha_{i_3}^{(1)} + (\eta_x \tfrac{\mu_t}{\sqrt{n}})^2 \sum_{i_4 \neq i_3, i_2, i_1} \alpha_{i_4}^{(1)} \Big) \Big]$$

$$+ \eta_x \tfrac{\mu_t}{\sqrt{n}} \Big[ \sum_{i_2 \neq i_1} C_{i_2}^{(0)}(1-\eta_x)^3 + 3\eta_x \tfrac{\mu_t}{\sqrt{n}}(1-\eta_x)^2 \sum_{i_3 \neq i_2, i_1} C_{i_3}^{(0)}$$

$$+ 3(\eta_x \tfrac{\mu_t}{\sqrt{n}})^2 (1-\eta_x)^1 \sum_{i_4 \neq i_3, i_2, i_1} C_{i_4}^{(0)} + (\eta_x \tfrac{\mu_t}{\sqrt{n}})^3 \sum_{i_5 \neq i_4, i_3, i_2, i_1} C_{i_5}^{(0)} \Big].$$

Notice that the terms have a binomial series like form. To reveal this structure, let each $\alpha_j^{(\ell)} \leq \alpha_{\max}^{(\ell)}$ where $j = i_1, i_2, \ldots, i_k$. Similarly, let $C_j^{(0)} \leq C_{\max}^{(0)}$ for $j = i_1, i_2, \ldots, i_k$. Therefore, we have

$$C_{i_1}^{(4)} \leq \alpha_{i_1}^{(4)} + \eta_x \tfrac{\mu_t}{\sqrt{n}} \Big[ (k-1)\alpha_i^{(3)} + \alpha_i^{(2)} \Big( (1-\eta_x)^1(k-1) + \eta_x \tfrac{\mu_t}{\sqrt{n}}(k-2) \Big)$$

$$+ \alpha_i^{(1)} \Big( (k-1)(1-\eta_x)^2 + 2(k-2)\eta_x \tfrac{\mu_t}{\sqrt{n}}(1-\eta_x) + (k-3)(\eta_x \tfrac{\mu_t}{\sqrt{n}})^2 \Big) \Big]$$

$$+ \eta_x \tfrac{\mu_t}{\sqrt{n}} C_i^{(0)} \Big[ (k-1)(1-\eta_x)^3 + 3(k-2)\eta_x \tfrac{\mu_t}{\sqrt{n}}(1-\eta_x)^2$$

$$+ 3(k-3)(\eta_x \tfrac{\mu_t}{\sqrt{n}})^2(1-\eta_x)^1 + (k-4)(\eta_x \tfrac{\mu_t}{\sqrt{n}})^3 \Big].$$

Further upper-bounding the expression, we have

$$C_{i_1}^{(4)} \leq \alpha_{i_1}^{(4)} + (k-1)\eta_x \tfrac{\mu_t}{\sqrt{n}} \Big[ \alpha_i^{(3)} + \alpha_i^{(2)} \Big( (1-\eta_x) + \eta_x \tfrac{\mu_t}{\sqrt{n}} \Big)$$

$$+ \alpha_i^{(1)} \Big( (1-\eta_x)^2 + 2\eta_x \tfrac{\mu_t}{\sqrt{n}}(1-\eta_x) + (\eta_x \tfrac{\mu_t}{\sqrt{n}})^2 \Big) \Big]$$

$$+ (k-1)\eta_x \tfrac{\mu_t}{\sqrt{n}} C_i^{(0)} \Big[ (1-\eta_x)^3 + 3\eta_x \tfrac{\mu_t}{\sqrt{n}}(1-\eta_x)^2 + 3(\eta_x \tfrac{\mu_t}{\sqrt{n}})^2(1-\eta_x) + (\eta_x \tfrac{\mu_t}{\sqrt{n}})^3 \Big].$$

Therefore,

$$C_{i_1}^{(4)} \leq \alpha_{i_1}^{(4)} + (k-1)\eta_x \tfrac{\mu_t}{\sqrt{n}} \Big[ \alpha_i^{(3)} + \alpha_i^{(2)} \big( 1-\eta_x + \eta_x \tfrac{\mu_t}{\sqrt{n}} \big)^1 + \alpha_i^{(1)} \big( 1-\eta_x + \eta_x \tfrac{\mu_t}{\sqrt{n}} \big)^2 \Big]$$

$$+ (k-1)\eta_x \tfrac{\mu_t}{\sqrt{n}} C_i^{(0)} \big( 1-\eta_x + \eta_x \tfrac{\mu_t}{\sqrt{n}} \big)^3. \quad (29)$$

**Expression for $C_{i_1}^{(r+1)}$** – With this, we are ready to write the general term,

$$C_{i_1}^{(r+1)} \leq \alpha_{i_1}^{(r+1)} + (k-1)\eta_x \tfrac{\mu_t}{\sqrt{n}} \sum_{\ell=1}^{r} \alpha_{\max}^{(\ell)} \big( 1-\eta_x + \eta_x \tfrac{\mu_t}{\sqrt{n}} \big)^{r-\ell}$$

$$+ (k-1)\eta_x \tfrac{\mu_t}{\sqrt{n}} C_{\max}^{(0)} \big( 1-\eta_x + \eta_x \tfrac{\mu_t}{\sqrt{n}} \big)^r.$$

$\square$

**Claim 4 (An intermediate result for bounding the error in coefficient calculations).** *With probability $(1 - \delta_{\mathcal{T}}^{(t)} - \delta_{\beta}^{(t)})$,*

$$\sum_{\ell=1}^{R} \alpha_{\max}^{(\ell)} \big( 1-\eta_x + \eta_x \tfrac{\mu_t}{\sqrt{n}} \big)^{R-\ell} \leq C_i^{(0)} R \delta_R + \tfrac{1}{\eta_x(1-\tfrac{\mu_t}{\sqrt{n}})} \big( \tfrac{\epsilon_t^2}{2} |\mathbf{x}_{\max}^*| + t_\beta \big).$$

*Proof of Claim 4.* Using (18), the quantity $\alpha_i^{(\ell)}$ is defined as

$$\alpha_i^{(\ell)} = C_i^{(0)}(1-\eta_x)^\ell + (\lambda_i^{(t)}|\mathbf{x}_i^*| + \beta_i^{(t)}) \sum_{s=1}^{\ell} \eta_x (1-\eta_x)^{\ell-s+1}.$$

Therefore, we are interested in

$$\sum_{\ell=1}^{R} C_i^{(0)} (1-\eta_x)^\ell \big(1 - \eta_x + \eta_x \tfrac{\mu_t}{\sqrt{n}}\big)^{R-\ell}$$

$$+ (\lambda_i^{(t)} |\mathbf{x}_i^*| + \beta_i^{(t)}) \sum_{\ell=1}^{R} \big(1 - \eta_x + \eta_x \tfrac{\mu_t}{\sqrt{n}}\big)^{R-\ell} \sum_{s=1}^{\ell} \eta_x (1-\eta_x)^{\ell-s+1}.$$

Consider the first term which depends on $C_i^{(0)}$. Since $(1-\eta_x) \leq (1 - \eta_x + \eta_x \tfrac{\mu_t}{\sqrt{n}})$ , we have

$$C_i^{(0)} \sum_{\ell=1}^{R} (1-\eta_x)^\ell \big(1 - \eta_x + \eta_x \tfrac{\mu_t}{\sqrt{n}}\big)^{R-\ell} \leq C_i^{(0)} R \big(1 - \eta_x + \eta_x \tfrac{\mu_t}{\sqrt{n}}\big)^{R} \leq C_i^{(0)} R \delta_R,$$

where $\delta_R$ is a small constant, and a parameter which determines the number of iterations $R$ required for the coefficient update step. Now, coming back to the quantity of interest

$$\sum_{\ell=1}^{R} \alpha_i^{(\ell)} \big(1 - \eta_x + \eta_x \tfrac{\mu_t}{\sqrt{n}}\big)^{R-\ell} \leq C_i^{(0)} R \delta_R$$

$$+ (\lambda_i^{(t)} |\mathbf{x}_i^*| + \beta_i^{(t)}) \sum_{\ell=1}^{R} \big(1 - \eta_x + \eta_x \tfrac{\mu_t}{\sqrt{n}}\big)^{R-\ell} \sum_{s=1}^{\ell} \eta_x (1-\eta_x)^{\ell-s+1}.$$

Now, using sum of geometric series result, we have that $\sum_{s=1}^{\ell} \eta_x (1-\eta_x)^{\ell-s+1}$, and

$$\sum_{\ell=1}^{R} \big(1 - \eta_x + \eta_x \tfrac{\mu_t}{\sqrt{n}}\big)^{R-\ell} = \frac{1-\big(1-\eta_x+\eta_x\frac{\mu_t}{\sqrt{n}}\big)^R}{\eta_x - \eta_x \frac{\mu_t}{\sqrt{n}}} \leq \frac{1}{\eta_x(1-\frac{\mu_t}{\sqrt{n}})}.$$

Therefore, with probability at least $(1 - \delta_\beta^{(t)})$,

$$\sum_{\ell=1}^{R} \alpha_{\max}^{(\ell)} \big(1 - \eta_x + \eta_x \tfrac{\mu_t}{\sqrt{n}}\big)^{R-\ell} \leq C_i^{(0)} R \delta_R + \frac{1}{\eta_x(1-\frac{\mu_t}{\sqrt{n}})} \big(\tfrac{\epsilon_t^2}{2} |\mathbf{x}_{\max}^*| + t_\beta\big),$$

where $\lambda_i^{(t)} \leq \tfrac{\epsilon_t^2}{2}$ and $|\beta_i^{(t)}| = t_\beta$ with probability at least $(1 - \delta_\beta^{(t)})$ using Claim 2. $\qquad \square$

**Claim 5 (Bound on the noise term in the estimation of a coefficient element in the support).** *With probability $(1 - \delta_\beta^{(t)})$, each entry $\vartheta_{i_1}^{(R)}$ of $\vartheta^{(R)}$ is upper-bounded as*

$$|\vartheta_{i_1}^{(R)}| \leq \mathcal{O}(t_\beta).$$

*Proof of Claim 5.* From (19) $\vartheta_{i_1}^{(R)}$ is defined as

$$\vartheta_{i_1}^{(R)} := \sum_{r=1}^{R} \eta_x \xi_{i_1}^{(r)} (1-\eta_x)^{R-r} + \gamma_{i_1}^{(R)},$$

where $\gamma_{i_1}^{(R)} := (1-\eta_x)^R (\mathbf{x}_{i_1}^{(0)} - \mathbf{x}_{i_1}^* (1-\lambda_{i_1}^{(t)}))$. Further, $\xi_{i_1}^{(r)}$ is as defined in (10),

$$\xi_{i_1}^{(r)} = \beta_{i_1}^{(t)} + \sum_{i_2 \neq i_1} |\langle \mathbf{A}_{i_1}^{(t)}, \mathbf{A}_{i_2}^{(t)} \rangle| \mathrm{sign}(\langle \mathbf{A}_{i_1}^{(t)}, \mathbf{A}_{i_2}^{(t)} \rangle) C_{i_2}^{(r-1)} \mathrm{sign}(\mathbf{x}_{i_2}^* - \mathbf{x}_{i_2}^{(r)}).$$

Therefore, we have the following expression for $\vartheta_{i_1}^{(R)}$

$$\vartheta_{i_1}^{(R)} = \beta_{i_1}^{(t)} \sum_{r=1}^{R} \eta_x (1-\eta_x)^{R-r}$$

$$+ \sum_{r=1}^{R} \eta_x \sum_{i_2 \neq i_1} |\langle \mathbf{A}_{i_1}^{(t)}, \mathbf{A}_{i_2}^{(t)} \rangle| \mathrm{sign}(\langle \mathbf{A}_{i_1}^{(t)}, \mathbf{A}_{i_2}^{(t)} \rangle) C_{i_2}^{(r-1)} \mathrm{sign}(\mathbf{x}_{i_2}^* - \mathbf{x}_{i_2}^{(r)})(1-\eta_x)^{R-r} + \gamma_{i_1}^{(R)}.$$

$$(30)$$

Now $\vartheta_{i_1}^{(R)}$ can be upper-bounded as

$$\vartheta_{i_1}^{(R)} \leq \beta_{i_1}^{(t)} \sum_{r=1}^{R} \eta_x (1-\eta_x)^{R-r} + \eta_x \frac{\mu_t}{\sqrt{n}} \sum_{r=1}^{R} \sum_{i_2 \neq i_1} C_{i_2}^{(r-1)} (1-\eta_x)^{R-r} + \gamma_{i_1}^{(R)},$$

$$\leq \beta_{i_1}^{(t)} + (k-1)\eta_x \frac{\mu_t}{\sqrt{n}} \sum_{r=1}^{R} C_{i_2}^{(r-1)} (1-\eta_x)^{R-r} + \gamma_{i_1}^{(R)}.$$

Since from Claim 6 we have

$$C_{i_2}^{(r-1)} (1-\eta_x)^{R-r} \leq (\lambda_{\max}^{(t)} |\mathbf{x}_{\max}^*| + \beta_{\max}^{(t)}) \Big[ \sum_{s=1}^{r-1} \eta_x (1-\eta_x)^{R-s} + k c_x (1-\eta_x)^{R-r} \Big]$$
$$+ k \eta_x \frac{\mu_t}{\sqrt{n}} C_{\max}^{(0)} \delta_{R-2}.$$

Further, since $1 - (1-\eta_x)^{r-1} \leq 1$, we have that

$$\sum_{r=1}^{R} \sum_{s=1}^{r-1} \eta_x (1-\eta_x)^{R-s} = \sum_{r=1}^{R} \eta_x (1-\eta_x)^{R-r+1} \frac{1-(1-\eta_x)^{r-1}}{\eta_x} \leq \sum_{r=1}^{R} (1-\eta_x)^{R-r+1} \leq \frac{1}{\eta_x}.$$

Therefore,

$$|\vartheta_{i_1}^{(R)}| \leq |\beta_{i_1}^{(t)}| + (k-1)\frac{\mu_t}{\sqrt{n}} (\lambda_{\max}^{(t)} |\mathbf{x}_{\max}^*| + |\beta_{\max}^{(t)}|)(1 + k c_x) + \big(k \eta_x \frac{\mu_t}{\sqrt{n}}\big)^2 R C_{\max}^{(0)} \delta_{R-2} + \gamma_{i_1}^{(R)}.$$

Now, since each $|\beta_i^{(t)}| = t_\beta$ with probability at least $(1 - \delta_\beta^{(t)})$ for the $t$-th iterate, and $k = \mathcal{O}^*(\frac{\sqrt{n}}{\mu \log(n)})$, therefore $k c_x < 1$, we have that

$$|\vartheta_{i_1}^{(R)}| \leq \mathcal{O}(t_\beta).$$

with probability at least $(1 - \delta_\beta^{(t)})$. $\qquad\qquad\qquad\qquad\qquad\qquad\qquad\qquad\qquad\qquad\qquad\square$

**Claim 6** (**An intermediate result for $\vartheta_{i_1}^{(R)}$ calculations**)**.** For $c_x = \frac{\mu_t}{\sqrt{n}} / (1 - \frac{\mu_t}{\sqrt{n}})$, we have

$$C_{i_2}^{(r-1)} (1-\eta_x)^{R-r} \leq (\lambda_{\max}^{(t)} |\mathbf{x}_{\max}^*| + \beta_{\max}^{(t)}) \left[ \sum_{s=1}^{r-1} \eta_x (1-\eta_x)^{R-s} + k c_x (1-\eta_x)^{R-r} \right]$$
$$+ k \eta_x \frac{\mu_t}{\sqrt{n}} C_{\max}^{(0)} \delta_{R-2}.$$

*Proof of Claim 6.* Here, from Claim 3 we have that for any $i_1$,

$$C_{i_1}^{(r+1)} \leq \alpha_{i_1}^{(r+1)} + k \eta_x \frac{\mu_t}{\sqrt{n}} \sum_{\ell=1}^{r} \alpha_{\max}^{(\ell)} \big(1 - \eta_x + \eta_x \frac{\mu_t}{\sqrt{n}}\big)^{r-\ell} + k \eta_x \frac{\mu_t}{\sqrt{n}} C_{\max}^{(0)} \big(1 - \eta_x + \eta_x \frac{\mu_t}{\sqrt{n}}\big)^{r}.$$

therefore $C_{i_2}^{(r-1)}$ is given by

$$C_{i_2}^{(r-1)} \leq \alpha_{i_2}^{(r-1)} + k \eta_x \frac{\mu_t}{\sqrt{n}} \sum_{\ell=1}^{r-2} \alpha_{\max}^{(\ell)} \big(1 - \eta_x + \eta_x \frac{\mu_t}{\sqrt{n}}\big)^{r-\ell-2} + k \eta_x \frac{\mu_t}{\sqrt{n}} C_{\max}^{(0)} \big(1 - \eta_x + \eta_x \frac{\mu_t}{\sqrt{n}}\big)^{r-2}.$$

Further, the term of interest $C_{i_2}^{(r-1)} (1-\eta_x)^{R-r}$ can be upper-bounded by

$$C_{i_2}^{(r-1)} (1-\eta_x)^{R-r} \leq \alpha_{i_2}^{(r-1)} (1-\eta_x)^{R-r} + (1-\eta_x)^{R-r} k \eta_x \frac{\mu_t}{\sqrt{n}} \sum_{\ell=1}^{r-2} \alpha_{\max}^{(\ell)} \big(1 - \eta_x + \eta_x \frac{\mu_t}{\sqrt{n}}\big)^{r-\ell-2}$$
$$+ k \eta_x \frac{\mu_t}{\sqrt{n}} C_{\max}^{(0)} \big(1 - \eta_x + \eta_x \frac{\mu_t}{\sqrt{n}}\big)^{r-2} (1-\eta_x)^{R-r}.$$

From the definition of $\alpha_i^{(\ell)}$ from (18), $\alpha_{i_2}^{(r-1)}$ can be written as

$$\alpha_{i_2}^{(r-1)} = C_{\max}^{(0)} (1-\eta_x)^{r-1} + (\lambda_{\max}^{(t)} |\mathbf{x}_{\max}^*| + \beta_{\max}^{(t)}) \sum_{s=1}^{r-1} \eta_x (1-\eta_x)^{r-s}.$$

Therefore, we have

$$\alpha_{i_2}^{(r-1)} (1-\eta_x)^{R-r} = C_{\max}^{(0)} (1-\eta_x)^{R-1} + (\lambda_{\max}^{(t)} |\mathbf{x}_{\max}^*| + \beta_{\max}^{(t)}) \sum_{s=1}^{r-1} \eta_x (1-\eta_x)^{R-s}.$$

Next, to get a handle on $\alpha_{\max}^{(\ell)}\left(1 - \eta_x + \eta_x \frac{\mu_t}{\sqrt{n}}\right)^{r-\ell-2}$, consider the following using the definition of $\alpha_i^{(\ell)}$ from (18), where $\eta_x^{(i)} = \eta_x$ for all $i$,

$$\sum_{\ell=1}^{r} \alpha_{\max}^{(\ell)}\left(1 - \eta_x + \eta_x \frac{\mu_t}{\sqrt{n}}\right)^{r-\ell} = \sum_{\ell=1}^{r} C_{\max}^{(0)}(1 - \eta_x)^{\ell}\left(1 - \eta_x + \eta_x \frac{\mu_t}{\sqrt{n}}\right)^{r-\ell}$$

$$+ (\lambda_{\max}^{(t)}|\mathbf{x}_{\max}^*| + \beta_{\max}^{(t)}) \sum_{\ell=1}^{r} \left(1 - \eta_x + \eta_x \frac{\mu_t}{\sqrt{n}}\right)^{r-\ell} \sum_{s=1}^{\ell} \eta_x(1 - \eta_x)^{\ell-s+1},$$

$$\leq \sum_{\ell=1}^{r} C_{\max}^{(0)}\left(1 - \eta_x + \eta_x \frac{\mu_t}{\sqrt{n}}\right)^{r} + (\lambda_{\max}^{(t)}|\mathbf{x}_{\max}^*| + \beta_{\max}^{(t)}) \sum_{\ell=1}^{r} \left(1 - \eta_x + \eta_x \frac{\mu_t}{\sqrt{n}}\right)^{r-\ell}.$$

Therefore,

$$(1 - \eta_x)^{R-r} \sum_{\ell=1}^{r-2} \alpha_{\max}^{(\ell)}\left(1 - \eta_x + \eta_x \frac{\mu_t}{\sqrt{n}}\right)^{r-\ell-2} \leq \sum_{\ell=1}^{r-2} C_{\max}^{(0)}\left(1 - \eta_x + \eta_x \frac{\mu_t}{\sqrt{n}}\right)^{r-2}(1 - \eta_x)^{R-r}$$

$$+ (\lambda_{\max}^{(t)}|\mathbf{x}_{\max}^*| + \beta_{\max}^{(t)})(1 - \eta_x)^{R-r} \sum_{\ell=1}^{r-2} \left(1 - \eta_x + \eta_x \frac{\mu_t}{\sqrt{n}}\right)^{r-\ell-2},$$

$$\leq (R-2)C_{\max}^{(0)}\left(1 - \eta_x + \eta_x \frac{\mu_t}{\sqrt{n}}\right)^{R-2} + (\lambda_{\max}^{(t)}|\mathbf{x}_{\max}^*| + \beta_{\max}^{(t)})\frac{(1-\eta_x)^{R-r}}{\eta_x(1 - \frac{\mu_t}{\sqrt{n}})}.$$

Therefore,

$$(1 - \eta_x)^{R-r} \sum_{\ell=1}^{r-2} \alpha_{\max}^{(\ell)}\left(1 - \eta_x + \eta_x \frac{\mu_t}{\sqrt{n}}\right)^{r-\ell-2}$$

$$\leq (r-2)C_{\max}^{(0)}\left(1 - \eta_x + \eta_x \frac{\mu_t}{\sqrt{n}}\right)^{R-2} + (\lambda_{\max}^{(t)}|\mathbf{x}_{\max}^*| + \beta_{\max}^{(t)})\frac{(1-\eta_x)^{R-r}}{\eta_x(1 - \frac{\mu_t}{\sqrt{n}})}.$$

Further, since $(1 - \eta_x) \leq (1 - \eta_x + \eta_x \frac{\mu_t}{\sqrt{n}})$,

$$k\eta_x \frac{\mu_t}{\sqrt{n}} C_{\max}^{(0)}\left(1 - \eta_x + \eta_x \frac{\mu_t}{\sqrt{n}}\right)^{r-2}(1 - \eta_x)^{R-r} \leq k\eta_x \frac{\mu_t}{\sqrt{n}} C_{\max}^{(0)} \delta_{R-2}.$$

Therefore, combining all the results we have that, for a constant $c_x = \frac{\mu_t}{\sqrt{n}}/(1 - \frac{\mu_t}{\sqrt{n}})$,

$$C_{i_2}^{(r-1)}(1 - \eta_x)^{R-r}$$

$$\leq (\lambda_{\max}^{(t)}|\mathbf{x}_{\max}^*| + \beta_{\max}^{(t)})\left[\sum_{s=1}^{r-1} \eta_x(1 - \eta_x)^{R-s} + kc_x(1 - \eta_x)^{R-r}\right] + k\eta_x \frac{\mu_t}{\sqrt{n}} C_{\max}^{(0)} \delta_{R-2}.$$

$$\square$$

**Claim 7 (Bound on the noise term in expected gradient vector estimate).** $\|\Delta_j^{(t)}\|$ where $\Delta_j^{(t)} := \mathbf{E}[\mathbf{A}^{(t)}\vartheta^{(R)}\mathrm{sign}(\mathbf{x}_j^*)]$ is upper-bounded as,

$$\|\Delta_j^{(t)}\| = \mathcal{O}(\sqrt{m}q_{i,j}p_j\epsilon_t\|\mathbf{A}^{(t)}\|).$$

*Proof of Claim 7.*

$$\Delta_j^{(t)} = \mathbf{E}[\mathbf{A}^{(t)}\vartheta^{(R)}\mathrm{sign}(\mathbf{x}_j^*)] = \mathbf{E}_S[\mathbf{A}_S^{(t)}\mathbf{E}_{\mathbf{x}_S^*}[\vartheta_S^{(R)}\mathrm{sign}(\mathbf{x}_j^*)|S]]$$

From (30) we have the following definition for $\vartheta_j^{(R)}$

$$\vartheta_j^{(R)} = \beta_j^{(t)} + \sum_{r=1}^{R} \eta_x \sum_{i \neq j} |\langle \mathbf{A}_j^{(t)}, \mathbf{A}_i^{(t)}\rangle|\mathrm{sign}(\langle \mathbf{A}_j^{(t)}, \mathbf{A}_i^{(t)}\rangle)C_i^{(r-1)}\mathrm{sign}(\mathbf{x}_i^* - \mathbf{x}_i^{(r)})(1 - \eta_x)^{R-r} + \gamma_j^{(R)},$$

where $\beta_j^{(t)}$ is defined as the following (11)

$$\beta_j^{(t)} = \sum_{i \neq j}(\langle \mathbf{A}_j^*, \mathbf{A}_i^* - \mathbf{A}_i^{(t)}\rangle + \langle \mathbf{A}_j^* - \mathbf{A}_j^{(t)}, \mathbf{A}_i^{(t)}\rangle)\mathbf{x}_i^* + \sum_{i \neq j}\langle \mathbf{A}_j^{(t)} - \mathbf{A}_j^*, \mathbf{A}_i^*\rangle\mathbf{x}_i^*.$$

Consider $\mathbf{E}_{\mathbf{x}_S^*}[\vartheta_S^{(R)}\text{sign}(\mathbf{x}_j^*)|S]$, where $\vartheta_S^{(R)}$ is a vector with each element as defined in (30). Therefore, the elements of the vector $\mathbf{E}_{\mathbf{x}_S^*}[\vartheta_S^{(R)}\text{sign}(\mathbf{x}_j^*)|S]]$ are given by

$$\mathbf{E}_{\mathbf{x}_S^*}[\vartheta_i^{(R)}\text{sign}(\mathbf{x}_j^*)|S] = \begin{cases} \mathbf{E}_{\mathbf{x}_S^*}[\vartheta_i^{(R)}\text{sign}(\mathbf{x}_j^*)|S], & \text{for } i \neq j, \\ \mathbf{E}_{\mathbf{x}_S^*}[\vartheta_j^{(R)}\text{sign}(\mathbf{x}_j^*)|S], & \text{for } i = j. \end{cases}$$

Consider the general term of interest

$$\mathbf{E}_{\mathbf{x}_S^*}[\vartheta_i^{(R)}\text{sign}(\mathbf{x}_j^*)|S]$$
$$\leq \sum_{r=1}^{R} \eta_x(1-\eta_x)^{R-r}\underbrace{\mathbf{E}_{\mathbf{x}_S^*}[\beta_i\text{sign}(\mathbf{x}_j^*)|S]}_{\clubsuit}$$
$$+ \tfrac{\mu_t}{\sqrt{n}}\sum_{r=1}^{R}\eta_x(1-\eta_x)^{R-r}\sum_{s\neq i}\underbrace{\mathbf{E}_{\mathbf{x}_S^*}[C_s^{(r-1)}\text{sign}(\mathbf{x}_s^* - \mathbf{x}_s^{(r)})\text{sign}(\mathbf{x}_j^*)|S]}_{\spadesuit} + \gamma_i^{(R)}.$$

Further, since

$$\mathbf{E}_{\mathbf{x}_S^*}[\mathbf{x}_i^*\text{sign}(\mathbf{x}_j^*)|S] = \begin{cases} 0, & \text{for } i \neq j, \\ p_j, & \text{for } i = j, \end{cases}$$

we have that

$$\clubsuit := \mathbf{E}_{\mathbf{x}_S^*}[\beta_i^{(t)}\text{sign}(\mathbf{x}_j^*)|S] \leq \begin{cases} 3p_j\epsilon_t & \text{, for } i \neq j, \\ 0 & \text{, for } i = j. \end{cases} \tag{31}$$

Further, for $\spadesuit_s := \mathbf{E}_{\mathbf{x}_S^*}[C_s^{(r-1)}\text{sign}(\mathbf{x}_s^* - \mathbf{x}_s^{(r)})\text{sign}(\mathbf{x}_j^*)|S]$ we have that

$$\spadesuit_s = \begin{cases} \mathbf{E}_{\mathbf{x}_S^*}[C_j^{(r-1)}(\mathbf{x}_j^* - \mathbf{x}_j^{(r-1)})\text{sign}(\mathbf{x}_j^*)|S] \leq C_j^{(r-1)}, & \text{for } s = j \\ 0, & \text{for } s \neq j. \end{cases}$$

In addition, for $\sum_{s\neq i}\spadesuit_s$ we have that

$$\sum_{s\neq i}\spadesuit_s = \begin{cases} C_j^{(r-1)}, & \text{for } i \neq j \\ 0, & \text{for } i = j. \end{cases} \tag{32}$$

Therefore, using the results for $\clubsuit$ and $\sum_{s\neq i}\spadesuit_s$, we have that $\mathbf{E}_{\mathbf{x}_S^*}[\vartheta_j^{(R)}\text{sign}(\mathbf{x}_j^*)|S] = \gamma_i^{(R)}$ for $i = j$, and for $i \neq j$ we have

$$\mathbf{E}_{\mathbf{x}_S^*}[\vartheta_i^{(R)}\text{sign}(\mathbf{x}_j^*)|S]$$
$$\leq 3p_j\epsilon_t + \tfrac{\mu_t}{\sqrt{n}}\sum_{r=1}^{R}\mathbf{E}_{\mathbf{x}_S^*}[C_j^{(r-1)}\text{sign}(\mathbf{x}_j^* - \mathbf{x}_j^{(r)})\text{sign}(\mathbf{x}_j^*)|S]\eta_x(1-\eta_x)^{R-r} + \gamma_i^{(R)},$$
$$\leq 3p_j\epsilon_t + \tfrac{\mu_t}{\sqrt{n}}\sum_{r=1}^{R}C_j^{(r-1)}\eta_x(1-\eta_x)^{R-r} + \gamma_i^{(R)}. \tag{33}$$

Here, from Claim 6, for $c_x = \tfrac{\mu_t}{\sqrt{n}}/(1-\tfrac{\mu_t}{\sqrt{n}})$ we have

$$C_j^{(r-1)}(1-\eta_x)^{R-r}$$
$$\leq (\lambda_{\max}^{(t)}|\mathbf{x}_{\max}^*| + \beta_{\max}^{(t)})\left[\sum_{s=1}^{r-1}\eta_x(1-\eta_x)^{R-s} + kc_x(1-\eta_x)^{R-r}\right] + k\eta_x\tfrac{\mu_t}{\sqrt{n}}C_{\max}^{(0)}\delta_{R-2}.$$

Further, due to our assumptions on sparsity, $kc_x \leq 1$; in addition by Claim 2, and with probability at least $(1-\delta_\beta^{(t)})$ we have $|\beta_{\max}^{(t)}| \leq t_\beta$, substituting,

$$\sum_{r=1}^{R}C_j^{(r-1)}\eta_x(1-\eta_x)^{R-r}$$

$$\leq (\lambda_{\max}^{(t)}|\mathbf{x}_{\max}^*| + \beta_{\max}^{(t)})\left[\sum_{r=1}^{R} \eta_x \sum_{s=1}^{r-1} \eta_x(1-\eta_x)^{R-s} + kc_x \sum_{r=1}^{R} \eta_x(1-\eta_x)^{R-r}\right],$$

$$\leq (\lambda_{\max}^{(t)}|\mathbf{x}_{\max}^*| + t_\beta)(1 + kc_x),$$

$$= \mathcal{O}(t_\beta),$$

with probability at least $(1 - \delta_\beta^{(t)})$. Combining results from (31), (32) and substituting for the terms in (33) using the analysis above,

$$\mathbf{E}_{\mathbf{x}_S^*}[\vartheta_i^{(R)}\text{sign}(\mathbf{x}_j^*)|S] \begin{cases} \leq \gamma_i^{(R)}, & \text{for } i = j, \\ \leq 3p_j\epsilon_t + \frac{\mu}{\sqrt{n}}t_\beta + \gamma_i^{(R)} = \mathcal{O}(p_j\epsilon_t), & \text{for } i \neq j. \end{cases}$$

Note that since $\gamma_i^{(R)} := (1-\eta_x)^R(\mathbf{x}_i^{(0)} - \mathbf{x}_i^*(1-\lambda_i^{(t)}))$ can be made small by choice of $R$. Also, since $\mathbf{Pr}[i, j \in S] = q_{i,j}$, we have

$$\|\Delta_j^{(t)}\| = \|\mathbf{E}_S[\mathbf{A}_S^{(t)}\mathbf{E}_{\mathbf{x}_S^*}[\vartheta_S^{(R)}\text{sign}(\mathbf{x}_j^*)|S]]\|,$$

$$\leq \mathcal{O}(\sqrt{m}q_{i,j}p_j\epsilon_t\|\mathbf{A}^{(t)}\|).$$

$\square$

**Claim 8** (**An intermediate result for concentration results**). *With probability* $(1 - \delta_\beta^{(t)} - \delta_{\mathcal{T}}^{(t)} - \delta_{\text{HW}}^{(t)})$ $\|\mathbf{y} - \mathbf{A}^{(t)}\widehat{\mathbf{x}}\|$ *is upper-bounded by* $\widetilde{\mathcal{O}}(kt_\beta)$ .

*Proof of Claim 8.* First, using Lemma 4 we have

$$\widehat{\mathbf{x}}_{i_1} := \mathbf{x}_{i_1}^{(R)} = \mathbf{x}_{i_1}^*(1-\lambda_{i_1}^{(t)}) + \vartheta_{i_1}^{(R)}.$$

Therefore, the vector $\widehat{\mathbf{x}}_S$, for $S \in \text{supp}(\mathbf{x}^*)$ can be written as

$$\widehat{\mathbf{x}}_S := \mathbf{x}_S^{(R)} = (\mathbf{I} - \Lambda_S^{(t)})\mathbf{x}_S^* + \vartheta_S^{(R)}, \tag{34}$$

where $\widehat{\mathbf{x}}$ has the correct signed-support with probability at least $(1 - \delta_{\mathcal{T}})$ using Lemma 2. Using this result, we can write $\|\mathbf{y} - \mathbf{A}^{(t)}\widehat{\mathbf{x}}\|$ as

$$\|\mathbf{y} - \mathbf{A}^{(t)}\widehat{\mathbf{x}}\| = \|\mathbf{A}_S^*\mathbf{x}_S^* - \mathbf{A}_S^{(t)}(\mathbf{I} - \Lambda_S^{(t)})\mathbf{x}_S^* - \mathbf{A}_S^{(t)}\vartheta_S^{(R)}\|.$$

Now, since $\Lambda_{ii}^{(t)} \leq \frac{\epsilon_t^2}{2}$ we have

$$\|\mathbf{y} - \mathbf{A}^{(t)}\widehat{\mathbf{x}}\| \leq \|\mathbf{A}_S^*\mathbf{x}_S^* - (1 - \frac{\epsilon_t^2}{2})\mathbf{A}_S^{(t)}\mathbf{x}_S^* - \mathbf{A}_S^{(t)}\vartheta_S^{(R)}\|,$$

$$= \|\underbrace{((1 - \frac{\epsilon_t^2}{2})(\mathbf{A}_S^* - \mathbf{A}_S^{(t)}) + \frac{\epsilon_t^2}{2}\mathbf{A}_S^*)}_{\clubsuit}\mathbf{x}_S^* - \underbrace{\mathbf{A}_S^{(t)}\vartheta_S^{(R)}}_{\spadesuit}\|.$$

With $\mathbf{x}_S^*$ being independent and sub-Gaussian, using Lemma 13, which is a result based on the Hanson-Wright result (Hanson and Wright, 1971) for sub-Gaussian random variables, and since $\|\mathbf{A}_S^{(t)} - \mathbf{A}_S^*\| \leq \|\mathbf{A}_S^{(t)} - \mathbf{A}_S^*\|_F \leq \sqrt{k}\epsilon_t$, we have that with probability at least $(1 - \delta_{\text{HW}}^{(t)})$

$$\|\clubsuit\mathbf{x}_S^*\| = \|((1 - \frac{\epsilon_t^2}{2})(\mathbf{A}_S^* - \mathbf{A}_S^{(t)}) + \frac{\epsilon_t^2}{2}\mathbf{A}_S^*)\mathbf{x}_S^*\| \leq \widetilde{\mathcal{O}}(\|(1 - \frac{\epsilon_t^2}{2})(\mathbf{A}_S^* - \mathbf{A}_S^{(t)}) + \frac{\epsilon_t^2}{2}\mathbf{A}_S^*\|_F),$$

where $\delta_{\text{HW}}^{(t)} = \exp(-\frac{1}{\mathcal{O}(\epsilon_t)})$.

Now, consider the $\|\clubsuit\|_F$, since $\|\mathbf{A}_S^{(t)} - \mathbf{A}_S^*\|_F \leq \sqrt{k}\epsilon_t$

$$\|\clubsuit\|_F := \|(1 - \frac{\epsilon_t^2}{2})(\mathbf{A}_S^* - \mathbf{A}_S^{(t)}) + \frac{\epsilon_t^2}{2}\mathbf{A}_S^*\|_F \leq (1 - \frac{\epsilon_t^2}{2})\|(\mathbf{A}_S^* - \mathbf{A}_S^{(t)})\|_F + \frac{\epsilon_t^2}{2}\|\mathbf{A}_S^*\|_F,$$

$$\leq \sqrt{k}(1 - \frac{\epsilon_t^2}{2})\epsilon_t + \frac{\epsilon_t^2}{2}\|\mathbf{A}_S^*\|_F.$$

Consider the $\|\spadesuit\|$ term. Using Claim 5, each $\vartheta_j^{(R)}$ is bounded by $\mathcal{O}(t_\beta)$. with probability at least $(1 - \delta_\beta^{(t)})$ Therefore,

$$\|\spadesuit\| = \|\mathbf{A}_S^{(t)}\vartheta_S^{(R)}\| \leq \|\mathbf{A}_S^{(t)}\|\|\vartheta_S^{(R)}\| = \|\mathbf{A}_S^{(t)}\|\sqrt{k}\mathcal{O}(t_\beta).$$

Again, since $\|\mathbf{A}_S^{(t)} - \mathbf{A}_S^*\| \leq \|\mathbf{A}_S^{(t)} - \mathbf{A}_S^*\|_F \leq \sqrt{k}\epsilon_t$,

$$\|\mathbf{A}_S^{(t)}\| \leq \|\mathbf{A}_S^{(t)} - \mathbf{A}_S^* + \mathbf{A}_S^*\| \leq \|\mathbf{A}_S^{(t)} - \mathbf{A}_S^*\| + \|\mathbf{A}_S^*\| \leq \sqrt{k}\epsilon_t + 2.$$

Finally, combining all the results and using the fact that $\|\mathbf{A}_S^*\|_F \leq \sqrt{k}\|\mathbf{A}_S^*\| \leq 2\sqrt{k},$,

$$\|\mathbf{y} - \mathbf{A}^{(t)}\widehat{\mathbf{x}}\| = \widetilde{\mathcal{O}}(\sqrt{k}(1 - \tfrac{\epsilon_t^2}{2})\epsilon_t + \epsilon_t^2\sqrt{k}) + \|\mathbf{A}_S^{(t)}\|\sqrt{k}\mathcal{O}(t_\beta),$$
$$= \widetilde{\mathcal{O}}(kt_\beta).$$

$\square$

**Claim 9 (Bound on variance parameter for concentration of gradient vector).** *For* $\mathbf{z} := (\mathbf{y} - \mathbf{A}^{(t)}\widehat{\mathbf{x}})\mathrm{sign}(\widehat{\mathbf{x}}_i)|i \in S$ *the variance parameter* $\mathbf{E}[\|\mathbf{z}\|^2]$ *is bounded as* $\mathbf{E}[\|\mathbf{z}\|^2] = \mathcal{O}(k\epsilon_t^2) + \mathcal{O}(kt_\beta^2)$ *with probability at least* $(1 - \delta_\beta^{(t)} - \delta_{\mathcal{T}}^{(t)})$.

*Proof of Claim 9.* For the variance $\mathbf{E}[\|\mathbf{z}\|^2]$, we focus on the following,

$$\mathbf{E}[\|\mathbf{z}\|^2] = \mathbf{E}[\|(\mathbf{y} - \mathbf{A}^{(t)}\widehat{\mathbf{x}})\mathrm{sign}(\widehat{\mathbf{x}}_i)\|^2|i \in S].$$

Here, $\widehat{\mathbf{x}}_S$ is given by

$$\widehat{\mathbf{x}}_S = (\mathbf{I} - \Lambda_S^{(t)})\mathbf{x}_S^* + \vartheta_S^{(R)}.$$

Therefore, $\mathbf{E}[\|\mathbf{z}\|^2]$ can we written as

$$\mathbf{E}[\|(\mathbf{y} - \mathbf{A}^{(t)}\widehat{\mathbf{x}})\mathrm{sign}(\widehat{\mathbf{x}}_i)\|^2|i \in S]$$
$$= \mathbf{E}[\|(\mathbf{y} - \mathbf{A}_S^{(t)}(\mathbf{I} - \Lambda_S^{(t)})\mathbf{x}_S^* - \mathbf{A}_S^{(t)}\vartheta_S^{(R)})\mathrm{sign}(\widehat{\mathbf{x}}_i)\|^2|i \in S],$$
$$\leq \underbrace{\mathbf{E}[\|(\mathbf{A}_S^* - \mathbf{A}_S^{(t)}(\mathbf{I} - \Lambda_S^{(t)}))\mathbf{x}_S^*\|^2|i \in S]}_{\heartsuit} + \underbrace{\mathbf{E}[\|\mathbf{A}_S^{(t)}\vartheta_S^{(R)}\mathrm{sign}(\widehat{\mathbf{x}}_i)\|^2|i \in S]}_{\diamondsuit}. \quad (35)$$

We will now consider each term in (35) separately. We start with $\heartsuit$. Since $\mathbf{x}_S^*$s are conditionally independent of $S$, $\mathbf{E}[\mathbf{x}_S^*\mathbf{x}_S^{*\top}] = \mathbf{I}$. Therefore, we can simplify this expression as

$$\heartsuit := \mathbf{E}[\|(\mathbf{A}_S^* - \mathbf{A}_S^{(t)}(\mathbf{I} - \Lambda_S^{(t)}))\mathbf{x}_S^*\|^2|i \in S] = \mathbf{E}[\|\mathbf{A}_S^* - \mathbf{A}_S^{(t)}(\mathbf{I} - \Lambda_S^{(t)})\|_F^2|i \in S].$$

Rearranging the terms we have the following for $\heartsuit$,

$$\heartsuit = \mathbf{E}[\|\mathbf{A}_S^* - \mathbf{A}_S^{(t)}(\mathbf{I} - \Lambda_S^{(t)})\|_F^2|i \in S] = \mathbf{E}[\|\mathbf{A}_S^*\Lambda_S^{(t)} + (\mathbf{A}_S^* - \mathbf{A}_S^{(t)})(\mathbf{I} - \Lambda_S^{(t)})\|_F^2|i \in S].$$

Therefore, $\heartsuit$ can be upper-bounded as

$$\heartsuit \leq \underbrace{\mathbf{E}[\|\mathbf{A}_S^*\Lambda_S^{(t)}\|_F^2|i \in S]}_{\heartsuit_1} + \underbrace{\mathbf{E}[\|(\mathbf{A}_S^* - \mathbf{A}_S^{(t)})(\mathbf{I} - \Lambda_S^{(t)})\|_F^2|i \in S]}_{\heartsuit_2}$$
$$+ \underbrace{2\mathbf{E}[\|\mathbf{A}_S^*\Lambda_S^{(t)}\|_F\|(\mathbf{A}_S^* - \mathbf{A}_S^{(t)})(\mathbf{I} - \Lambda_S^{(t)})\|_F|i \in S]}_{\heartsuit_3}.$$
(36)

For $\heartsuit_1$, since $\|\mathbf{A}_S^{(t)}\| \leq \sqrt{k}\epsilon_t + 2$, we have

$$\heartsuit_1 := \mathbf{E}[\|\mathbf{A}_S^*\Lambda_S^{(t)}\|_F^2|i \in S] \leq \mathbf{E}[\|\mathbf{A}_S^*\|\|\Lambda_S^{(t)}\|_F^2|i \in S] \leq \|\mathbf{A}_S^*\|\sum_{j \in S}(\lambda_j^{(t)})^2 \leq k(\sqrt{k}\epsilon_t + 2)\tfrac{\epsilon_t^4}{4}.$$

Next, since $(1 - \lambda_j^{(t)}) \leq 1$, we have the following bound for $\heartsuit_2$

$$\heartsuit_2 := \mathbf{E}[\|(\mathbf{A}_S^* - \mathbf{A}_S^{(t)})(\mathbf{I} - \Lambda_S^{(t)})\|_F^2|i \in S] \leq \mathbf{E}[\|\mathbf{A}_S^* - \mathbf{A}_S^{(t)}\|_F^2|i \in S] \leq \|\mathbf{A}_S^* - \mathbf{A}_S^{(t)}\|_F^2 \leq k\epsilon_t^2.$$

Further, $\heartsuit_3$ can be upper-bounded by using bounds for $\heartsuit_1$ and $\heartsuit_2$. Combining the results of upper-bounding $\heartsuit_1$, $\heartsuit_2$, and $\heartsuit_3$ we have the following for (36)

$$\heartsuit \leq \mathbf{E}[\|(\mathbf{A}_S^* - \mathbf{A}_S^{(t)}(\mathbf{I} - \Lambda_S^{(t)}))\mathbf{x}_S^*\|^2|i \in S] = \mathcal{O}(k\epsilon_t^2).$$

Next, by Claim 5, $\vartheta_j^{(R)}$ is upper-bounded as $|\vartheta_j^{(R)}| \leq \mathcal{O}(t_\beta)$. with probability $(1 - \delta_\beta^{(t)})$. Therefore, the term $\diamondsuit$, the second term of (35), can be bounded as

$$\diamondsuit \leq \|\mathbf{A}_S^{(t)} \vartheta_S^{(R)} \text{sign}(\widehat{\mathbf{x}}_i)\|^2 \leq (\sqrt{k}\epsilon_t + 2)^2 k\mathcal{O}(t_\beta)^2 = \mathcal{O}(kt_\beta^2).$$

Finally, combining all the results, the term of interest in (35) has the following form

$$\mathbf{E}[\|(\mathbf{y} - \mathbf{A}^{(t)}\widehat{\mathbf{x}})\text{sign}(\widehat{\mathbf{x}}_i)\|^2 | i \in S] = \mathcal{O}(k\epsilon_t^2) + \mathcal{O}(kt_\beta^2).$$

$\square$

**Claim 10** (**Bound on variance parameter for concentration of gradient matrix**). *With probability* $(1 - \delta_{\mathcal{T}}^{(t)} - \delta_\beta^{(t)})$, *the variance parameter* $\|\mathbf{E}[(\mathbf{y} - \mathbf{A}^{(t)}\widehat{\mathbf{x}})(\mathbf{y} - \mathbf{A}\widehat{\mathbf{x}})^\top]\|$ *is upper-bounded by* $\mathcal{O}(\frac{k^2 t_\beta^2}{m})\|\mathbf{A}^*\|^2$.

*Proof of Claim 10.* Let $\mathcal{F}_{\mathbf{x}^*}$ be the event that $\text{sign}(\mathbf{x}^*) = \text{sign}(\widehat{\mathbf{x}})$, and let $\mathbb{1}_{\mathcal{F}_{\mathbf{x}^*}}$ denote the indicator function corresponding to this event. As we show in Lemma 2, this event occurs with probability at least $(1 - \delta_\beta^{(t)} - \delta_{\mathcal{T}}^{(t)})$, therefore,

$$\begin{aligned}
&\mathbf{E}[(\mathbf{y} - \mathbf{A}^{(t)}\widehat{\mathbf{x}})(\mathbf{y} - \mathbf{A}^{(t)}\widehat{\mathbf{x}})^\top] \\
&\quad = \mathbf{E}[(\mathbf{y} - \mathbf{A}^{(t)}\widehat{\mathbf{x}})(\mathbf{y} - \mathbf{A}^{(t)}\widehat{\mathbf{x}})^\top \mathbb{1}_{\mathcal{F}_{\mathbf{x}^*}}] + \mathbf{E}[(\mathbf{y} - \mathbf{A}^{(t)}\widehat{\mathbf{x}})(\mathbf{y} - \mathbf{A}^{(t)}\widehat{\mathbf{x}})^\top \mathbb{1}_{\bar{\mathcal{F}}_{x^*}}], \\
&\quad = \mathbf{E}[(\mathbf{y} - \mathbf{A}^{(t)}\widehat{\mathbf{x}})(\mathbf{y} - \mathbf{A}^{(t)}\widehat{\mathbf{x}})^\top \mathbb{1}_{\mathcal{F}_{x^*}}] \pm \gamma.
\end{aligned}$$

Here, $\gamma$ is small. Under the event $\mathcal{F}_{x^*}$, $\widehat{\mathbf{x}}$ has the correct signed-support. Again, since $\mathbb{1}_{\mathcal{F}_{\mathbf{x}^*}} = 1 - \mathbb{1}_{\bar{\mathcal{F}}_{\mathbf{x}^*}}$,

$$\begin{aligned}
\mathbf{E}[(\mathbf{y} - \mathbf{A}^{(t)}\widehat{\mathbf{x}})(\mathbf{y} - \mathbf{A}^{(t)}\widehat{\mathbf{x}})^\top] &= \mathbf{E}[(\mathbf{y} - \mathbf{A}^{(t)}\widehat{\mathbf{x}})(\mathbf{y} - \mathbf{A}^{(t)}\widehat{\mathbf{x}})^\top (1 - \mathbb{1}_{\bar{\mathcal{F}}_{x^*}})] \pm \gamma, \\
&= \mathbf{E}[(\mathbf{y} - \mathbf{A}^{(t)}\widehat{\mathbf{x}})(\mathbf{y} - \mathbf{A}^{(t)}\widehat{\mathbf{x}})^\top] \pm \gamma.
\end{aligned}$$

Now, using Lemma 4 with probability at least $(1 - \delta_{\mathcal{T}}^{(t)} - \delta_\beta^{(t)})$, $\widehat{\mathbf{x}}_S$ admits the following expression

$$\widehat{\mathbf{x}}_S := \mathbf{x}_S^{(R)} = (\mathbf{I} - \Lambda_S^{(t)})\mathbf{x}_S^* + \vartheta_S^{(R)}.$$

Therefore we have

$$\mathbf{y} - \mathbf{A}^{(t)}\widehat{\mathbf{x}} = (\mathbf{A}_S^* - \mathbf{A}_S^{(t)}(\mathbf{I} - \Lambda_S^{(t)}))\mathbf{x}_S^* - \mathbf{A}_S^{(t)}\vartheta_S^{(R)}.$$

Using the expression above $\mathbf{E}[(\mathbf{y} - \mathbf{A}^{(t)}\widehat{\mathbf{x}})(\mathbf{y} - \mathbf{A}^{(t)}\widehat{\mathbf{x}})^\top]$ can be written as

$$\begin{aligned}
&\mathbf{E}[(\mathbf{y} - \mathbf{A}^{(t)}\widehat{\mathbf{x}})(\mathbf{y} - \mathbf{A}^{(t)}\widehat{\mathbf{x}})^\top] \\
&\quad = \mathbf{E}[((\mathbf{A}_S^* - \mathbf{A}_S^{(t)}(\mathbf{I} - \Lambda_S^{(t)}))\mathbf{x}_S^* - \mathbf{A}_S^{(t)}\vartheta_S^{(R)})((\mathbf{A}_S^* - \mathbf{A}_S^{(t)}(\mathbf{I} - \Lambda_S^{(t)}))\mathbf{x}_S^* - \mathbf{A}_S^{(t)}\vartheta_S^{(R)})^\top].
\end{aligned}$$

Sub-conditioning, we have

$$\begin{aligned}
&\mathbf{E}[(\mathbf{y} - \mathbf{A}^{(t)}\widehat{\mathbf{x}})(\mathbf{y} - \mathbf{A}^{(t)}\widehat{\mathbf{x}})^\top] \\
&\quad = \mathbf{E}_S[(\mathbf{A}_S^* - \mathbf{A}_S^{(t)}(\mathbf{I} - \Lambda_S^{(t)}))\mathbf{E}_{\mathbf{x}_S^*}[\mathbf{x}_S^* \mathbf{x}_S^{*\top}|S](\mathbf{A}_S^{*\top} - (\mathbf{I} - \Lambda_S^{(t)})\mathbf{A}_S^{(t)\top})] \\
&\qquad - \mathbf{E}_S[\mathbf{A}_S^{(t)}\mathbf{E}_{\mathbf{x}_S^*}[\vartheta_S^{(R)}\mathbf{x}_S^{*\top}|S](\mathbf{A}_S^{*\top} - (\mathbf{I} - \Lambda_S^{(t)})\mathbf{A}_S^{(t)\top})] \\
&\qquad - \mathbf{E}_S[(\mathbf{A}_S^* - \mathbf{A}_S^{(t)}(\mathbf{I} - \Lambda_S^{(t)}))\mathbf{E}_{\mathbf{x}_S^*}[\mathbf{x}_S^*(\vartheta_S^{(R)})^\top|S]\mathbf{A}_S^{(t)\top}] \\
&\qquad + \mathbf{E}_S[\mathbf{A}_S^{(t)}\mathbf{E}_{\mathbf{x}_S^*}[\vartheta_S^{(R)}(\vartheta_S^{(R)})^\top|S]\mathbf{A}_S^{(t)\top}].
\end{aligned}$$

Now, since $\mathbf{E}_{\mathbf{x}_S^*}[\mathbf{x}_S^* \mathbf{x}_S^{*\top}|S] = \mathbf{I}$,

$$\|\mathbf{E}[(\mathbf{y} - \mathbf{A}^{(t)}\widehat{\mathbf{x}})(\mathbf{y} - \mathbf{A}^{(t)}\widehat{\mathbf{x}})^\top]\| \leq \underbrace{\|\mathbf{E}_S[(\mathbf{A}_S^* - \mathbf{A}_S^{(t)}(\mathbf{I} - \Lambda_S^{(t)}))(\mathbf{A}_S^{*\top} - (\mathbf{I} - \Lambda_S^{(t)})\mathbf{A}_S^{(t)\top})]\|}_{\clubsuit}$$

$$+ \underbrace{\|\mathbf{E}_S[\mathbf{A}_S^{(t)}\mathbf{E}_{\mathbf{x}_S^*}[\vartheta_S^{(R)}\mathbf{x}_S^{*\top}|S](\mathbf{A}_S^{*\top} - (\mathbf{I} - \Lambda_S^{(t)})\mathbf{A}_S^{(t)\top})]\|}_{\spadesuit}$$

$$+ \underbrace{\|\mathbf{E}_S[(\mathbf{A}_S^* - \mathbf{A}_S^{(t)}(\mathbf{I} - \Lambda_S^{(t)}))\mathbf{E}_{\mathbf{x}_S^*}[\mathbf{x}_S^*(\vartheta_S^{(R)})^\top|S]\mathbf{A}_S^{(t)\top}]\|}_{\heartsuit}$$

$$+ \underbrace{\|\mathbf{E}_S[\mathbf{A}_S^{(t)}\mathbf{E}_{\mathbf{x}_S^*}[\vartheta_S^{(R)}(\vartheta_S^{(R)})^\top|S]\mathbf{A}_S^{(t)\top}]\|}_{\diamond}. \tag{37}$$

Let's start with the first term ($\clubsuit$) of (37), which can be written as

$$\clubsuit :\leq \underbrace{\|\mathbf{E}_S[\mathbf{A}_S^*\mathbf{A}_S^{*\top}]\|}_{\clubsuit_1} + \underbrace{\|\mathbf{E}_S[\mathbf{A}_S^*(\mathbf{I} - \Lambda_S^{(t)})\mathbf{A}_S^{(t)\top}]\|}_{\clubsuit_2} + \underbrace{\|\mathbf{E}_S[\mathbf{A}_S^{(t)}(\mathbf{I} - \Lambda_S^{(t)})\mathbf{A}_S^{*\top}]\|}_{\clubsuit_3}$$

$$+ \underbrace{\|\mathbf{E}_S[\mathbf{A}_S^{(t)}(\mathbf{I} - \Lambda_S^{(t)})^2\mathbf{A}_S^{(t)\top}]\|}_{\clubsuit_4}. \tag{38}$$

Now consider each term of equation (38). First, since

$$\mathbf{E}_S[\mathbf{A}_S^*\mathbf{A}_S^{*\top}] = \mathbf{E}_S\Big[\sum_{i,j\in S}\mathbf{A}_i^*\mathbf{A}_j^{*\top}\mathbb{1}_{i,j\in S}\Big] = \sum_{i,j=1}^m \mathbf{A}_i^*\mathbf{A}_i^{*\top}\mathbf{E}_S[\mathbb{1}_{i,j\in S}],$$

and $\mathbf{E}_S[\mathbb{1}_{i,j\in S}] = \mathcal{O}(\frac{k^2}{m^2})$, we can upper-bound $\clubsuit_1 := \|\mathbf{E}_S[\mathbf{A}_S^*\mathbf{A}_S^{*\top}]\|$ as

$$\clubsuit_1 := \|\mathbf{E}_S[\mathbf{A}_S^*\mathbf{A}_S^{*\top}]\| = \mathcal{O}(\tfrac{k^2}{m^2})\|\mathbf{A}^*\mathbf{1}^{m\times m}\mathbf{A}^{*\top}\| = \mathcal{O}(\tfrac{k^2}{m})\|\mathbf{A}^*\|^2,$$

where $\mathbf{1}^{m\times m}$ denotes an $m \times m$ matrix of ones. Now, we turn to $\clubsuit_2 := \|\mathbf{E}_S[\mathbf{A}_S^*(\mathbf{I} - \Lambda_S^{(t)})\mathbf{A}_S^{(t)\top}]\|$ in (38), which can be simplified as

$$\clubsuit_2 \leq \Big\|\sum_{i,j=1}^m \mathbf{A}_i^*\mathbf{A}_j^{(t)\top}\mathbf{E}_S[\mathbb{1}_{i,j\in S}]\Big\| \leq \mathcal{O}(\tfrac{k^2}{m})\|\mathbf{A}^*\|\|\mathbf{A}^{(t)}\|.$$

Further, since $\mathbf{A}^{(t)}$ is $(\epsilon_t, 2)$-close to $\mathbf{A}^*$, we have that $\|\mathbf{A}^{(t)}\| \leq \|\mathbf{A}^{(t)} - \mathbf{A}^*\| + \|\mathbf{A}^*\| \leq 3\|\mathbf{A}^*\|$, therefore

$$\clubsuit_2 := \|\mathbf{E}_S[\mathbf{A}_S^*(\mathbf{I} - \Lambda_S^{(t)})\mathbf{A}_S^{(t)\top}]\| = \mathcal{O}(\tfrac{k^2}{m})\|\mathbf{A}^*\|^2.$$

Similarly, $\clubsuit_3$ (38) is also $\mathcal{O}(\frac{k^2}{m})\|\mathbf{A}^*\|^2$. Next, we consider $\clubsuit_4 := \|\mathbf{E}_S[\mathbf{A}_S^{(t)}(\mathbf{I} - \Lambda_S^{(t)})^2\mathbf{A}_S^{(t)\top}]\|$ in (38) which can also be bounded similarly as

$$\clubsuit_4 = \mathcal{O}(\tfrac{k^2}{m})\|\mathbf{A}^*\|^2.$$

Therefore, we have the following for $\clubsuit$ in (37)

$$\clubsuit := \mathbf{E}_S[(\mathbf{A}_S^* - \mathbf{A}_S^{(t)}(\mathbf{I} - \Lambda_S^{(t)}))(\mathbf{A}_S^{*\top} - (\mathbf{I} - \Lambda_S^{(t)})\mathbf{A}_S^{(t)\top})] = \mathcal{O}(\tfrac{k^2}{m})\|\mathbf{A}^*\|^2. \tag{39}$$

Consider $\spadesuit$ in (37). Letting $\mathbf{M} = \mathbf{E}_{\mathbf{x}_S^*}[\vartheta_S^{(R)}\mathbf{x}_S^{*\top}|S]$, and using the analysis similar to that shown in 7, we have that elements of $\mathbf{M} \in \mathbb{R}^{k\times k}$ are given by

$$\mathbf{M}_{i,j} = \mathbf{E}_{\mathbf{x}_S^*}[\vartheta_i^{(R)}\mathbf{x}_j^*|S] \begin{cases} \leq \mathcal{O}(\gamma_i^{(R)}), & \text{for } i = j, \\ = \mathcal{O}(\epsilon_t), & \text{for } i \neq j. \end{cases}$$

We have the following,

$$\spadesuit := \mathbf{E}_S[\mathbf{A}_S^{(t)}\mathbf{E}_{\mathbf{x}_S^*}[\vartheta_S^{(R)}\mathbf{x}_S^{*\top}|S](\mathbf{A}_S^{*\top} - (\mathbf{I} - \Lambda_S^{(t)})\mathbf{A}_S^{(t)\top})] = \mathbf{E}_S[\mathbf{A}_S^{(t)}\mathbf{M}(\mathbf{A}_S^{*\top} - (\mathbf{I} - \Lambda_S^{(t)})\mathbf{A}_S^{(t)\top})].$$

Therefore, since $\mathbf{E}_S[\mathbb{1}_{i,j\in S}|S] = \mathcal{O}(\frac{k^2}{m^2})$, and $\|\mathbf{1}^{m\times m}\| = m$,

$$\spadesuit := \|\mathbf{E}_S[\mathbf{A}_S^{(t)}\mathbf{M}(\mathbf{A}_S^{*\top} - (\mathbf{I} - \Lambda_S^{(t)})\mathbf{A}_S^{(t)\top})]\|$$

$$= \| \sum_{i,j=1}^{m} \mathbf{M}_{i,j} \mathbf{A}_i^{(t)} (\mathbf{A}_j^{*\top} - (1 - \lambda_j^{(t)}) \mathbf{A}_j^{(t)\top}) \mathbf{E}_S[\mathbb{1}_{i,j \in S}|S]\|,$$

$$= \mathcal{O}(\epsilon_t) \| \sum_{i,j=1}^{m} \mathbf{A}_i^{(t)} (\mathbf{A}_j^{*\top} - (1 - \lambda_j^{(t)}) \mathbf{A}_j^{(t)\top}) \mathbf{E}_S[\mathbb{1}_{i,j \in S}|S]\|,$$

$$= \mathcal{O}(\epsilon_t) \mathcal{O}(\tfrac{k^2}{m^2}) (\|\mathbf{A}^{(t)} \mathbf{1}^{m \times m} \mathbf{A}^{*\top}\| + \|\mathbf{A}^{(t)} \mathbf{1}^{m \times m} \mathbf{A}^{(t)\top}\|),$$

$$= \mathcal{O}(\epsilon_t) \mathcal{O}(\tfrac{k^2}{m}) \|\mathbf{A}^*\|^2.$$

Therefore,

$$\spadesuit := \|\mathbf{E}_S[\mathbf{A}_S^{(t)} \mathbf{M}(\mathbf{A}_S^{*\top} - (\mathbf{I} - \Lambda_S^{(t)}) \mathbf{A}_S^{(t)\top})]\| = \mathcal{O}(\tfrac{k^2}{m}) \epsilon_t \|\mathbf{A}^*\|^2.$$

Similarly, $\heartsuit$ in (37) is also bounded as $\spadesuit$. Next, we consider $\diamondsuit$ in (37). In this case, letting $\mathbf{E}_{\mathbf{x}_S^*}[\vartheta_S^{(R)}(\vartheta_S^{(R)})^\top|S] = \mathbf{N}$, where $\mathbf{N} \in \mathbb{R}^{k \times k}$ is a matrix whose each entry $\mathbf{N}_{i,j} \leq |\vartheta_i^{(R)}||\vartheta_j^{(R)}|$. Further, by Claim 5, each element $\vartheta_j^{(R)}$ is upper-bounded as

$$|\vartheta_j^{(R)}| \leq \mathcal{O}(t_\beta).$$

with probability at least $(1 - \delta_\beta^{(t)})$. Therefore,

$$\diamondsuit = \| \sum_{i,j=1}^{m} \mathbf{N}_{i,j} \mathbf{A}_i^{(t)} \mathbf{A}_j^{(t)\top} \mathbf{E}_S[\mathbb{1}_{i,j \in S}|S]\| = \max_{i,j} |\vartheta_i^{(R)}||\vartheta_j^{(R)}| \mathcal{O}(\tfrac{k^2}{m^2}) \| \sum_{i,j=1}^{m} \mathbf{A}_i^{(t)} \mathbf{A}_j^{(t)\top}\|.$$

Again, using the result on $|\vartheta_{i_1}^{(R)}|$, we have

$$\diamondsuit := \|\mathbf{E}_S[\mathbf{A}_S^{(t)} \mathbf{N} \mathbf{A}_S^{(t)\top}]\| = m \max_{i,j} |\vartheta_i^{(R)}||\vartheta_j^{(R)}| \mathcal{O}(\tfrac{k^2}{m^2}) \|\mathbf{A}^{(t)}\| \|\mathbf{A}^{(t)}\| = \mathcal{O}(\tfrac{k^2 t_\beta^2}{m}) \|\mathbf{A}^*\|^2.$$

Combining all the results for $\clubsuit$, $\spadesuit$, $\heartsuit$ and $\diamondsuit$, we have,

$$\|\mathbf{E}[(\mathbf{y} - \mathbf{A}^{(t)} \widehat{\mathbf{x}})(\mathbf{y} - \mathbf{A}^{(t)} \widehat{\mathbf{x}})^\top]\|$$
$$= \mathcal{O}(\tfrac{k^2}{m}) \|\mathbf{A}^*\|^2 + \mathcal{O}(\tfrac{k^2}{m}) \epsilon_t \|\mathbf{A}^*\|^2 + \mathcal{O}(\tfrac{k^2}{m}) \epsilon_t \|\mathbf{A}^*\|^2 + \mathcal{O}(\tfrac{k^2 t_\beta^2}{m}) \|\mathbf{A}^*\|^2,$$
$$= \mathcal{O}(\tfrac{k^2 t_\beta^2}{m}) \|\mathbf{A}^*\|^2.$$

$\square$

# E ADDITIONAL EXPERIMENTAL RESULTS

We now present some additional results to highlight the features of NOODL. Specifically, we compare the performance of NOODL (for both dictionary and coefficient recovery) with the state-of-the-art provable techniques for DL presented in Arora et al. (2015) (when the coefficients are recovered via a sparse approximation step after DL)[3]. We also compare the performance of NOODL with the popular online DL algorithm in Mairal et al. (2009), denoted by Mairal '09. Here, the authors show that alternating between a $\ell_1$-based sparse approximation and dictionary update based on block co-ordinate descent converges to a stationary point, as compared to the true factors in case of NOODL.

**Data Generation:** We generate a $(n = 1000) \times (m = 1500)$ matrix, with entries drawn from $\mathcal{N}(0, 1)$, and normalize its columns to form the ground-truth dictionary $\mathbf{A}^*$. Next, we perturb $\mathbf{A}^*$ with random Gaussian noise, such that the unit-norm columns of the resulting matrix, $\mathbf{A}^{(0)}$ are $2/\log(n)$ away from $\mathbf{A}^*$, in $\ell_2$-norm sense, i.e., $\epsilon_0 = 2/\log(n)$; this satisfies the initialization assumptions in A.4. At each iteration, we generate $p = 5000$ samples $\mathbf{Y} \in \mathbb{R}^{1000 \times 5000}$ as $\mathbf{Y} = \mathbf{A}^* \mathbf{X}^*$, where $\mathbf{X}^* \in \mathbb{R}^{m \times p}$ has at most $k = 10$, 20, 50, and 100, entries per column, drawn from the Radamacher distribution. We report the results in terms of relative Frobenius error for all the experiments, i.e., for a recovered matrix $\widehat{\mathbf{M}}$, we report $\|\widehat{\mathbf{M}} - \mathbf{M}^*\|_F / \|\mathbf{M}^*\|_F$. To form the coefficient estimate for Mairal '09 via Lasso (Tibshirani, 1996) we use the FISTA (Beck and Teboulle, 2009) algorithm by searching across 10 values of the regularization parameter at each iteration. Note that, although our phase transition analysis for NOODL shows that $p = m$ suffices, we use $p = 5000$ in our convergence analysis for a fair comparison with related techniques.

---

[3]The associated code is made available at https://github.com/srambhatla/NOODL.

### E.1 COEFFICIENT RECOVERY

Table 4 summarizes the results of the convergence analysis shown in Fig. 2. Here, we compare the dictionary and coefficient recovery performance of NOODL with other techniques. For `Arora15(``biased'')` and `Arora15(``unbiased'')`, we report the error in recovered coefficients after the HT step ($\mathbf{X}_{\mathrm{HT}}$) and the best error via sparse approximation using Lasso[4] Tibshirani (1996), denoted as $\mathbf{X}_{\mathrm{Lasso}}$, by scanning over 50 values of regularization parameter. For `Mairal '09` at each iteration of the algorithm we scan across 10 values[5] of the regularization parameter, to recover the best coefficient estimate using Lasso ( via FISTA), denoted as $\mathbf{X}_{\mathrm{Lasso}}$.

We observe that NOODL exhibits significantly superior performance across the board. Also, we observe that using sparse approximation after dictionary recovery, when the dictionary suffers from a bias, leads to poor coefficient recovery[6], as is the case with `Arora15(``biased'')`, `Arora15(``unbiased'')`, and `Mairal '09`. This highlights the applicability of our approach in real-world machine learning tasks where coefficient recovery is of interest. In fact, it is a testament to the fact that, even in cases where dictionary recovery is the primary goal, making progress on the coefficients is also important for dictionary recovery.

In addition, the coefficient estimation step is also online in case of NOODL, while for the state-of-the-art provable techniques (which only recover the dictionary and incur bias in estimation) need additional sparse approximation step for coefficient recovery. Moreover, these sparse approximation techniques (such as Lasso) are expensive to use in practice, and need significant tuning.

### E.2 COMPUTATIONAL TIME

In addition to these convergence results, we also report the computational time taken by each of these algorithms in Table 4. The results shown here were compiled using 5 cores and 200GB RAM of Intel Xeon $E5 - 2670$ Sandy Bridge and Haswell E5-2680v3 processors.

The primary takeaway is that although NOODL takes marginally more time per iteration as compared to other methods when accounting for just one Lasso update step for the coefficients, it (a) is in fact faster per iteration since it does not involve any computationally expensive tuning procedure to scan across regularization parameters; owing to its geometric convergence property (b) achieves orders of magnitude superior error at convergence, and as a result, (c) overall takes significantly less time to reach such a solution. Further, NOODL's computation time can be further reduced via implementations using the neural architecture illustrated in Section 4.

Note that since the coefficient estimates using just the HT step at every step may not yield a usable result for `Arora15(``unbiased'')` and `Arora15(``biased'')` as shown in Table 4, in practice, one has to employ an additional $\ell_1$-based sparse recovery step. Therefore, for a fair comparison, we account for running sparse recovery step(s) using Lasso (via the Fast Iterative Shrinkage-Thresholding Algorithm (FISTA) (Beck and Teboulle, 2009) ) at every iteration of the algorithms `Arora15(``biased'')` and `Arora15(``unbiased'')`.

For our technique, we report the average computation time taken per iteration. However, for the rest of the techniques, the coefficient recovery using Lasso (via FISTA) involves a search over various values of the regularization parameters (10 values for this current exposition). As a result, we analyze the computation time per iteration via two metrics. First of these is the average computation time taken per iteration by accounting for the average time take per Lasso update (denoted as "*Accounting for one Lasso update*"), and the second is the average time taken per iteration to scan over all (10) values of the regularization parameter (denoted as "*Overall Lasso search*") .

---

[4]We use the Fast Iterative Shrinkage-Thresholding Algorithm (FISTA) (Beck and Teboulle, 2009), which is among the most efficient algorithms for solving the $\ell_1$-regularized problems. Note that, in our experiments we fix the step-size for FISTA as $1/L$, where $L$ is the estimate of the Lipschitz constant (since $\mathbf{A}$ is not known exactly).

[5]Note that, although scanning across 50 values of the regularization parameter for this case would have led to better coefficient estimates and dictionary recovery, we choose 10 values for this case since it is very expensive to scan across 50 of regularization parameter at each step. This also highlights why `Mairal '09` may be prohibitive for large scale applications.

[6]When the dictionary is not known exactly, the guarantees may exist on coefficient recovery only in terms of closeness in $\ell_2$-norm sense, due to the error-in-variables (EIV) model for the dictionary (Fuller, 2009; Wainwright, 2009).

**Table 4:** Final error in recovery of the factors by various techniques and the computation time taken per iteration (in seconds) corresponding to Fig. 2 across techniques. We report the coefficient estimate after the HT step (in Arora et al. (2015)) as $\mathbf{X}_{\mathrm{HT}}$. For the techniques presented in Arora et al. (2015), we scan across 50 values of the regularization parameter for coefficient estimation using Lasso after learning the dictionary ($\mathbf{A}$), and report the optimal estimation error for the coefficients ($\mathbf{X}_{\mathrm{Lasso}}$), while for Mairal '09, at each step the coefficients estimate is chosen by scanning across 10 values of the regularization parameters. For $k = 100$, the algorithms of Arora et al. (2015) do not converge (shown as N/A).

| Technique | Recovered Factor and Timing | k = 10 | k = 20 | k = 50 | k = 100 |
|---|---|---|---|---|---|
| NOODL | **A** | $9.44 \times 10^{-11}$ | $8.82 \times 10^{-11}$ | $9.70 \times 10^{-11}$ | $7.33 \times 10^{-11}$ |
| | **X** | $1.14 \times 10^{-11}$ | $1.76 \times 10^{-11}$ | $3.58 \times 10^{-11}$ | $4.74 \times 10^{-11}$ |
| | **Avg. Time/iteration** | 46.500 sec | 53.303 sec | 64.800 sec | 96.195 sec |
| Arora15 (''biased'') | **A** | 0.013 | 0.031 | 0.137 | N/A |
| | $\mathbf{X}_{\mathrm{HT}}$ | 0.077 | 0.120 | 0.308 | N/A |
| | $\mathbf{X}_{\mathrm{Lasso}}$ | 0.006 | 0.018 | 0.097 | N/A |
| | **Avg. Time/iteration** (*Accounting for one Lasso update*) | 39.390 sec | 39.371 sec | 39.434 sec | 40.063 sec |
| | **Avg. Time/iteration** (*Overall Lasso search*) | 389.368 sec | 388.886 sec | 389.566 sec | 395.137 sec |
| Arora15 (''unbiased'') | **A** | 0.011 | 0.027 | 0.148 | N/A |
| | $\mathbf{X}_{\mathrm{HT}}$ | 0.078 | 0.122 | 0.371 | N/A |
| | $\mathbf{X}_{\mathrm{Lasso}}$ | 0.005 | 0.015 | 0.0921 | N/A |
| | **Avg. Time/iteration** (*Accounting for one Lasso update*) | 567.830 sec | 597.543 sec | 592.081 sec | 686.694 sec |
| | **Avg. Time/iteration** (*Overall Lasso search*) | 917.809 sec | 947.059 sec | 942.212 sec | 1041.767 sec |
| Mairal '09 | **A** | 0.009 | 0.015 | 0.021 | 0.037 |
| | $\mathbf{X}_{\mathrm{Lasso}}$ | 0.183 | 0.209 | 0.275 | 0.353 |
| | **Avg. Time/iteration** (*Accounting for one Lasso update*) | 39.110 sec | 39.077 sec | 39.163 sec | 39.672 sec |
| | **Avg. Time/iteration** (*Overall Lasso search*) | 388.978 sec | 388.614 sec | 389.512 sec | 394.566 sec |

As shown in Table 4, in comparison to NOODL the techniques described in Arora et al. (2015) still incur a large error at convergence, while the popular online DL algorithm of Mairal et al. (2009) exhibits very slow convergence rate. Combined with the convergence results shown in Fig. 2, we observe that due to NOODL's superior convergence properties, it is overall faster and also geometrically converges to the true factors. This again highlights the applicability of NOODL in practical applications, while guaranteeing convergence to the true factors.

# F    APPENDIX: STANDARD RESULTS

**Definition 6** (sub-Gaussian Random variable)**.** Let $x \sim \mathrm{subGaussian}(\sigma^2)$. Then, for any $t > 0$, it holds that

$$\mathbf{Pr}[|x| > t] \leq 2 \exp\left(\tfrac{t^2}{2\sigma^2}\right).$$

## F.1    CONCENTRATION RESULTS

**Lemma 10** (Matrix Bernstein (Tropp, 2015))**.** Consider a finite sequence $\mathbf{W}_k \in \mathbb{R}^{n \times m}$ of independent, random, centered matrices with dimension $n$. Assume that each random matrix satisfies $\mathbf{E}[\mathbf{W}_k] = 0$ and $\|\mathbf{W}_k\| \leq R$ almost surely. Then, for all $t \geq 0$,

$$\mathbf{Pr}\big\{\|\textstyle\sum_k \mathbf{W}_k\| \geq t\big\} \leq (n+m)\exp\big(\tfrac{-t^2/2}{\sigma^2 + Rt/3}\big),$$

where $\sigma^2 := \max\{\|\sum_k \mathbf{E}[\mathbf{W}_k \mathbf{W}_k^\top]\|, \|\sum_k \mathbf{E}[\mathbf{W}_k^\top \mathbf{W}_k]\|\}$.

Furthermore,

$$\mathbf{E}[\|\textstyle\sum_k \mathbf{W}_k\|] \leq \sqrt{2\sigma^2 \log(n+m)} + \tfrac{1}{3}R \log(n+m).$$

**Lemma 11** (Vector Bernstein (Tropp, 2015)). Consider a finite sequence $\mathbf{w}_k \in \mathbb{R}^n$ of independent, random, zero mean vectors with dimension $n$. Assume that each random vector satisfies $\mathbf{E}[\mathbf{w}_k] = 0$ and $\|\mathbf{w}_k\| \leq R$ almost surely. Then, for all $t \geq 0$,

$$\mathbf{Pr}\big\{\|\sum_k \mathbf{w}_k\| \geq t\big\} \leq 2n\exp\big(\frac{-t^2/2}{\sigma^2 + Rt/3}\big),$$

where $\sigma^2 := \|\sum_k \mathbf{E}[\|\mathbf{w}_k\|^2]\|$. Furthermore,

$$\mathbf{E}[\|\sum_k \mathbf{w}_k\|] \leq \sqrt{2\sigma^2 \log(2n)} + \tfrac{1}{3}R\log(2n).$$

**Lemma 12. Chernoff Bound for sub-Gaussian Random Variables** Let $w$ be an independent sub-Gaussian random variables with variance parameter $\sigma^2$, then for any $t > 0$ it holds that

$$\mathbf{Pr}[|w| > t] \leq 2\exp(-\tfrac{t^2}{2\sigma^2}).$$

**Lemma 13** (Sub-Gaussian concentration (Rudelson and Vershynin, 2013)). Let $\mathbf{M} \in \mathbb{R}^{n \times m}$ be a fixed matrix. Let $\mathbf{w}$ be a vector of independent, sub-Gaussian random variables with mean zero and variance one. Then, for an absolute constant $c$,

$$\mathbf{Pr}[\|\mathbf{M}\mathbf{x}\|_2 - \|\mathbf{M}\|_F > t] \leq \exp(-\tfrac{ct^2}{\|\mathbf{M}\|^2}).$$

## F.2 RESULTS FROM (ARORA ET AL., 2015)

**Lemma 14** ((Arora et al., 2015) Lemma 45). Suppose that the distribution of $\mathbf{Z}$ satisfies $\mathbf{Pr}[\|\mathbf{Z}\| \geq L(\log(1/\rho))^C] \leq \rho]$ for some constant $C > 0$, then

1. If $p = n^{\mathcal{O}(1)}$ then $\|\mathbf{Z}^{(j)}\| \leq \widetilde{\mathcal{O}}(L)$ holds for each $j$ with probability at least$(1 - \rho)$ and,

2. $\|\mathbf{E}[\mathbf{Z}\mathbb{1}_{\|\mathbf{Z}\| \geq \widetilde{\Omega}(L)}]\| = n^{-\omega(1)}$.

In particular, if $\frac{1}{p}\sum_{j=1}^{p}\mathbf{Z}^{(j)}(1 - \mathbb{1}_{\|\mathbf{Z}\| \geq \widetilde{\Omega}(L)})$ is concentrated with probability $(1 - \rho)$, then so is $\frac{1}{p}\sum_{j=1}^{p}\mathbf{Z}^{(j)}$.

**Lemma 15** (Theorem 40 (Arora et al., 2015)). Suppose random vector $\mathbf{g}^{(t)}$ is a $(\rho_-, \rho_+, \zeta_t)$-correlated with high probability with $\mathbf{z}^*$ for $t \in [T]$ where $T \leq poly(n)$, and $\eta_A$ satisfies $0 < \eta_A \leq 2\rho_+$, then for any $t \in [T]$,

$$\mathbf{E}[\|\mathbf{z}^{(t+1)} - \mathbf{z}^*\|^2] \leq (1 - 2\rho_-\eta_A)\|\mathbf{z}^{(t)} - \mathbf{z}^*\| + 2\eta_A\zeta_t.$$

In particular, if $\|\mathbf{z}^{(0)} - \mathbf{z}^*\| \leq \epsilon_0$ and $\zeta_t \leq (\rho_-)o((1 - 2\rho_-\eta)^t)\epsilon_0^2 + \zeta$, where $\zeta = \max_{t\in[T]}\zeta_t$, then the updates converge to $\mathbf{z}^*$ geometrically with systematic error $\zeta/\rho_-$ in the sense that

$$\mathbf{E}[\|\mathbf{z}^{(t+1)} - \mathbf{z}^*\|^2] \leq (1 - 2\rho_-\eta_A)^t\epsilon_0^2 + \zeta/\rho_-.$$

