# OpenReview forum: "NOODL: Provable Online Dictionary Learning and Sparse Coding"
_ICLR.cc/2019/Conference_

### Official Review · AnonReviewer3 · 2018-10-29
**We found this work interesting. We think that some issues need to be further addressed by the authors.**

**Rating:** 7
**Confidence:** 2

**Review:**

The main contributions of this work are essentially on the theoretical aspects. It seems that the proposed algorithm is not very original because its two parts, namely prediction (coefficient estimation) and learning (dictionary update) have been widely used in the literature, using respectively a IHT and a gradient descent. The authors need to describe in detail the algorithmic novelty of their work.

The definition of “recovering true factor exactly” need to be given. The proposed algorithm involves several tuning parameters, when alternating between two updating rules, an IHT-based update for coefficients and a gradient descent-based update for the dictionary. Therefore, an appropriate choice of their values need to be given.

In the algorithm, the authors need to define the HT function in (3) and (4).

In the experiments, the authors compare the proposed method to only the one proposed by Arora et al. 2015. We think that this is not enough, and more extensive experimental results would provide a better paper.

There are some typos that can be easily found, such as “of the out algorithm”.

---

> ### Author Response · Authors · 2018-11-18
> **Additional comparative experimental evaluations of convergence and computational time, clarifications on tuning and recovery guarantees**
>
> We are grateful to the reviewer for the comments. In this revision, we have corrected the minor typos, added additional comparisons, and added a proof map for easier navigation of the results. Specific comments are addressed below.
>
> 1. Regarding exact recovery guarantees — NOODL converges geometrically to the true factors. Therefore, the error drops exponentially with iterations t. In other words, as t —> infinity A_i —> A^*_i for i in [1,m] and x_j —> x^*_j for j in [1,m], where x_j is in R^m. We have added this clarification in Section 1.1.
>
> 2. On tuning parameters — There are primarily three tuning parameters, namely eta_x (step-size for the IHT step), tau (threshold for the IHT step), and eta_A (step-size for the dictionary update step.) Our main result prescribes the theoretical values of these as shown in assumptions A.5 and A.6. Here, eta_x = Omega_tilde(k/sqrt(n)), tau = Omega_tilde(kˆ2/n), and eta_A = Theta(m/k). We have updated A.6. to include the order of these parameters.
>
> The specific choices of these parameters, like other similar problems, depend on some a priori unknown parameters (e.g. the sparsity k, and the incoherence mu) which makes some level of tuning unavoidable. This is true for Arora '15 and Mairal '09, as well, where tuning is required for the choice of step-size for dictionary update, and for choice of regularization parameter and the step-size for coefficient estimation via FISTA. Note that, in our experiments we fix the step-size for FISTA as 1/L, where L is the estimate of the Lipschitz constant (since A is not known exactly).
>
> Alternately, since NOODL involves gradient-based updates for the coefficients and the dictionary, tuning (the step-sizes and the threshold) is relatively straightforward in practice, since it is based on a gradient descent strategy. In fact, to compile the experiments presented in this paper, we fixed step-size, eta_x, and threshold, tau, and tuned the step-size parameter eta_A only (Theta(m/k)). The choices of eta_A are 30 for k = 10,20 and eta_A = 15 for k =50,100, as shown in Fig.2., eta_A mostly effects the convergence rate as long as it is chosen in Theta(m/k).
>
> Also, as shown in Table 4 (Appendix E), the tuning process for l1-based algorithms (i.e. FISTA) takes more time, since one needs to scan over the range of the regularization parameter to find one that works. This (a) adds to the computational time, and (b) since the dictionary is not known exactly, may guarantee recovery of coefficients only in terms of closeness in l2-norm sense, due to the error-in-variables (EIV) model for the dictionary. In this sense, NOODL is (a) simple to tune, (b) assures guaranteed recovery of both factors, and (c) is fast due to its geometric convergence properties. These factors highlight its applicability in practical DL problems.
>
> 3. Definition of Hard Thresholding (HT) — As per the recommendation of the reviewer, we have repeated the definition of hard-thresholding (HT) initially presented in the "Notation" sub-section, in Section 2 for clarity.
>
> 4. Comparison to other Online DL algorithms — As correctly observed by the reviewer, the overall structure of NOODL is similar to successful online DL algorithms. These successful algorithms (such as Mairal '09) leverage the progress made on both factors for convergence, however, do not guarantee recovery of the factors. On the other hand, the state-of-the-art provable DL algorithms focus on the progress made on only one of factors (the dictionary), and do not have good performance in practice, since they incur a non-negligible bias; see Section 5 and Appendix E. NOODL bridges the gap between these two. In addition to our main theoretical result, which establishes conditions for exact recovery of both factors at a geometric rate, NOODL also has superior empirical performance, leading to a neurally-plausible practical online DL algorithm with strong guarantees; see Section 3 and 4. Our work also paves way for the development and analysis of related alternating optimization-based techniques.
>
> On reviewer's recommendation, we compare the performance of NOODL with one of the most popular alternating minimization-based online DL algorithm used in practice -- Mairal `09 -- in Fig. 2 and Table 4 (Appendix E). In this work, the authors show that alternating between a l1-based sparse approximation step and dictionary update based on block co-ordinate descent converges to a stationary point. The other comparable techniques shown in Table 1, are not ``online’’ and/or require stringent initializations, in terms of closeness to the true dictionary, as compared to NOODL.
>
> Our experiments show that due to the geometric convergence to the true factors, NOODL outperforms competing state-of-the-art provable online DL techniques both in terms of overall computational time, and convergence performance. These additional expositions further showcase the contributions of our work both on theoretical and practical online DL front.

---

### Official Review · AnonReviewer1 · 2018-11-02
**Interesting paper, clear contribution. More intuition would be great.**

**Rating:** 6
**Confidence:** 2

**Review:**

The paper deals with the problem of recovering an exact solution for both the dictionary and the activation coefficients. As other works, the solution is based on a proper initialization of the dictionary. The authors suggest using Aurora 2015 as a possible initialization. The contribution improves Arora 2015 in that it converges linearly and recovers both the dictionary and the coefficients with no bias.

The main contribution is the use of a IHT-based strategy to update the coefficients, with a gradient-based update for the dictionary (NOODL algorithm). The authors show that, combined with a proper initialization, this has exact recovery guaranties. Interestingly, their experiments show that NOODL converges linearly in number of iterations, while Arora gets stuck after some iterations.

I think the paper is relevant and proposes an interesting contribution. The paper is well written and the key elements are in the body. However, there is a lot of important material in the Appendix, which I think may be relevant to the readers. It would be nice to have some more intuitive explanations at least of Theorem 1. Also, it is clear in the experiments the superiority with respect to Arora in terms of iterations (and error), but what about computational time?

---

> ### Author Response · Authors · 2018-11-18
> **Additional intuition behind the main result, and comparative evaluation of computational time**
>
> We thank the reviewer for the comments. As correctly observed by the reviewer, Arora et. al. 2015 suffers from a bias in estimation both in the analysis and in the empirical evaluations. The source of this bias term is an irreducible error in the coefficient estimate (formed using the hard-thresholding step). NOODL overcomes this issue by introducing a iterative hard-thresholding (IHT)-based coefficient update step, which removes the dependence of the error in estimated coefficient on this irreducible error, and ultimately the dictionary estimate.
>
> Intuitively, this approach highlights the symbiotic relationship between the two unknown factors — the dictionary and the coefficients. In other words, to make progress on one, it is imperative to make progress on the other. To this end, in Theorem 1 we first show that the coefficient error only depends on the dictionary error (given an appropriate number of IHT iterations R), i.e. we remove the dependence on x_0 which is the source of bias in Arora et. al. 2015. We have added the intuition corresponding to this in the revised paper after the statement of Theorem 1 in Section 3.
>
> Analysis of Computational Time — We have added the average per iteration time taken by various algorithms considered in our analysis in Table~4 and Appendix E.
> The primary takeaway is that although NOODL takes marginally more time per iteration as compared to other methods when accounting for just one (Lasso-based) sparse recovery for coefficient update, it (a) is in fact faster per iteration since it does not involve any computationally expensive tuning procedure to scan across regularization parameters; owing to its geometric convergence property (b) achieves orders of magnitude superior error at convergence, and as a result, (c) overall takes significantly less time to reach such a solution; see Appendix E for details.
>
> We would like to add that since NOODL involves simple separable update steps, this computation time can be further lowered by distributing the processing of individual samples across cores of a GPU (e.g. via TensorFlow) by utilizing the architecture shown in Fig. 1. We plan to release all the relevant code as a package in the future.
>
> In this revision, we have added comparison to Mairal '09, a popular online DL algorithm. Further, we have also added a proof map, in addition to the Table 3, for easier navigation of the results.

---

### Official Review · AnonReviewer2 · 2018-11-05
**A novel, alternating minimization algorithm for sparse coding**

**Rating:** 7
**Confidence:** 2

**Review:**

The paper considers the problem of dictionary learning. Here the model that we are given samples y, where we know that y = Ax where A is a dictionary matrix, and x is a random sparse vector. The goal is typically to recover the dictionary A, from which one can also recover the x under suitable conditions on A. The paper shows that there is an alternating optimization-based algorithm for this problem that under standard assumptions provably converges exactly to the true dictionary and the true coefficients x (up to some negligible bias).

The main comparison with prior work is with [1]. Both give algorithms of this type for the same problem, with similar assumptions (although there is some difference; see below). In [1], the authors give two algorithms: one with a better sample complexity than the algorithm presented here, but which has some systematic, somewhat large, error floor which it cannot exceed, and another which can obtain similar rates of convergence to the exact solution, but which requires polynomial sample complexity (the explicit bound is not stated in the paper). The algorithm here seems to build off of the former algorithm; essentially replacing a single hard thresholding step with an IHT-like step. This update rule is able to remove the error floor and achieve exact recovery. However, this makes the analysis substantially more difficult.

I am not an expert in this area, but this seems like a nice and non-trivial result. The proofs are quite dense and I was unable to verify them carefully.

Comments:

- The analysis in [1] handles the case of noisy updates, whereas the analysis given here only works for exact updates. The authors claim that some amount of noise can be tolerated, but do not quantify how much.

- A.4 makes it sound like eps_t needs to be assumed to be bounded, when all that is required is the bound on eps_0.

[1] Arora, S. Ge, R., Ma, T. and Moitra, A. Simple, Efficient, and Neural Algorithms for Sparse Coding. COLT 2015.

---

> ### Author Response · Authors · 2018-11-18
> **Clarifications on noise tolerance properties and assumption A.4.**
>
> We would like to thank the reviewer for the comments and for raising some subtle yet important questions. We address and clarify specific comments below. We have also made corresponding changes in the revised paper, and have added a proof map, in addition to the Table 3, for easier navigation of the results. We have also added comparisons with Mairal `09, and experimental evaluation of computational time.
>
> 1. Noise Tolerance — NOODL also has similar tolerance to noise as Arora et. al. 2015 and can be used in noisy settings as well. We focus on the noiseless case here to convey the main idea, since the analysis is already very involved. Nevertheless, the proposed algorithm can tolerate i.i.d. sub-Gaussian noise, including Gaussian noise and bounded noise, as long as the ``noise’’ is dominated by the ``signal’’. Under the noisy case, the recovered dictionary and coefficients will converge to a neighborhood of the true factors, where the neighborhood is defined by the properties of the additive noise.
>
> In other words, the noise terms will lead to additional terms which will need to be controlled for the convergence analysis. Specifically, the noise will add a term to the coefficient update in Lemma 2, and will effect the threshold, tau. For the dictionary, the noise will result in additional terms in Lemma 9 (which ensures that the updated dictionary maintains the closeness property). A precise characterization of the relationship between the level of noise the size of convergence neighborhood requires careful analysis, which we defer to future effort.
>
> 2. On eps_t and A.4. —  Indeed, we don’t need to assume that eps_t is bounded. Specifically, using the result of Lemma 7, we have that eps_0 undergoes a contraction at every step, therefore, eps_t <= eps_0. For our analysis we fix eps_t = O^*(1/log(n)), which follows from the assumption on eps_0= O^*(1/log(n)) and Lemma 7.  On reviewer’s comments, we have updated A.4., and moved the note about eps_t = O^*(1/log(n)) to the Appendix A.
>
> 3. Exact recovery of factors — Also, we would like to point that NOODL recovers both the dictionary and coefficients exactly at a geometric rate. This means that as t—> infinity both the dictionary and coefficients estimates converge to the true factors without incurring any bias. We have added a clarification corresponding to this in the revised paper in Section 1.1 and after the statement of Theorem 1 in Section 3.

---

### Meta-Review · Area_Chair1 · 2018-12-17
**contribution towards tractable dictionary learning and sparse coding.**

**Confidence:** 4
**Recommendation:** Accept (Poster)

**Metareview:**

Alternating minimization is surprisingly effective for low-rank matrix factorization and dictionary learning problems. Better theoretical characterization of these methods is well motivated. This paper fills up a gap by providing simultaneous guarantees for support recovery as well as coefficient estimates for  linearly convergence to the true factors, in the online learning setting. The reviewers are largely in agreement that the paper is well written and makes a valuable contribution.  The authors are advised to address some of the review comments around relationship to prior work highlighting novelties.